# Mesoderm-derived PDGFRA⁺ cells regulate the emergence of hematopoietic stem cells in the dorsal aorta

Vashe Chandrakanthan [1,2,18] ✉, Prunella Rorimpandey[1,2], Fabio Zanini[1,3,4,5], Diego Chacon[6], Jake Olivier[7], Swapna Joshi[1,2], Young Chan Kang[1,2], Kathy Knezevic[1,2], Yizhou Huang[3,7,8], Qiao Qiao[1], Rema A. Oliver[9], Ashwin Unnikrishnan [1,3], Daniel R. Carter[3,7,8], Brendan Lee[10], Chris Brownlee[10], Carl Power[10], Robert Brink[11,12], Simon Mendez-Ferrer [13], Grigori Enikolopov[14], William Walsh[9], Berthold Göttgens [13], Samir Taoudi[15,16], Dominik Beck[6] and John E. Pimanda [1,2,17,18] ✉

**Mouse haematopoietic stem cells (HSCs) first emerge at embryonic day 10.5 (E10.5), on the ventral surface of the dorsal aorta, by endothelial-to-haematopoietic transition. We investigated whether mesenchymal stem cells, which provide an essential niche for long-term HSCs (LT-HSCs) in the bone marrow, reside in the aorta–gonad–mesonephros and contribute to the development of the dorsal aorta and endothelial-to-haematopoietic transition. Here we show that mesoderm-derived PDGFRA⁺ stromal cells (*Mesp1*der PSCs) contribute to the haemogenic endothelium of the dorsal aorta and populate the E10.5–E11.5 aorta–gonad–mesonephros but by E13.5 were replaced by neural-crest-derived PSCs (*Wnt1*der PSCs). Co-aggregating non-haemogenic endothelial cells with *Mesp1*der PSCs but not *Wnt1*der PSCs resulted in activation of a haematopoietic transcriptional programme in endothelial cells and generation of LT-HSCs. Dose-dependent inhibition of PDGFRA or BMP, WNT and NOTCH signalling interrupted this reprogramming event. Together, aorta–gonad–mesonephros *Mesp1*der PSCs could potentially be harnessed to manufacture LT-HSCs from endothelium.**

Haematopoietic stem cells (HSCs) have extensive self-renewal capacity and are the source of daughter cells that proliferate, mature and develop into blood cells of all types[1]. As such, understanding the rules that govern HSC emergence, proliferation and maturation is important to reproduce these phenomena in vitro[2]. Advances in knowledge of embryonic haematopoiesis have informed methods that have been used to produce HSC-like cells in vitro[3–5].

The haematopoietic system in the embryo develops in successive waves[6]. The first blood progenitors to emerge (from the extra-embryonic yolk sac) are primitive erythrocytes, followed by erythroid–myeloid progenitors[7]. In mouse embryos, the first HSCs appear mid-gestationally (embryonic day 10.5, E10.5) from haemogenic endothelial cells[8–10] lining the ventral surface of the dorsal aorta through endothelial-to-haematopoietic transition (EHT)[11,12] in a region known as the aorta–gonad–mesonephros (AGM)[13,14]. These HSCs are amplified in the fetal liver[13] and the placenta[15,16]; they take up residence in the bone marrow, which will serve as the major adult site of haematopoiesis. Haemogenic endothelium is specified between E8.5 and E10.5 (ref. [17]) and progresses through pre-HSC stages to generate HSCs in the AGM between E10.5 and E12.5 (ref. [18]). The development of HSCs in the AGM is influenced by NOTCH[19], WNT[20], BMP[21,22] and other signals[23,24] from surrounding cells[25]. These signals facilitate haematopoiesis in part by regulating the expression of critical haematopoietic transcription factors, including components of the FLI1, GATA2 and SCL transcriptional network, GFI1–GFI1B and RUNX1 (refs. [26–28]).

Mesenchymal stem cells (MSCs) and their progeny are important constituents of the niche that regulates the size of the HSC pool in adult bone marrow[29,30]. Although there are resident stromal cells in the AGM that support haematopoiesis[31,32], little is known about their developmental origins, transcriptional and functional identity and contributions to the generation of long-term repopulating HSCs (LT-HSCs).

## Results

**PDGFRA⁺ stromal cells (PSCs) in the AGM have MSC properties.** Although the existence of stromal cells in the AGM that support haematopoiesis is known and AGM-derived stromal cell lines

[1]Adult Cancer Program, Lowy Cancer Research Centre, UNSW Sydney, Sydney, NSW, Australia. [2]Department of Pathology, School of Medical Sciences, UNSW Sydney, Sydney, NSW, Australia. [3]School of Clinical Medicine, UNSW Sydney, Sydney, NSW, Australia. [4]Garvan-Weizmann Centre for Cellular Genomics, Sydney, Australia. [5]UNSW Futures Institute for Cellular Genomics, Sydney, Australia. [6]Centre for Health Technologies and the School of Biomedical Engineering, University of Technology Sydney, Sydney, NSW, Australia. [7]School of Mathematics and Statistics, UNSW Sydney, Sydney, NSW, Australia. [8]Children's Cancer Institute Australia for Medical Research, Lowy Cancer Research Centre, UNSW Sydney, Sydney, NSW, Australia. [9]Surgical & Orthopaedic Research Laboratories, Prince of Wales Clinical School, UNSW Sydney, Sydney, NSW, Australia. [10]Biological Resources Imaging Laboratory, Mark Wainwright Analytical Centre, Lowy Cancer Research Centre, UNSW Sydney, Sydney, NSW, Australia. [11]Garvan Institute of Medical Research, Sydney, NSW, Australia. [12]UNSW Sydney, Sydney, NSW, Australia. [13]Wellcome Trust-Medical Research Council Cambridge Stem Cell Institute and Department of Haematology, University of Cambridge, Cambridge, UK. [14]Center for Developmental Genetics and Department of Anesthesiology, Stony Brook University, Stony Brook, NY, USA. [15]Epigenetics and development division, Walter and Eliza Hall Institute, Melbourne, VIC, Australia. [16]Department of Medical Biology, University of Melbourne, Melbourne, VIC, Australia. [17]Department of Haematology, The Prince of Wales Hospital, Sydney, NSW, Australia. [18]These authors contributed equally: Vashe Chandrakanthan, John E. Pimanda. ✉e-mail: v.chandrakanthan@unsw.edu.au; jpimanda@unsw.edu.au

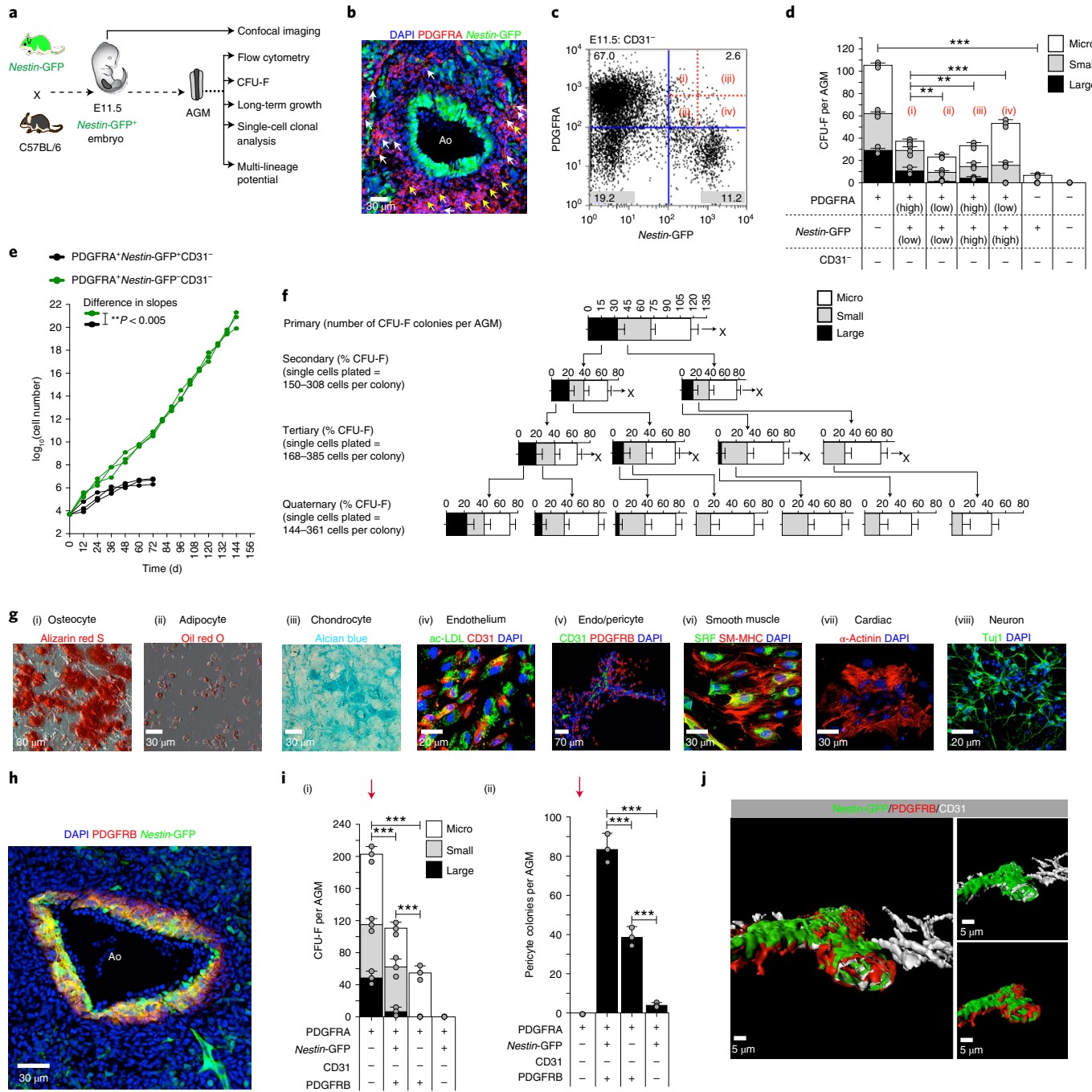

**g**
(i) Osteocyte — Alizarin red S
(ii) Adipocyte — Oil red O
(iii) Chondrocyte — Alcian blue
(iv) Endothelium — ac-LDL CD31 DAPI
(v) Endo/pericyte — CD31 PDGFRB DAPI
(vi) Smooth muscle — SRF SM-MHC DAPI
(vii) Cardiac — α-Actinin DAPI
(viii) Neuron — Tuj1 DAPI

have proven to be a powerful tool for the identification of environmental HSC regulators[21,31,33], we lack knowledge of the characteristics of these cells and their influence on EHT. It has previously been reported that all bone-marrow MSCs in *Nestin*-GFP transgenic mice (where expression of green fluorescent protein (GFP) is regulated by *Nestin*)[34] were GFP[+] and that ablation of these bone-marrow MSCs resulted in significant loss of LT-HSCs in 12–16-week-old mice[30]. We therefore used *Nestin*-GFP transgenic mice to investigate stromal cell populations in the AGM (Fig. 1a).

Confocal imaging of the E11.5 AGM of these mice showed that aortic endothelial and sub-endothelial blood cells as well as blood cells adjacent to the aortic endothelium were *Nestin*-GFP[+] (Extended Data Fig. 1a). Both *Nestin*-GFP[+] and *Nestin*-GFP[−] stromal cell fractions in the E11.5 AGM were found to express platelet-derived growth factor receptor alpha (PDGFRA), a tyrosine

kinase receptor expressed on the surface of MSCs[35] (Fig. 1b,c) and on early embryonic mesodermal cells that contribute to haemogenic endothelium and haematopoietic cells[36]. These PDGFRA[+] cells (*Nestin*-GFP[−], yellow arrows; and *Nestin*-GFP[+], white arrows) were distributed deeper in the aortic parenchyma and surrounded the PDGFRA[−]*Nestin*-GFP[+] cells, which were more concentrated towards the aortic lumen (Fig. 1b).

To explore the transitional functional properties of PDGFRA and *Nestin*-GFP-expressing and non-expressing cells in the AGM, we used an in vitro colony-forming unit–fibroblast (CFU-F) assay[37]. We first assessed the CFU-F potential in E9.5–E13.5 AGMs; we noted that colonies were composed of cells of mesenchymal cell morphology and varied in size[38] and that their numbers peaked at E11.5 (Extended Data Fig. 1b(i),(ii)). Freshly isolated fluorescence-activated-cell (FAC)-sorted PDGFRA[+]*Nestin*-GFP[−]

**Fig. 1 | The E11.5 AGM has resident long- and short-term repopulating CFU-Fs that can be discriminated by expression of PDGFRA and *Nestin*-GFP.**
**a**, Schematic outline of experiments performed using E11.5 *Nestin*-GFP⁺ embryos. **b**, Confocal image of an E11.5 *Nestin*-GFP⁺ dorsal aorta stained for PDGFRA. *Nestin*-GFP⁻ and *Nestin*-GFP⁺ PDGFRA⁺ cells are indicated with yellow and white arrows, respectively. **c**, Flow cytometry analysis of E11.5 *Nestin*-GFP⁺ AGM (*n* = 3) showing that 1:5.3 CD31⁻*Nestin*-GFP⁺ stromal cells are also PDGFRA⁺. The percentages of cells in the different quadrants (delineated in blue) are indicated. (i)–(iv) The PDGFRA⁺*Nestin*-GFP⁺ cells were further fractionated into high and low positive subpopulations. **d**, CFU-F potential of E11.5 CD31⁻*Nestin*-GFP⁺ AGM (*n* = 5) cells, sorted based on CD31, PDGFRA and *Nestin*-GFP expression according to the gating strategy shown in **c**. **e**, Long-term growth of E11.5 *Nestin*-GFP⁺ AGM-derived CFU-Fs based on CD31, *Nestin*-GFP and PDGFRA expression. **f**, Single-cell clonal analysis of CD31⁻PDGFRA⁺*Nestin*-GFP⁻ CFU-Fs. The CFU-F colony numbers are representative of *n* = 4 (primary plating) and *n* = 11–15 (secondary–quaternary plating). **g**, In vitro differentiation of CD31⁻PDGFRA⁺*Nestin*-GFP⁻ cells (*n* = 3); ac-LDL, acetylated apoprotein low-density lipoprotein. **h**, Confocal microscopy image of an E11.5 *Nestin*-GFP⁺ dorsal aorta showing that a subset of PDGFRB⁺ cells co-express *Nestin*-GFP (*n* = 3). PDGFRB⁺*Nestin*-GFP⁺ cells are stained in yellow; PDGFRB⁺*Nestin*-GFP⁻ cells are stained in red. **i**, (i) CFU-Fs in FAC-sorted fractions from E11.5 *Nestin*-GFP⁺ AGMs (*n* = 4). (ii) Pericyte colony-forming potential in FAC-sorted fractions from E11.5 *Nestin*-GFP⁺ AGMs (*n* = 7). **j**, Z-stack reconstruction of confocal microscopy images showing vessel-like structures lined by *Nestin*-GFP⁺ endothelial cells with surrounding PDGFRB⁺ pericytes. These images were taken from tissues harvested 3 weeks after subcutaneous transplantation of a Matrigel plug loaded with PDGFRA⁺*Nestin*-GFP⁻CD31⁻PDGFRB⁻ FAC-sorted cells from E11.5 *Nestin*-GFP⁺ AGMs (see red arrows in **i**(i) and (ii)); CD31 staining is shown in white. **d**–**f**,**i**, Data represent the mean ± s.d. **d**,**i**, Data were derived from *n* = 3 biologically independent experiments. **d**,**e**,**i**, A random-effects Poisson regression was used to compare colony counts (**d**,**i**) and a linear mixed model was used to compare the growth curves (**e**); **P < 0.01, ***P < 0.005. Ao, aortic lumen; DAPI, 4′,6-diamidino-2-phenylindole dihydrochloride. Colony sizes: micro colonies, <2 mm, 2–24 cells; small colonies, 2–4 mm, >25 cells; and large colonies, >4 mm, >100 cells. Precise *P* values are provided in the source data.

and PDGFRA⁺*Nestin*-GFP⁺ populations also produced CFU-Fs of different sizes (Fig. 1d). Although both PDGFRA⁺*Nestin*-GFP⁻ and PDGFRA⁺*Nestin*-GFP⁺ cells produced large CFU-F colonies (Fig. 1d), their number was lower in the latter and proportionate to the intensity of PDGFRA and *Nestin*-GFP expression. Only PDGFRA⁺*Nestin*-GFP⁻ cells showed long-term replating capacity (Fig. 1e).

Furthermore, serial replating of single cells from PDGFRA⁺*Nestin*-GFP⁻ large CFU-F colonies produced consistent numbers of large colonies of CFU-Fs (Fig. 1f), and these cells could be differentiated in vitro into mesodermal and ectodermal derivatives (Fig. 1g(i)–(viii) and Supplementary Video 1). By contrast, single cells from PDGFRA⁺*Nestin*-GFP⁺ large CFU-F colonies showed limited capacity to generate large colonies of CFU-Fs (Extended Data Fig. 1c) and could only be differentiated into adipocytes, endothelium and smooth muscle (Extended Data Fig. 1d). The differentiation potential observed in bulk PDGFRA⁺*Nestin*-GFP⁻ and PDGFRA⁺*Nestin*-GFP⁺ cells was best reflected in cells that formed large CFU-F colonies (Extended Data Fig. 2a,b). Together, these data show that CFU-F potential in the E11.5 AGM resides largely in PDGFRA⁺ cells and that Nestin expression marks a subpopulation of PDGFRA⁺ cells with more restricted CFU-F and differentiation potential.

Pericytes are characterized by the expression of platelet-derived growth factor receptor beta (PDGFRB)[39,40] and were distributed concentrically in the sub-endothelium of the E11.5 dorsal aorta (Fig. 1h). To investigate the relationship between *Nestin*-GFP⁺ cells and pericytes, we fractionated cell populations (FAC-sorted) from E11.5 AGMs of *Nestin*-GFP transgenic mice based on PDGFRA, *Nestin*-GFP, CD31 and PDGFRB expression (Extended Data Fig. 2c) and performed assays for formation of CFU-F and pericyte colonies as well as a long-term replating assay (Fig. 1i(i),(ii) and Extended Data Fig. 2d). Among the CD31⁻PDGFRA⁺ cells, PDGFRB co-expression was proportionally higher in the *Nestin*-GFP⁺ subpopulation than the *Nestin*-GFP⁻ cells (Extended Data Fig. 2c). Although the latter showed the highest large-CFU-F colony and long-term replating potential, unlike the former cells, they lacked potential to form pericyte colonies (Fig. 1i(i),(ii) and Extended Data Fig. 2d). Interestingly, CFU-F potential in the *Nestin*-GFP⁺ fraction was exclusively within the PDGFRB⁺ subfraction (Fig. 1i(i)). We further assessed the contribution of different PDGFRA⁺ fractions (Fig. 1i) towards in vivo morphological and functional vascular contents. We purified PDGFRA⁺ fractions using flow cytometry, mixed those cells with Matrigel and transplanted them subcutaneously into C57BL/6 mice. Only purified CD31⁻PDGFRA⁺*Nestin*-GFP⁻PDGFRB⁻ CFU-Fs formed vessel-like structures (Fig. 1j, Extended Data Fig. 2e

**Fig. 2 | E9.5 PDGFRA⁺ cells contribute to haemogenic endothelium and LT-HSCs. a**, (i) Schematic outline of experiments performed using E8.5, E9.5, E10.5 and E11.5 *Pdgfra*–nGFP embryos. (ii) Confocal microscopy images of E.8.5, E9.5 and E10.5 *Pdgfra*–nGFP embryos showing the distribution of PDGFRA-expressing cells in relation to the developing aorta. (iii) Spatial distribution of *Pdgfra*–nGFP-, NESTIN- and CD31-expressing cells in a *Pdgfra*–nGFP E11.5 AGM. The region outlined in white in the main image (left) has been magnified (right) and different cell populations are labelled: I, *Pdgfra*–nGFP^high^NESTIN⁻CD31⁻; II, *Pdgfra*–nGFP^high^NESTIN⁺CD31⁻; III, *Pdgfra*–nGFP^low^NESTIN⁺CD31⁻; IV, *Pdgfra*–nGFP^low^NESTIN⁺CD31⁺; V, *Pdgfra*–nGFP⁻NESTIN⁻CD31⁺. **b**, (i) Schematic outline of lineage-tracing experiments using *Pdgfra*–cre^ERT2^; *R26R*–eYFP embryos. (ii) Confocal image of an E11.5 *Pdgfra*–cre^ERT2^; *R26R*–eYFP AGM following *cre* activation at E9.5 showing eYFP⁺ blood cells (white arrows) and endothelium (orange arrows). (iii) Contribution of donor eYFP⁺ cells to PDGFRA⁺ cells, pericytes (PDGFRB⁺), endothelium (CD31⁺) and blood cells (CD45⁺) in the E11.5 *Pdgfra*–cre^ERT2^; *R26R*–eYFP AGM following *cre* activation at E9.5. *Pdgfra*-eYFP; PDGFRA cells in the top left panel are boxed in pink, and the adjacent flow cytometry plot to the right (boxed in pink) shows corresponding CD45; CD31 expression. High and low *Pdgfra*-eYFP expressing cells in the flow cytometry plots to the left in each of the four panels in (iii) are boxed in red and green, respectively. Correspondingly coloured boxes to the right in each of the four panels show expanded phenotypic profiles for these cells. **c**, (i) Schematic outline of lineage-tracing experiments using *Pdgfra*–cre^ERT2^; *R26R*–eYFP embryos; e.e., embryonic equivalent. (ii) Contribution of donor eYFP⁺ cells to peripheral blood in primary and secondary transplants at 4 months post transplantation (*n* = 5). The arrow indicates the sample for which the expanded flow cytometry profiles are shown in (iii). (iii) Flow cytometry analysis of the contribution of donor eYFP⁺ cells to peripheral blood in primary transplant. **d**, (i) Schematic outline of lineage-tracing experiments using *Pdgfra*–cre^ERT2^; *R26R*–eYFP embryos. (ii) Confocal image of a *Pdgfra*–cre^ERT2^; *R26R*–eYFP neonatal long-bone section following *cre* activation at E9.5, showing eYFP⁺CD45⁺ blood cells in the bone marrow. (iii) Contribution of donor eYFP⁺ cells to peripheral blood (PB), bone marrow (BM), thymus and spleen in primary (6 months post transplantation; left) and secondary (4 months post tranplantation; right) transplants (*n* = 5). Ao, aortic lumen; NT, neural tube; DAPI, 4′,6-diamidino-2-phenylindole dihydrochloride; FSC-A, forward scatter area. The percentage of cells in the different quadrants in the flow cytometry plots are indicated. Data were derived from biologically independent samples, animals and experiments (*n* = 5). Data represent the mean ± s.d.

and Supplementary Video 2), the luminal surfaces of which were lined with *Nestin*-GFP⁺CD31⁺ endothelial cells and enveloped by PDGFRB⁺ pericytes.

**PSCs contribute to haemogenic endothelium and HSCs.** If PSCs are a reservoir for endothelial and sub-endothelial cells in the developing aorta, they could also contribute to long-term repopulating HSCs that emerge at E11.5. PDGFRA⁺ cells, when labelled at E7.5–E8, have previously been shown to contribute to blood cells budding from the endothelial lining of the dorsal aorta in E10.5 AGM as well as B, T and Lin⁻Kit⁺Sca-1⁺ (LSK) cells in the bone marrow of adult mice[36]. To explore whether cells expressing PDGFRA and CD31 proteins in the AGM were early and late constituents of a differentiation continuum, we evaluated the distribution of CD31 in the E8.5, E9.5, E10.5 and E11.5 AGMs of *Pdgfra*–nGFP knock-in mice (that is, mice whose *Pdgfra*-expressing cells retain GFP in their nuclei[41]; Fig. 2a(i)). *Pdgfra*–nGFP cells are in proximity with endothelial cells of the paired dorsal aorta at early embryonic time points (Fig. 2a(ii)). At E11.5 (Fig. 2a(iii)), cells furthest from the aortic lumen showed robust *Pdgfra*–nGFP^high expression but no NESTIN (NES) protein (layer I). Cells co-expressing both *Pdgfra*–nGFP^high and NES (layer II) were interspersed between these cells (layer I) and cells that were *Pdgfra*–nGFP^low but NES⁺ (layer III), which also co-expressed the smooth-muscle marker aSMA (Extended Data Fig. 3a). Endothelial cells lining the aortic lumen (layer V) were *Pdgfra*–nGFP⁻ and expressed CD31 but little or no NES (in contrast to the longer-lasting GFP in nGFP transgenic mice; Fig. 1a). A few cells were NES⁺ and CD31⁺ but low in *Pdgfra*–nGFP (layer IV). It is salient that

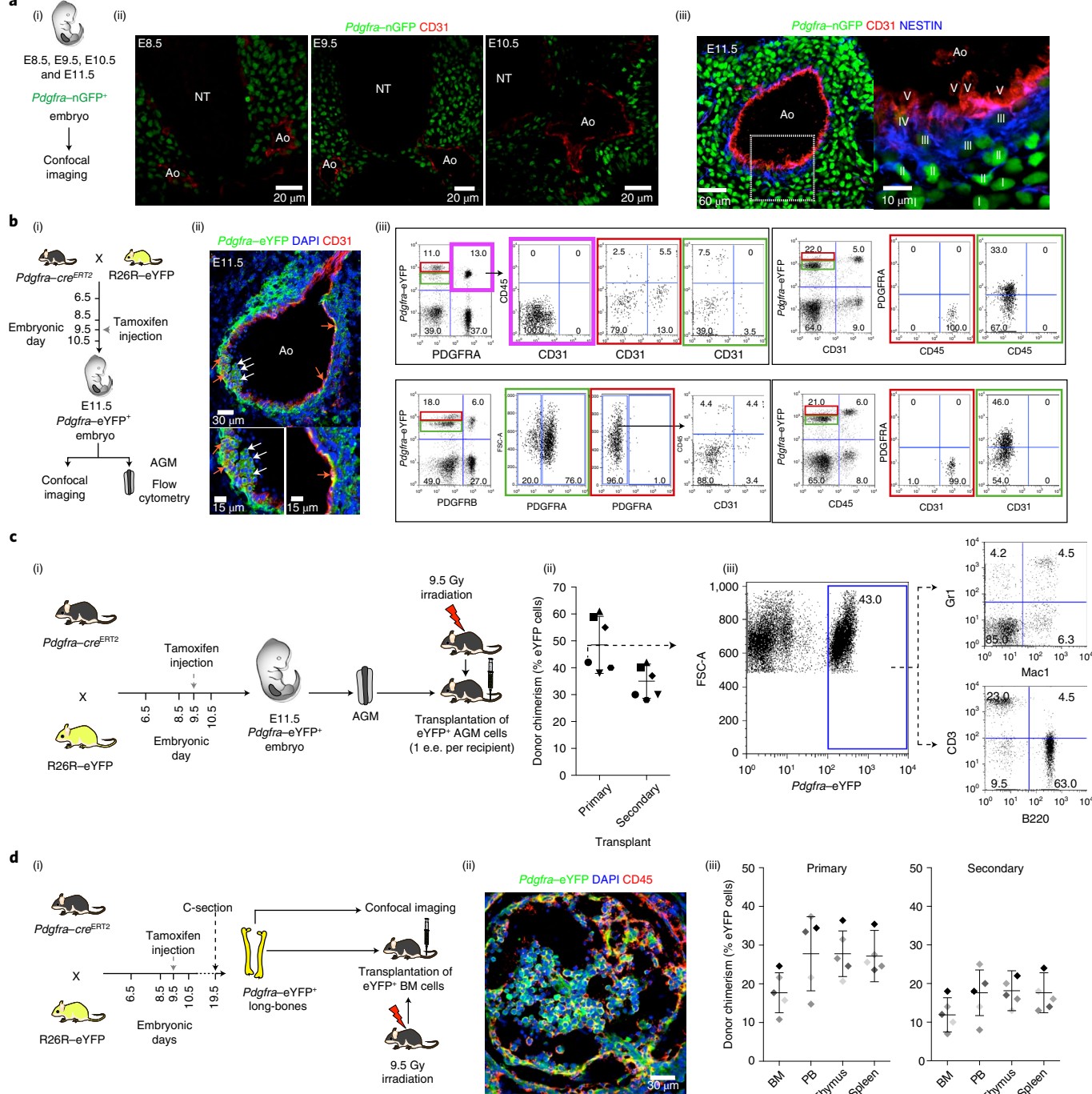

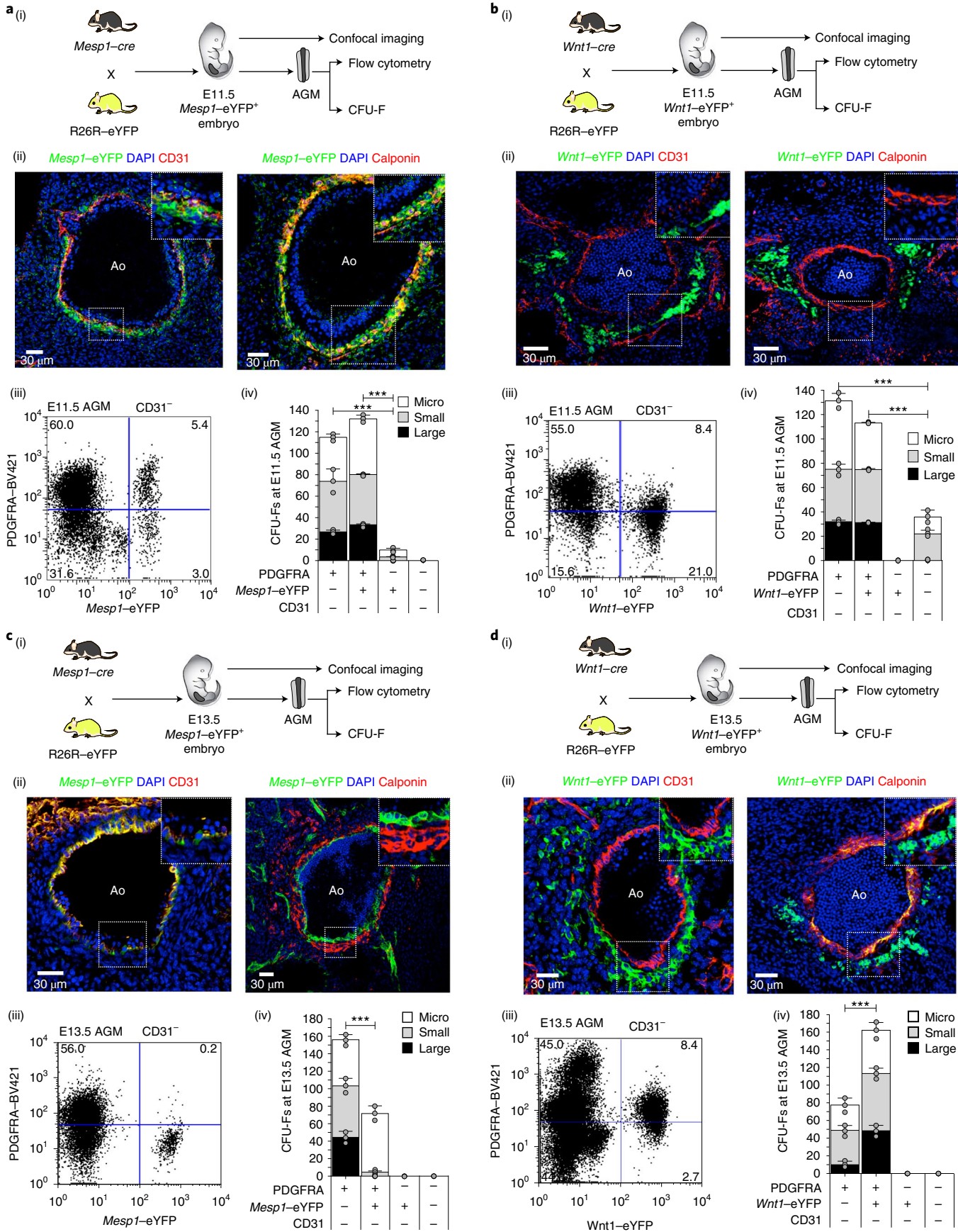

**Fig. 3 | Developmental origins of AGM endothelium and CFU-Fs. a**, (i) Schematic outlining the genetic cross used to harvest *Mesp1–cre;R26R*–eYFP (*Mesp1*–eYFP[+]) embryos at E11.5. (ii) Confocal microscopy images of E11.5 *Mesp1*–eYFP AGM showing the contribution of *Mesp1*-derived cells to the endothelium (left; CD31) and smooth muscle (right; Calponin). Insets: magnified views (2-fold) of the region in the white box in the main image. (iii) Percentage of CD31[−]*Mesp1*–eYFP[+]PDGFRA[+] cells in AGMs at E11.5, determined by flow cytometry. (iv) Number of CFU-Fs in cell fractions sorted from *Mesp1*–eYFP[+] AGMs (*n* = 7) at E11.5. **b**, (i) Schematic outlining the genetic cross used to harvest *Wnt1*–eYFP embryos at E11.5. (ii) Confocal microscopy images of E11.5 *Wnt1*–eYFP AGM showing the absence of contribution to endothelium (left; CD31), sub-endothelial smooth muscle (right; Calponin) and sub-endothelial stroma (left and right). Insets: magnified views (2-fold) of the region in the white box in the main image. (iii) Percentage of CD31[−]*Wnt1*–eYFP[+]PDGFRA[+] cells in AGMs at E11.5, determined by flow cytometry. (iv) Number of CFU-Fs in cell fractions sorted from *Wnt1*–eYFP[+] AGMs (*n* = 5) at E11.5. **c**, (i) Schematic outlining the genetic cross used to harvest *Mesp1*–eYFP[+] embryos at E13.5. (ii) Confocal microscopy images of E13.5 *Mesp1*–eYFP AGMs showing the contribution of *Mesp1*-derived cells to the endothelium (left; CD31) but not to smooth muscle (right; Calponin). Insets: magnified views (2-fold) of the region in the white box in the main image. (iii) Percentage of CD31[−]*Mesp1*–eYFP[+]PDGFRA[+] cells in AGMs at E13.5, determined by flow cytometry. (iv) Number of CFU-Fs in cell fractions sorted from *Mesp1*–eYFP[+] AGMs (*n* = 7) at E13.5. **d**, (i) Schematic outlining the genetic cross used to harvest *Wnt1*–eYFP embryos at E13.5. (ii) Confocal microscopy images of E13.5 *Wnt1*–eYFP AGM showing that *Wnt1*-derived cells do not contribute to the endothelium (left; CD31) but do contribute to smooth muscle (right; Calponin) and sub-endothelial stroma (left and right). Insets: magnified views (2-fold) of the region in the white box in the main image. (iii) Percentage of CD31[−]*Wnt1*–eYFP[+]PDGFRA[+] cells in AGMs at E13.5, determined by flow cytometry. (iv) Number of CFU-Fs in cell fractions sorted from *Wnt1*–eYFP[+] AGMs (*n* = 5) at E13.5. Ao, aortic lumen; DAPI, 4′,6-diamidino-2-phenylindole dihydrochloride; BV421, brilliant violet 421. Colony sizes: micro, <2 mm, 2–24 cells; small, 2–4 mm, >25 cells; and large; >4 mm, >100 cells. CFU-F data were derived from biologically independent experiments (*n* = 3) using 5–7 embryos per experiment. Data represent the mean ± s.d. The percentage of cells in the different quadrants in the flow cytometry plots are indicated. A random-effects Poisson regression was used to compare colony counts (**a–d**(iv)); ***$P < 0.005$. The precise *P* values are provided in the source data.

*Pdgfra*–nGFP[+] and PDGFRA[+] cells in the E11.5 AGM were comparable in their CFU-F potential and long-term growth potential (Extended Data Fig. 3b). When GFP[+]CD31[−]CD45[−]PDGFRA[+] cells were harvested from the E9.5 AGM of ubiquitous GFP mice and cultured on OP9 cells ex vivo, they contributed robustly to GFP[+]CD31[+]CD45[+] cells (Extended Data Fig. 3c).

To formally establish a lineage relationship between PDGFRA[+] cells at E9.5 and their progeny, we crossed *Pdgfra–cre*[ERT2][42] mice with R26R–enhanced yellow fluorescent protein (eYFP)[43] mice to generate *Pdgfra–cre*[ERT2];R26R–eYFP compound transgenic embryos (Fig. 2b(i)) and induced *cre* recombination at E9.5 by delivering single injections of tamoxifen to pregnant mothers and harvesting embryos at E11.5. CD31[+] endothelial cells in the E9.5 AGM do not express *Pdgfra* (Extended Data Fig. 3d(i),(ii))[44]. There was sufficient recombination with 6.4% of limb bud cells expressing eYFP following a single injection of tamoxifen at E9.5 (Extended Data Fig. 3e). Bearing in mind that PDGFRA[+] cells labelled at E7.5 and E8 also contribute to the endothelium of the dorsal aorta at E10.5 (ref. [36]), *Pdgfra–cre*[ERT2];R26R–eYFP recombination at E9.5 also resulted in eYFP[+] aortic endothelial, sub-endothelial and blood cells in the E11.5 AGM (Fig. 2b(ii),(iii)), marking approximately a third of all

CD31[+]CD45[+] cells (Extended Data Fig. 3f(i),(ii)). Only a minority of eYFP[+] cells still expressed PDGFRA protein (Fig. 2b(iii) and Extended Data Fig. 3g). The eYFP[+]PDGFRA[+] cells were CD31[−] and CD45[−] and had lower eYFP fluorescence than CD31[+] or CD45[+] cells (Fig. 2b). There were no eYFP[+]CD31[+] endothelial cells in the E11.5 yolk sac, placenta or umbilical and vitelline vessels (Extended Data Fig. 3h).

To evaluate whether these eYFP[+] cells included LT-HSCs, we again induced *cre* recombination in *Pdgfra–cre*[ERT2];R26R–eYFP compound transgenic embryos at E9.5, harvested E11.5 embryos and performed transplantation assays with eYFP[+] AGM cells (Fig. 2c(i)). These cells were able to reconstitute haematopoiesis in lethally irradiated mice following primary and secondary transplantation and contributed to multiple blood lineages (Fig. 2c(ii),(iii)). To establish whether *Pdgfra*–eYFP[+] cells populate the bone marrow, *Pdgfra–cre*[ERT2];R26R–eYFP compound transgenic embryos were matured to term following induction of recombination at E9.5 and delivered by caesarean section (owing to difficulties in parturition; Fig. 2d(i)). eYFP[+]CD45[+] blood cells were present in the bone marrow of the *Pdgfra–cre*[ERT2];R26R–eYFP compound neonatal mice (Fig. 2d(ii)). These cells were able to reconstitute haematopoiesis

**Fig. 4 | Co-aggregate cultures of endothelial cells with E11.5 *Mesp1*[der] PSCs generate endothelium-derived LT-HSCs. a**, (i) Schematic outlining the process for harvesting the cell types used in co-aggregate cultures. PSCs (150,000 cells) from E10.5 and E11.5 embryos were co-aggregated with endothelial cells from E10.5, E11.5 and E13.5 AGM, adult heart, lung, aorta and inferior vena cava (25,000 cells) and cultured for 96 h. The PSCs were DsRed[+] or DsRed[−] (*Mesp1*–DsRed[+/−]PDGFRA[+]PDGFRB[−]CD31[−]VE-Cad[−]CD41[−]CD45[−]); the endothelial cells were GFP[+] (*UBC*–GFP[+]PDGFRA[−]PDGFRB[−]CD31[+]VE-Cad[+]CD41[−]CD45[−]). CD31 staining is shown in white. (ii) Confocal microscopy images of cryosections of a co-aggregate of E11.5 *Mesp1*–DsRed[+] PSCs and E13.5 endothelial cells at 96 h showing GFP[+]CD45[+] cells. A magnified view (2-fold) of the region in the white box in the main image (top) is shown (bottom). DAPI, 4′,6-diamidino-2-phenylindole dihydrochloride. (iii) CFU-C potential of embryonic and adult aortic endothelium co-aggregated with E10.5 or E11.5 *Mesp1*–DsRed[+], or E11.5 *Mesp1*–DsRed[−] PSCs. BFU-E, burst-forming unit–erythroid; CFU-GM, colony-forming unit–granulocyte/macrophage; CFU-GEMM, colony-forming unit–granulocyte/erythrocyte/macrophage/megakaryocyte. (iv) Percentage of GFP[+] cells in peripheral blood of irradiated recipients 4–6 months after transplantation of co-aggregates (one co-aggregate for each adult irradiated recipient) using PSCs from E10.5 (left) and E11.5 (right) *Mesp1*–DsRed+ embryos. The coloured arrows and corresponding symbols (#, *, $) indicate groups of mice that were used in secondary transplants. (v) Percentage of GFP[+] cells in the peripheral blood of irradiated recipients 4 months after bone-marrow transplants from corresponding mice in (iv). **b**, (i) Schematic outline of the experimental procedure used to fractionate GFP[+]CD45[+] or GFP[+]CD45[−] cells and DsRed[+] E11.5 PSCs from co-aggregate cultures at 96 h for transplantation. (ii) Percentage of GFP[+] cells in the peripheral blood of irradiated recipients 4 months after primary transplantation of co-aggregate fractions (co-aggregates were pooled and sorted into GFP[+]CD45[+], GFP[+]CD45[−] or DsRed[+] fractions, and equivalent cell volumes of GFP[+]CD45[+] or GFP[+]CD45[−] cells were injected with or without DsRed[+] cells, such that each adult irradiated recipient received cells from approximately one co-aggregate) as shown. Data represent the mean ± s.d. Data were derived from biologically independent samples, animals and experiments (**a**, *n* = 3; **b**, *n* = 5). Ao, adult aorta; H, adult heart; L, adult lung; IVC, adult inferior vena cava. A random-effects Poisson regression was used to compare colony counts (**a**(iii)), and analysis of variance (ANOVA) was used to compare donor chimerism (**a**(iv),(v) and **b**(ii)); *$P < 0.05$, **$P < 0.01$, ***$P < 0.005$. The precise *P* values are provided in the source data.

in lethally irradiated mice following primary and secondary transplantation (Fig. 2d(iii)) and contributed to multiple blood lineages (Extended Data Fig. 3i,j).

Given the contributions of E9.5 PDGFRA[+] cells to structures of the aorta and blood cells that arise therein at E11.5, we predicted that ablation of these cells and their progeny would have a profoundly deleterious impact on the developing aorta and haematopoiesis. To

explore this, we crossed *Pdgfra–cre*[ERT2] mice[42] with inducible diphtheria toxin receptor (iDTR) mice[45] to generate *Pdgfra–cre*[ERT2];*iDTR* embryos (Extended Data Fig. 4a). We conditionally induced expression of diphtheria toxin receptor in E9.5 PDGFRA[+] cells through treatment with tamoxifen, followed by ablation of these cells using diphtheria toxin at E10.5 in *Pdgfra–cre*[ERT2];*iDTR* embryos (Extended Data Fig. 4a(i)). We then studied the resulting impact

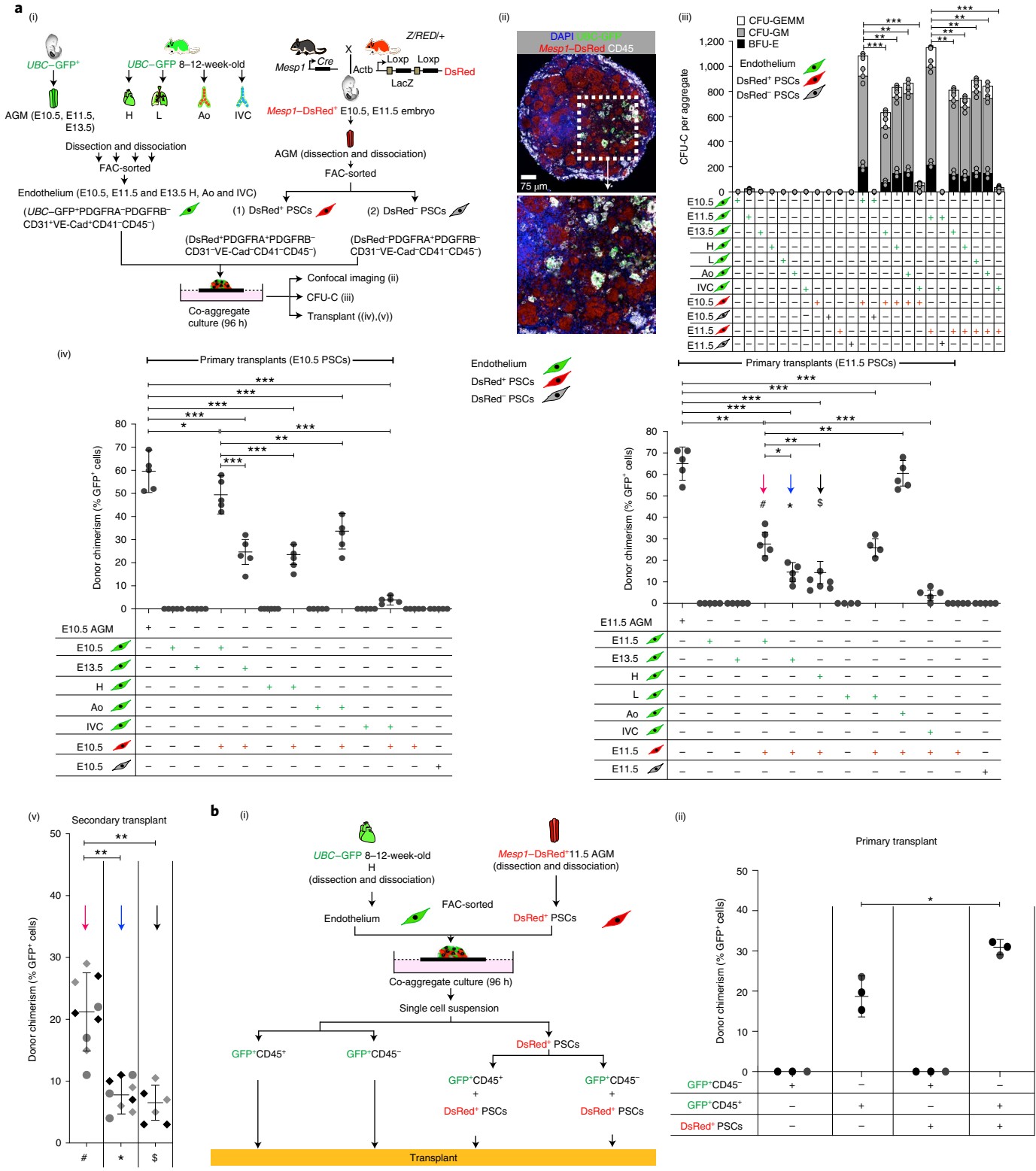

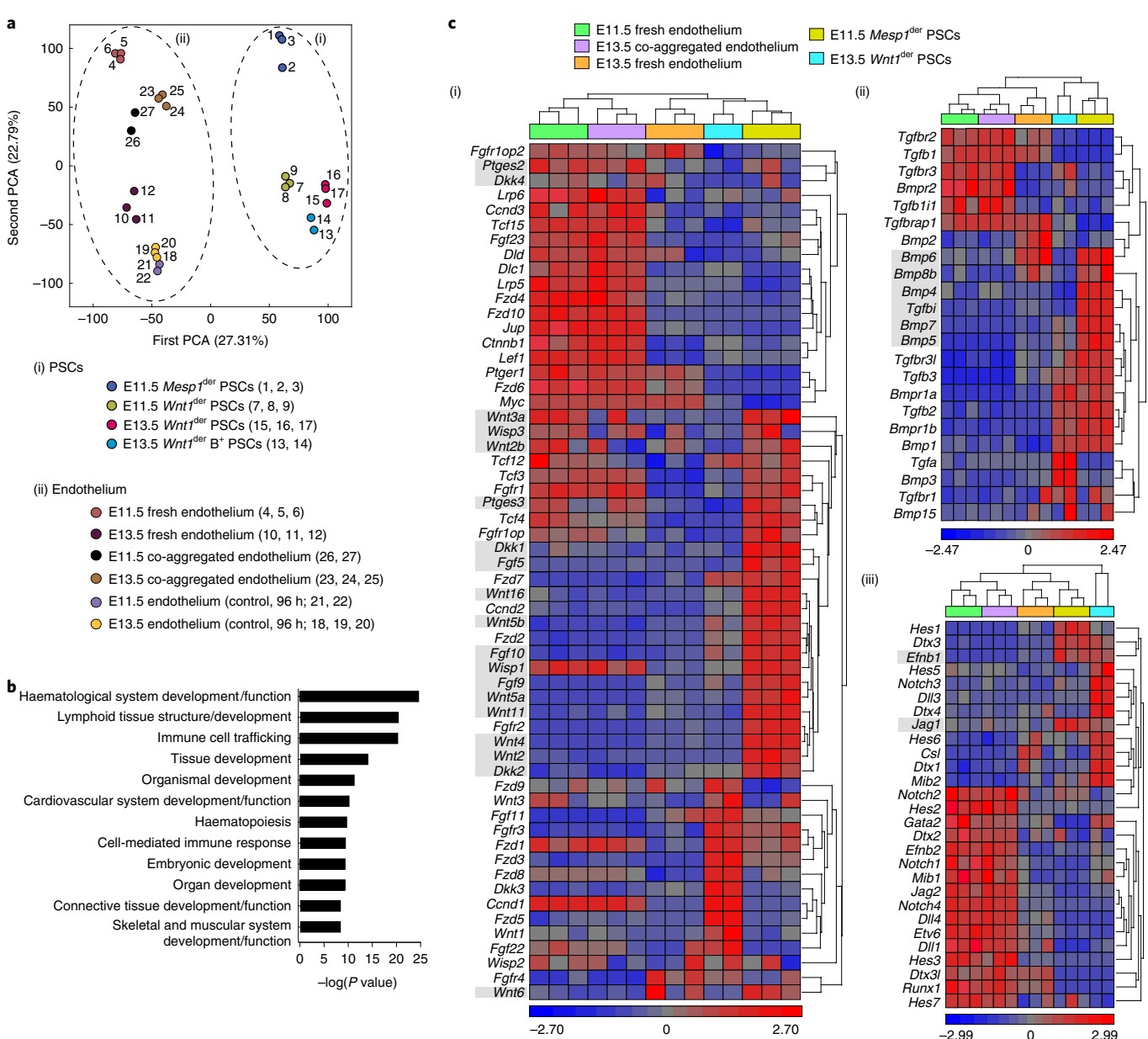

**Fig. 5 | Reciprocal ligand–receptor expression between *Mesp1*der PSCs and haemogenic endothelium. a**, Two-dimensional PCA plot generated from transcriptomes (top). (i) Fresh PSCs at E11.5 (*Mesp1*–eYFP+PDGFRA+, *Mesp1*der PSCs), at E11.5 and E13.5 (*Wnt1*–eYFP+ PDGFRA+, *Wnt1*der PSCs) and at E13.5 (*Wnt1*–eYFP+PDGFRB+, *Wnt1*der B+ PSCs). (ii) Endothelial cells: fresh at E11.5 and E13.5; E11.5 and E13.5 endothelial cells taken from co-aggregates with E11.5 *Mesp1*–eYFP+ PSCs (96 h); and E11.5 and E13.5 control endothelial cells without PSCs (96 h). **b**, Molecular functions annotated by Ingenuity Pathway Analysis of differentially expressed genes—first in E13.5 endothelial cells co-aggregated with E11.5 *Mesp1*–eYFP+ PSCs and second in E13.5 control fresh endothelium. **c**, Hierarchical clustering of gene expression profiles of haematopoietic mediators associated with WNT (i), BMP (ii) and NOTCH (iii) signalling in E11.5 *Mesp1*der PSCs, E13.5 *Wnt1*der PSCs and E13.5 endothelial cells following co-aggregation with *Mesp1*der PSCs and control E11.5 and E13.5 fresh endothelium. Ligands that were expressed in *Mesp1*der PSCs but not *Wnt1*der PSCs are highlighted in grey. Data were derived from biologically independent samples, animals and experiments (*n* = 3). Each replicate is displayed separately.

on the AGM architecture at E11.5. In whole-mount and tissue sections of compound transgenic embryos, there was severe disruption of normal dorsal aorta development (Extended Data Fig. 4a(ii)). In these embryos, there was concomitant reduction in the number of various cell types: endothelial (CD31+), blood (SCA1+CD45+), perivascular (PDGFRB+) and CFU-Fs (PDGFRA+; Extended Data Fig. 4a(iii)) as well as blood progenitors (Extended Data Fig. 4a(iv)) and CFU-Fs (Extended Data Fig. 4a(v)). These data indicate that the absence of PDGFRA-expressing cells in the developing embryo should have a profoundly deleterious impact on AGM

haematopoiesis. Mice carrying a targeted null mutation of *Pdgfra* show early embryonic lethality[46], and PDGFRA signalling has previously been reported to be essential for establishing a microenvironment that supports definitive haematopoiesis[47]. To directly test whether LT-HSCs were generated in the absence of *Pdgfra*, we crossed tdTomato/Rosa26;*Pdgfra*–nGFP knock-in (KI) heterozygote mice to generate *Pdgfra* KI/KI (null) and KI/+ (heterozygote) embryos with ubiquitous tdTomato expression, and performed colony-forming unit–culture (CFU-C) and transplantation assays with individual E10.5 and E11.5 AGMs from GFP+ KI

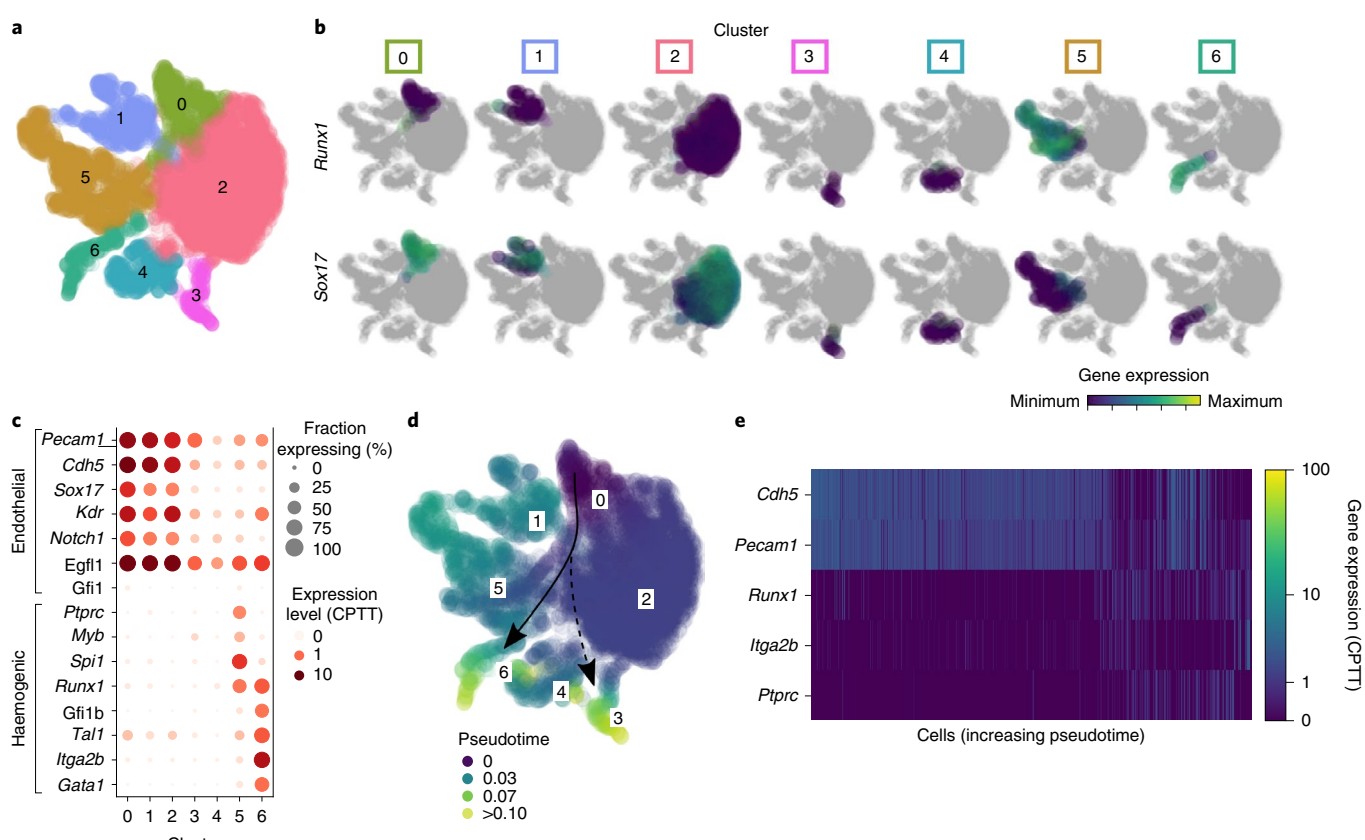

**Fig. 6 | *Mesp1*^der PSCs induce a haemogenic transcriptome in endothelial cells.** **a**, UMAP representation of single-cell transcriptomes from adult cardiac endothelial cells (*UBC–gfp* mice; GFP⁺CD31⁺VE-Cad⁺PDGFRA⁻) following 0, 24, 48, 72 and 96 h co-aggregated with *Mesp1*^der PSCs. Endothelial cells from each time point were pooled, and data from three independent pools are shown together. **b**, Distribution of *Runx1* (top) and *Sox17* (bottom) expression intensities in each cell cluster in **a**. **c**, Bubble plot representing the distribution of expression levels of various endothelial and haematopoietic genes in each cluster in **a**. The size of each bubble represents the fraction of cells in each cluster expressing that gene. **d**, UMAP embedding developmental progression of cells in **a** using pseudotime with cluster 0 as the starting reference. The arrows are guides to the eye indicating the expected transition from endothelial to haemogenic cells (solid arrow) and separately (dashed arrow) to a branching transcriptomic profile (cluster 3). **e**, Heatmap illustrating the expression levels of select endothelial and haematopoietic genes in single cells (GFP⁺CD31⁺VE-Cad⁺PDGFRA⁻ cells) progressing through pseudotime in **d**. Data were derived from biologically independent samples, animals and experiments (*n* = 3). The replicates were incorporated in the display. CPTT, counts per ten thousand unique molecular identifiers.

embryos with retrospective genotyping of yolk sacs (Extended Data Fig. 4b(i)). Consistent with our expectations, *Pdgfra*-null E10.5 and E11.5 AGMs produced significantly fewer CFU-Cs and no LT-HSCs (Extended Data Fig. 4b(ii),(iii)).

**Distinct waves of PSCs serially populate the AGM.** To investigate the source of AGM CFU-Fs, we first crossed R26R–eYFP mice with *Mesp1–cre* (mesoderm) mice[48] and harvested embryos for confocal imaging as well as AGM flow cytometry and CFU-F assays (Fig. 3a(i)). Lineage-tracing studies using *Mesp1–cre* mice have previously shown *Mesp1*-derived cell contributions to endothelial cells of the dorsal aorta[49]. At E11.5, CD31⁺ aortic endothelial cells were *Mesp1*–eYFP⁺ and surrounded by a rim of *Mesp1*–eYFP⁺CD31⁻ sub-endothelial cells (Fig. 3a(ii), left). Sub-endothelial cells expressing the smooth-muscle marker Calponin were also *Mesp1*–eYFP⁺ (Fig. 3a(ii), right)). A survey of E8.5, E9.5 and E10.5 AGMs in *Mesp1*–eYFP⁺ embryos showed that *Mesp1*-derived stromal cells also contributed to the aortic endothelium even at these early time points (Extended Data Fig. 5a). These data collectively show that at the time of HSC emergence, sub-endothelial stromal cells were mesodermal derivatives.

Approximately two-thirds of the *Mesp1*–eYFP⁺ cells were PDGFRA⁺ (Fig. 3a(iii)). In E11.5 *Pdgfra*–nGFP⁺;*Mesp1*–DsRed double-transgenic embryos, Pdgfra–nGFP⁺DsRed⁺ cells were distributed in the AGM stroma (Extended Data Fig. 5b(i),(ii)). In contrast, *Nestin*-GFP⁺ cells in *Nestin*-GFP⁺;*Mesp1*–DsRed double-transgenic embryos were largely restricted to endothelial and sub-endothelial cells in the E11.5 AGM (Extended Data Fig. 5b(iii),(iv)). Whereas *Mesp1*–eYFP⁻PDGFRA⁺ cells from E10.5 AGMs generated significantly lower numbers of CFU-Fs than the *Mesp1*–eYFP⁺PDGFRA⁺ cells (Extended Data Fig. 5c), this difference was not observed in E11.5 AGMs (Fig. 3a(iv)). PDGFRA⁻ cells on the other hand had limited CFU-F capacity at both time points and formed no large colonies. Together, these data show that the aortic endothelium, sub-endothelium and a proportion of CFU-Fs in the E11.5 AGM were derived from *Mesp1*⁺ cells but that a comparable number of CFU-Fs were not.

To explore whether the *Mesp1*–eYFP⁻ cells were derived from *Wnt1*⁺ cells, we next crossed R26R–eYFP mice with *Wnt1–cre* (neural crest) mice[50] and harvested embryos for confocal imaging as well as AGM flow cytometry and CFU-F assays (Fig. 3b(i)). In contrast to *Mesp1*–eYFP⁺ cells at corresponding embryonic time points (E8.5–E11.5), *Wnt1*–eYFP⁺ cells did not contribute to the endothelium or sub-endothelium and were located deeper in the AGM stroma (E11.5; Fig. 3b(ii)) or distant to the ventral surface of the dorsal aorta (E8.5–E10.5; Extended Data Fig. 5d). However,

one-third of the *Wnt1*–eYFP+ cells were PDGFRA+ (Fig. 3b(iii)), and these cells formed significantly fewer CFU-Fs than *Mesp1*–eYFP+PDGFRA+ cells at E10.5 (compare Extended Data Fig. 5c,e), but their contributions were comparable at E11.5 (compare Fig. 3a(iv),b(iv)). Although only a minority of PDGFRA+ cells were either *Mesp1*–eYFP+ (approximately 1:12; Fig. 3a(iii)) or *Wnt1*–eYFP+ (approximately 1:8; Fig. 3b(iii)), these two subfractions collectively accounted for CFU-F potential in the E10.5 and E11.5 AGM.

Unlike MSCs in the E14.5 embryonic trunk, which were reported to be derived from Sox1+ neuroepithelium[35], *Sox1*–eYFP+PDGFRA+ cells were very rare in the E11.5 AGM (2%) and did not contribute to large CFU-Fs (Extended Data Fig. 5f(i)–(iii)).

Therefore, the E11.5 AGM has at least two populations of PDGFRA+ CFU-Fs that have different lineage ancestries (*Mesp1*-derived, *Mesp1*der; and *Wnt1*-derived, *Wnt1*der) and occupy distinct anatomical locations with respect to the haemogenic endothelium. Furthermore, *Mesp1*der PSCs showed greater multilineage differentiation capacity than non-*Mesp1*der PSCs (Extended Data Fig. 5g).

In contrast to the haemogenic E10.5 and E11.5 AGM, the E13.5 AGM is no longer haemogenic[18]. To evaluate whether CFU-F populations changed during this transition, we crossed R26R–eYFP mice with *Mesp1*–cre mice and harvested embryos for confocal imaging as well as AGM flow cytometry and CFU-F assays at E13.5 (Fig. 3c(i)). Whereas the dorsal aorta was still lined by *Mesp1*–eYFP+CD31+ endothelial cells (Fig. 3c(ii), left), Calponin+ sub-endothelial cells were *Mesp1*–eYFP– (Fig. 3c(ii), right). Furthermore, *Mesp1*–eYFP+ PSCs, which were relatively abundant at E11.5 (5.4%; Fig. 3a(iii)), were rare at E13.5 (0.2%; Fig. 3c(iii)). Large-CFU-F potential in the E10.5 and E11.5 AGM was seen in both the *Mesp1*–eYFP+ and eYFP– PDGFRA+ fractions (Fig. 3a(iv) and Extended Data Fig. 5c), but in the absence of *Mesp1*–eYFP+PDGFRA+ cells at E13.5, they were derived exclusively from *Mesp1*–eYFP–PDGFRA+ cells (Fig. 3c(iv)).

We then crossed R26R–eYFP mice with *Wnt1*–cre mice and harvested embryos at E13.5 for confocal imaging as well as AGM flow cytometry and CFU-F assays (Fig. 3d(i)). As observed in E8.5–E11.5 embryos (Fig. 3b(ii), left and Extended Data Fig. 5d), there was no evidence of *Wnt1*–eYFP+-derived endothelial cells at E13.5 (Fig. 3d(ii), left), but the layer of *Mesp1*–eYFP+CD31–Calponin+ sub-endothelial cells that were evident at E11.5 had been replaced by *Wnt1*–eYFP+CD31–Calponin+ cells (Fig. 3d(ii), right). There were equal proportions (8.4%) of *Wnt1*–eYFP PSCs at E13.5 and E11.5 (compare Fig. 3b(iii),d(iii)). In the absence of *Mesp1*–eYFP+

PSCs at E13.5, large-CFU-F potential was mostly seen in *Wnt1*–eYFP+ PSCs (Fig. 3d(iv)).

It is important to note that *Mesp1* transcripts were absent in PDGFRA+ (CFU-F), PDGFRB+ (pericytes) and CD31+ (endothelial) cells in the AGM at both E11.5 and E13.5 (Extended Data Fig. 6a). Therefore, *Mesp1*–eYFP+ cells in the AGM at these time points are Mesp1-derived cells that do not currently express *Mesp1*. Although *Wnt1* transcripts were absent in E13.5 cells, there was variable and low-level *Wnt1* expression at E11.5 in PDGFRA+ but not PDGFRB+ or CD31+ cells (Extended Data Fig. 6b).

Together, these data show that at the time of HSC emergence at E11.5 (and E10.5), sub-endothelial stromal cells were mesodermal (that is, *Mesp1*) derivatives. The loss of *Mesp1*der cells in the sub-endothelium, along with replacement by *Wnt1*der cells at E13.5, temporally coincides with the loss of EHT in the dorsal aorta.

### *Mesp1*der PSCs induce EHT in non-haemogenic endothelium.

To determine whether there were EHT-promoting attributes in E10.5 and E11.5 *Mesp1*der PSCs that were absent in E11.5 and E13.5 *Wnt1*-derived progenitors, we performed co-aggregate cultures of FAC-sorted *Mesp1*der and *Wnt1*der PSCs with endothelial cells from ubiquitous GFP+ (*UBC*–*gfp*/BL6)[51] mice. The *Mesp1*der and *Wnt1*der PSCs were harvested from the AGMs of compound transgenic embryos generated by crossing *Mesp1*–cre or *Wnt1*–cre mice with STOCK Tg(CAG-Bgeo-DsRed*MST)1Nagy/J (Z/Red) reporter mice[52].

Endothelial cells (*UBC*–GFP+PDGFRA–PDGFRB–CD31+VE-cadherin(VE-Cad)+CD41–CD45–) from E10.5, E11.5 and E13.5 AGM or 12–16-week-old female adult mice (heart, lung, aorta and inferior vena cava) were co-aggregated with stromal cells (PDGFRA+PDGFRB–CD31–VE-Cad–CD41–CD45–) from E10.5 and E11.5 *Mesp1*–DSRed+ AGM (Fig. 4a(i) and Extended Data Fig. 6c(i),(ii)). Following 96 h of culture, the co-aggregates were cryosectioned for confocal imaging or used for flow cytometry, CFU-C and transplantation assays to establish progenitor and stem cell potential of emerging blood cells. Confocal microscopy and flow cytometry showed GFP+CD45+ cells in all endothelial and *Mesp1*der PSC co-aggregates (Fig. 4a(ii) and Extended Data Fig. 6d; E13.5 AGM endothelium and E11.5 *Mesp1*der PSCs). No DsRed+CD45+ cells were found in any co-aggregate. Co-aggregation of both E10.5 and E11.5 *Mesp1*der PSCs with E11.5 (haemogenic) or E13.5 (non-haemogenic) AGM or adult heart, lung, aortic or inferior vena cava endothelium resulted in the emergence of *UBC*–GFP+ CFU-Cs (Fig. 4a(iii)) and endothelial cell-derived (*UBC*–GFP+) LT-HSCs with robust multilineage haematopoietic reconstitution (Fig. 4a(iv),(v) and Extended

**Fig. 7 | PDGFRA-mediated cell signalling is important for LT-HSC generation. a**, (i) Hierarchical clustering of expression profiles of components of PDGFR signalling in E11.5 *Mesp1*der PSCs, E13.5 *Wnt1*der PSCs, E13.5 endothelial cells following co-aggregation with *Mesp1*der PSCs and control fresh E11.5 and E13.5 endothelium. (ii) Confocal microscopy images of an E11.5 *Pdgfra*–nGFP AGM showing high PDGF-A protein in the aortic endothelium (white arrows) and stroma (*n* = 5). (iii) Confocal microscopy images of a E13.5 *Pdgfra*–nGFP AGM showing low PDGF-A protein in the aortic endothelium and stroma (*n* = 5). CD31 staining is shown in white. **b**, (i) Schematic outlining the process for harvesting the cell types used in co-aggregate cultures. Co-aggregate cultures of GFP+ endothelial cells (25,000 cells; *UBC*–GFP+PDGFRA–PDGFRB–CD31+VE-Cad+CD41–CD45–) with DsRed+*Mesp1*der PSCs (150,000 cells; *Mesp1*-DSRed+PDGFRA+PDGFRB–CD31–VE-Cad–CD41–CD45–) were incubated in the absence or presence of increasing concentrations of the PDGFRA-specific inhibitor APA5 or PDGFRB-specific inhibitor APB5. (ii) CFU-C analysis of co-aggregate cultures of *Mesp1*der PSCs with E11.5 aortic endothelium at 96 h (*n* = 3). (iii) Percentage of GFP+ cells in irradiated recipients (one co-aggregate for each adult irradiated recipient) 6 months following transplantation with co-aggregate cultures (primary transplant; *n* = 5); e.e., embryonic equivalent. **c**, (i) Schematic outlining an experiment in which E10.0 embryos were cultured ex vivo in the absence or presence of APA5. (ii) Confocal microscopy images of the AGM at E11.5 in the absence (top) or presence (bottom) of APA5. Emergent blood clusters are indicated by yellow arrows. (iii) Flow cytometry analysis of cultured E11.5 embryo AGM in the absence (top) or presence (bottom) of APA5 showing a reduction in the number of SCA1+cKIT+ cells in the presence of APA5. The percentage of cells in each quadrant is indicated. (iv) CFU-C analysis of ex vivo cultured E11.5 embryo AGM in the absence or presence of APA5 (*n* = 3). (v) CFU-F analysis of ex vivo cultured E11.5 embryo AGM in the absence or presence of APA5 (*n* = 3). Ao, aortic lumen; CFU-GM, colony-forming unit–granulocyte/macrophage; CFU-GEMM, colony-forming unit–granulocyte/erythrocyte/macrophage/megakaryocyte; mix, CFU-C with indistinct boundary—possibly co-localization of separate CFU-C or an early split from a single CFU-C. Colony sizes: micro, <2 mm, 2–24 cells; small, 2–4 mm, >25 cells; and large, >4 mm, >100 cells. Data were derived from biologically independent samples, animals and experiments (**b**(ii),**c**, *n* = 3; **b**(iii), *n* = 5). Data represent the mean ± s.d. A random-effects Poisson regression was used to compare colony counts (**b**(ii),(iv),(v)), and an ANOVA was used to compare donor chimerism (**b**(iii)); **P < 0.01, ***P < 0.005. The precise P values are listed in the source data.

Data Fig. 7a,b). There were no PSC-derived (DsRed⁺) haematopoietic cells in any co-aggregate transplant recipient (Extended Data Fig. 7a(ii)–(iv)). Transplantation of aggregates composed of endothelial cells or PSCs alone did not contribute to haematopoietic cells (Extended Data Fig. 8a). Co-aggregates of E10.5 and E11.5 *Mesp1*der PSCs with inferior vena cava endothelium yielded CFU-Cs, but their number was lower than those from aortic endothelium with correspondingly low donor chimerism (Fig. 4a(iv),(v)). These data were suggestive of inherent differences in endothelial cell types that made them more or less receptive to EHT.

As these injected co-aggregates potentially contained transitional EHT cells and were being injected with *Mesp1*der PSCs, we also

fractionated GFP⁺ (endothelial-derived) CD45⁺ and CD45⁻ cells as well as *Mesp1*der PSCs, from 96-h adult heart endothelial cell–*Mesp1*der PSC co-aggregates (Extended Data Fig. 8b), and transplanted cells as shown (Fig. 4b(i)). Although GFP⁺CD45⁻ cells did not contribute to donor chimerism with or without *Mesp1*der PSCs, CD45⁺ cells in these co-aggregates engrafted and contributed to long-term donor chimerism with or without *Mesp1*der PSCs (Fig. 4b(ii)). Together, these data show that (1) endothelial cell-derived (GFP⁺) CD45⁺ cells in the co-aggregates could engraft and contribute to long-term reconstitution without ongoing support from *Mesp1*der PSCs, (2) *Mesp1*der PSCs could not facilitate in vivo maturation/engraftment of CD45⁻ cells and (3) donor chimerism

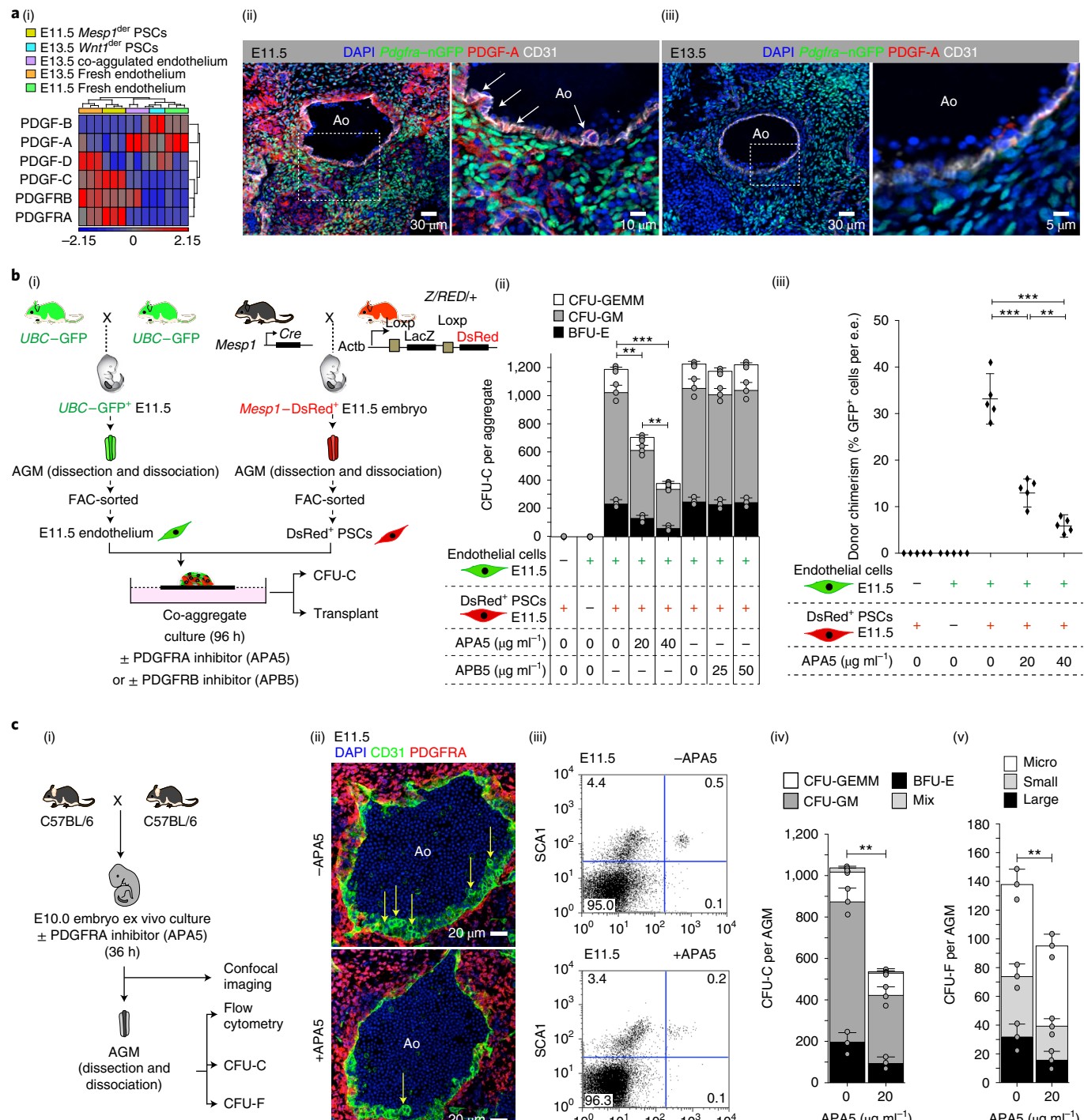

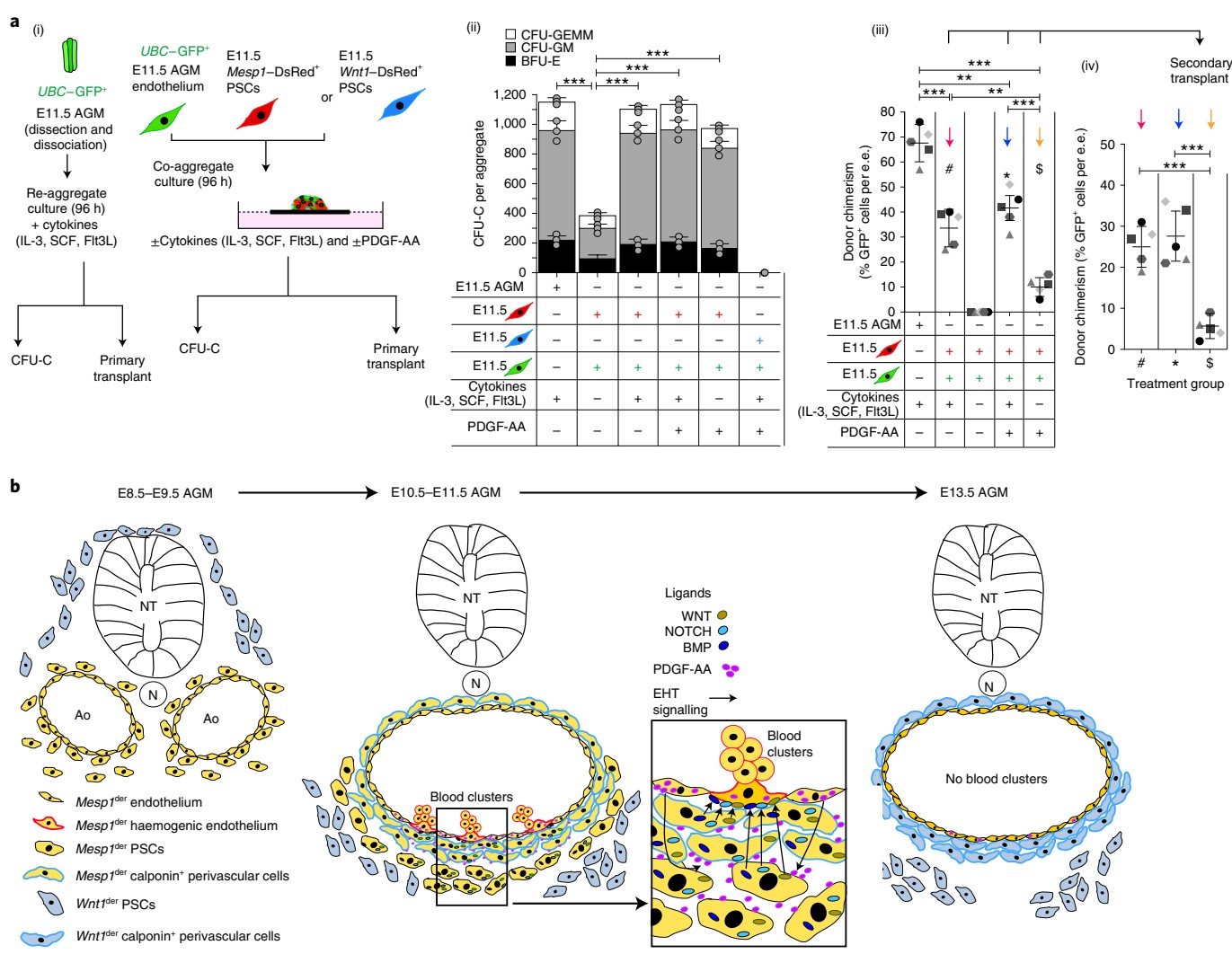

**Fig. 8 | PDGF-AA can partially compensate for the absence of cytokines in co-aggregate cultures. a**, (i) Schematic showing the experimental strategy for co-aggregation of E11.5 *UBC*–GFP⁺ aortic endothelial cells with *Mesp1*der PSCs or *Wnt1*der PSCs in the presence and absence of cytokines or PDGF-AA. E11.5 *UBC*–GFP⁺ AGM re-aggregates were used as controls. (ii) CFU-C assays from 96-h co-aggregate cultures (E11.5 *UBC*–GFP⁺ aortic endothelial cells and *Mesp1*der PSCs, *n* = 7–10 co-aggregates per culture condition; E11.5 *UBC*–GFP⁺ aortic endothelial cells and *Wnt1*der PSCs; *n* = 9–13 co-aggregates per culture condition). BFU-E, burst-forming unit–erythroid; CFU-GM, colony-forming unit–granulocyte/macrophage; and CFU-GEMM, colony-forming unit–granulocyte/erythrocyte/macrophage/megakaryocyte. (iii) Percentage of GFP⁺ cells in the peripheral blood of primary transplant recipients (*n* = 5 co-aggregates per culture condition); e.e., embryonic equivalent. (iv) Percentage of GFP⁺ cells in the peripheral blood of secondary transplant recipients from the indicated primary recipients. One co-aggregate per recipient mouse; each point represents an individual recipient. **b**, Model incorporating the changing landscape of *Mesp1*- and *Wnt1*-derived PSCs in the AGM stroma and their role in generating endothelial and sub-endothelial cells as well as LT-HSCs. Pre-E9.5 and E9.5 *Mesp1*der PSCs contribute to the aortic endothelium. E10.5 and E11.5 endothelium secretes PDGF-AA, which acts on PSCs that secrete haemogenic factors to promote EHT. *Mesp1*der PSCs are replaced by *Wnt1*der PSCs at E13.5. This is accompanied by the loss of high PDGF-AA in the AGM and interruption of a PDGFRA-mediated signalling axis involving *Mesp1*der PSC-dependent induction of EHT. AoE, adult aortic endothelium; HE, heart endothelium; NT, neural tube; and N, notochord. Data were derived from biologically independent samples, animals and experiments (**b**(ii), *n* = 3; **b**(iii),(iv), *n* = 5). Data represent the mean ± s.d. A random-effects Poisson regression was used to compare colony counts (**a**(ii)) and an ANOVA was used to compare donor chimerism (**a**(iii),(iv)); **\****P* < 0.01, **\*\*\****P* < 0.005. The precise *P* values are listed in the source data.

increased when CD45⁺ cells were co-transplanted with *Mesp1*–DsRed PSCs, possibly due to the latter providing additional in vivo maturation/engraftment support.

Tissues ventral to the AGM have an enhancing effect on HSC emergence, whereas tissues on the dorsal side decrease HSC production[53,54]. *Mesp1*der PSCs were indeed more abundant (7:1) in the mesenchyme around the ventral half of the dorsal aorta compared with the dorsal half (Extended Data Fig. 8c(i),(ii)). To study whether there were intrinsic differences in ventrally and dorsally distributed *Mesp1*der PSCs and their capacity to induce EHT, we

dissected E11.5 AGM from Mesp1–DsRed⁺ embryos and separated the dorsal and ventral halves[55] before FACS purification of *Mesp1*der PSCs (Extended Data Fig. 8c(i)). The CFU-F and replating capacity of the ventral fraction of *Mesp1*der PSCs were comparable to the whole AGM, whereas the dorsal fraction failed to form large colonies and had limited replating capacity (Extended Data Fig. 8c(ii),(iii)). Furthermore, whereas the dorsal fraction of *Mesp1*der PSCs failed to induce CFU-Cs from either E11.5 or adult heart endothelium, those from the ventral fraction induced robust numbers including LT-HSCs (Extended Data Fig. 8c(iv),(v)).

By contrast, co-aggregation of E11.5 or E13.5 *Wnt1*[der] PSCs with E11.5 (haemogenic) or E13.5 (non-haemogenic) aortic endothelium generated CFU-Cs at very low numbers or not at all, and without any long-term reconstitution when transplanted (Extended Data Fig. 9a(i)–(iv)). CFU-C generation was also comparable across different lots of fetal calf serum (FCS; Extended Data Fig. 9b).

***Mesp1*[der] PSCs induce haemogenic transcriptional programmes.** The cell signalling pathways that coordinate AGM haematopoiesis are not entirely clear, although the roles of WNT[20], NOTCH[19,56] and BMP[21,22] signalling in regulating critical transcription factors have been described. In addition, HSC production is known to be modulated by activation of nitric oxide synthesis mediated by shear stress and blood flow[57,58], components of prostaglandin E2 (ref. [59]) and inflammatory signals[60]. Catecholamine signalling in sub-aortic mesenchymal cells has also been shown to impact HSC emergence during embryonic development[61].

To explore the transcriptional changes in non-haemogenic endothelial cells during their transition to haemogenic endothelium and to identify features in E11.5 *Mesp1*[der] PSCs that distinguish them from E11.5 or E13.5 *Wnt1*[der] PSCs, we performed RNA sequencing on freshly isolated endothelial cells and *Mesp1*[der] PSCs as well as on endothelial cells extracted from co-aggregate cultures at 96 h (with *Mesp1*[der] PSCs or without *Mesp1*[der] PSC controls). Principal component analysis (PCA) of the transcriptomes of E11.5 and E13.5 endothelial cells extracted from *Mesp1*[der] PSC co-aggregates were more closely aligned with each other and with freshly isolated E11.5 endothelium than with freshly isolated E13.5 endothelium or control endothelial cells from 96-h co-aggregates (Fig. 5a). The PSC fractions were more closely aligned with each other than with endothelial cells, but consistent with their distinct germline derivations and temporal extractions, they were distributed across discrete sectors in transcriptomic space. Consistent with their acquisition of haemogenic properties, genes annotated by Ingenuity Pathway Analysis as associated with development of haematological systems were more differentially expressed in E13.5 endothelial cells extracted from *Mesp1*[der] PSC co-aggregates compared with freshly isolated non-haemogenic E13.5 AGM endothelial cells (Fig. 5b).

We evaluated the expression levels of WNT[20], BMP[21,22] and NOTCH[56,62] components in E11.5 and E13.5 endothelial cells (both freshly isolated and those extracted from *Mesp1*[der] PSC co-aggregates in the case of E13.5), E11.5 *Mesp1*[der] PSCs and E13.5 *Wnt1*[der] PSCs (Fig. 5c(i)–(iii)). E11.5 *Mesp1*[der] PSCs expressed higher levels of several secreted proteins that have previously been associated with EHT, and the *Wnt1*[der] PSCs expressed lower levels. Interestingly, there were higher expression levels in many of the corresponding receptors, signal transducers and target genes of these ligands in E13.5 endothelial cells extracted from co-aggregate cultures than in freshly isolated non-haemogenic E13.5 endothelial cells, such that they clustered with E11.5 fresh endothelium (Fig. 5c(i)–(iii)). These data suggested a role for WNT, BMP and NOTCH signalling in the induction of EHT in endothelial–*Mesp1*[der] PSC co-aggregates. To test this, we performed E10.5 and E11.5 *Mesp1*[der] PSC co-aggregation with adult heart and aortic endothelial cells in the presence of WNT, BMP or NOTCH inhibitors, followed by CFU-C and transplantation assays. With increasing concentrations of WNT (WIF-1 and draxin; Extended Data Fig. 10a(i)), NOTCH (LY450139 and MK0752; Extended Data Fig. 10a(ii)) and BMP (USAG1; Extended Data Fig. 10a(iii)) inhibitors, there was dose-dependent inhibition of CFU-Cs and a corresponding reduction in donor chimerism in hosts at 6 months following the transplantation of aggregates (Extended Data Fig. 10a). Combined inhibition of all three signalling pathways (WIF-1, MK0752 and USAG1) did not compromise the co-aggregate/re-aggregate cell viability (Extended Data Fig. 10b(i)) but ablated CFU-C and LT-HSC activity (Extended Data Fig. 10b(ii),(iii)).

To capture transcriptional changes in single endothelial cells as they gained haemogenic potential when co-aggregated with AGM stroma, we FAC-sorted adult cardiac endothelial cells from *UBC*–GFP[+] mice (GFP[+]CD31[+]VE-Cad[+]CD41[−]CD45[−]PDGFRA[−]PDGFRB[−]) and performed co-aggregate cultures with *Mesp1*[der] PSCs, FAC-sorted from E11.5 *Mesp1*–DsRed[+] AGMs (DsRed[+]PDGFRA[+]PDGFRB[−]CD31[−]VE-Cad[−]CD41[−]CD45[−]). Endothelial cells were FAC-sorted (GFP[+]CD31[+]VE-Cad[+]PDGFRA[−]) from co-aggregate cultures at 24, 48, 72 and 96 h, pooled with freshly sorted adult cardiac endothelium, and single-cell next generation sequencing libraries were prepared using a 10X Genomics Chromium platform. Uniform manifold approximation and projection (UMAP) representation of endothelial cell transcriptomes from triplicate pools and Leiden clustering[63] detected several cell populations (Fig. 6a). SOX17 is expressed in endothelial cells lining arteries, but not veins, and is a key regulator of haemogenic endothelium[64]. RUNX1 represses the pre-existing arterial programme in haemogenic endothelium and is required for haematopoietic cell transition from haemogenic endothelium[27,65]. These transcription factors show a reciprocal expression pattern with most cells in clusters 5 and 6 showing robust *Runx1* expression, and those in clusters 0, 1 and 2 showing high *Sox17* expression (Fig. 6b). Endothelial cells were FAC-sorted based on their CD31 (also known as PECAM) and VE-Cad (also known as CDH5) surface protein expression. As cells gained *Runx1* expression, there was a concomitant gain of haematopoietic gene expression (for example, *Gfi1b*, *Tal1*, *Myb*, *Spi1*, *Gata1*, *Ptprc* (CD45) and *Itga2b* (CD41)) and loss of endothelial gene expression (for example, *Pecam1*, *Cdh5*, *Sox17*, *Kdr*, *Notch1* and *Egfl7*). The clusters with *Runx1*-expressing cells seem to have distinct identities with the cells in cluster 5 showing high levels of *Ptprc*, *Spi1* and *Myb* expression, and those in cluster 6 showing high levels of *Itga2b*, accompanied by *Gata1*, *Tal1* and *Gfi1b*, which is suggestive of distinct haematopoietic lineage differentiation potential before complete downregulation of the endothelial signature (Fig. 6c). This is in keeping with reports from single-cell transcriptional analyses of human pluripotent stem cell-derived CD34[+] cells[66] in E11 AGM[67]. Pseudotime[68] (https://arxiv.org/abs/1802.03426) analysis starting from cluster 0 (that is, strongest endothelial identity; Fig. 6d) showed distinct progression towards cluster 6 (haemogenic), with alternate trajectories towards clusters 3 and 4 (non-haematopoietic; Fig. 6b) and 12–21% of sorted endothelial cells expressing *Runx1* (>1 or >0 counts per ten thousand) transcripts at 96 h (Fig. 6e).

**PDGFRA-mediated cell signalling is important for EHT.** The role of *Mesp1*[der] PSCs in the induction of EHT in co-aggregate cultures prompted us to explore a role for PDGF signalling in mediating endothelial–stromal cell crosstalk in the AGM. The PDGF signalling system consists of four ligands: platelet-derived growth factor (PDGF)-A, -B,-C and -D[69]. All four ligands assemble intracellularly to form disulfide-linked homodimers. PDGF-AA binds αα-receptor homodimers[69]. The expression levels of PDGF-A were highest in E11.5 fresh endothelium and E13.5 endothelium harvested from *Mesp1*[der] PSC co-aggregates (Fig. 7a(i)). The expression levels of PDGFRA (the receptor for PDGF-A) were highest in E11.5 *Mesp1*[der] PSCs (Fig. 7a(i)). Confocal microscopy of the E11.5 AGM in *Pdgfra*–nGFP transgenic mice showed abundant and dispersed PDGF-A protein in the endothelium and mesenchyme (Fig. 7a(ii)), but the PDGF-A protein levels were low and restricted to the endothelium at E13.5 (Fig. 7a(iii)). Importantly, E11.5 endothelial cells do not express PDGFRA, but sub-aortic stromal cells do (Fig. 7a(i)). To evaluate whether PDGFRA-mediated signalling was required for EHT mediated by *Mesp1*[der] PSCs at E11.5, we co-aggregated E11.5 *Mesp1*[der] PSCs with E11.5 aortic endothelium in the presence of a specific PDGFRA inhibitor (APA5; Fig. 7b(i)). There was dose-dependent reduction in CFU-Cs (Fig. 7b(ii)) and LT-HSCs (Fig. 7b(iii)). Treatment with a PDGFRB inhibitor (APB5) had no effect on CFU-Cs (Fig. 7b(ii)).

To further investigate the impact of PDGF-A–PDGFRA signalling on EHT, we harvested E10.0 embryos and cultured them ex vivo[70,71] in the presence or absence of APA5 (Fig. 7c(i)). Confocal immunofluorescence imaging after 36 h showed that inhibition of PDGFRA signalling at E10 did not severely impact the architecture of the dorsal aorta, but blood clusters along its ventrolateral surface were reduced (Fig. 7c(ii)). Flow cytometry showed that SCA1⁺CKIT⁺ and SCA1⁺CKIT⁻ cells were reduced by 60% and 22%, respectively (Fig. 7c(iii)). Both CFU-Cs (Fig. 7c(iv)) and CFU-Fs (Fig. 7c(v)) were significantly reduced in number. Conversely, PDGF-AA supplementation of media lacking the cytokine cocktail (IL-3, SCF and Flt3L) that is used in co-aggregate cultures rescued CFU-C and LT-HSC production from endothelial cells when co-aggregated with *Mesp1*der PSCs but had no impact on *Wnt1*der PSCs (Fig. 8a(i)–(iv)).

As summarized in Fig. 8b, the aortic endothelium is derived in part from E9.5 *Mesp1*der PSCs, which at E11.5 are distributed in the aortic sub-endothelium and express WNT, NOTCH and BMP haemogenic ligands. The E11.5 endothelium has complementary receptors for these ligands and produces soluble PDGF-AA. Interruption of PDGF-AA–PDGFRA signalling impedes *Mesp1*der PSC function and EHT. Coincident with the cessation of EHT at E13.5, sub-endothelial PDGFRA⁺ *Mesp1*der PSCs are replaced by *Wnt1*der PSCs. *Wnt1*der PSCs do not express haemogenic ligands. E13.5 endothelium does not express complementary receptors for these ligands or soluble PDGF-AA. However, EHT can be re-established in non-haemogenic endothelium by co-aggregating these cells with E10.5 or E11.5 *Mesp1*der PSCs to generate LT-HSCs.

## Discussion

PDGFRA⁺ is broadly expressed in primitive para-axial mesoderm, somites and mesenchyme[72], and has previously been shown to contribute to haemogenic endothelium and haematopoietic progenitor cells, but only during an E7.5–E8.5 developmental window[36]. However, here we show that PDGFRA⁺ cells continue to contribute to the AGM endothelium and transplantable haematopoietic stem cells at E9.5 (Fig. 2c,d). These variations could be due to differences in *cre*-recombinase efficiency or variations in fitness of reporter PDGFRA +/+ mesodermal cells (our transgenic model) and reporter PDGFRA KI/+ cells used in the previous model[36]. A previous study showed no difference in LSK haematopoietic progenitor cells in the E12.5 fetal livers of heterozygous and homozygous Pdgfra-mer-Cre-mer mice[36]. However, the numbers of transplantable haematopoietic stem cells in the E12 are few (50–60)[18] and E12 fetal liver haematopoietic progenitor cells are not the sole product of PDGFRA⁺ mesoderm, as yolk-sac erythromyeloid progenitors also contribute[73], and it is conceivable that in the absence of the former, there were compensatory increases in yolk-sac contributions to E12.5 fetal liver LSKs.

It has been proposed that haematopoietic production in the AGM depends on a limited, non-renewable, pool of haemogenic endothelial cells that are replaced by somite-derived endothelial cells and that this may account for the cessation of AGM haematopoiesis at E13.5 (refs. [74,75]). Whether replacement of *Mesp1*der PSCs (which promote EHT) by *Wnt1*der PSCs (which do not promote EHT) contributes to cessation of AGM haematopoiesis is an intriguing possibility. What accounts for the disappearance of *Mesp1*der PSCs from the AGM at E13.5 is unknown. It is possible that these cells went through terminal differentiation as a new wave of PSCs infiltrated the AGM. Alternatively, these cells could have undergone apoptosis or migrated elsewhere in the embryo. There are precedents to distinct subpopulations of MSCs contributing to specific tissues during development[76].

It is probable that *Mesp1*der PSCs are themselves a heterogeneous population of cells evident by intrinsic functional differences between cells in the dorsal or ventral halves of the AGM (Fig. 4b). Single-cell transcriptomics of *Mesp1*der PSCs may help resolve this heterogeneity. A study using *Runx1b* and *Gfi1/1b* transgenic reporter mice to isolate endothelial and mesenchymal cells from E10.5/E11.5 AGM for single-cell transcriptomics is a step in this direction[77]. Identification of equivalent PSCs in the human AGM or factors that could replace the use of embryonic PSCs altogether are steps required to generate autologous HSCs from human endothelial cells.

## Online content

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

## Methods

**Mice.** The mice were housed and bred in the Biological Resource Centre at the Lowy Cancer Research Centre, UNSW, a specific pathogen-free, physical containment level 2 facility with a semi-natural light cycle of 12:12 h light:dark and regulated air quality, ventilation (15 air changes hourly), humidity (40–70%) and temperature (22 ± 1 °C). The developmental stage of embryos was determined according to Theiler's criteria. All animal experiments were approved by the UNSW Sydney Animal Care and Ethics Committee. Strains are listed in Supplementary Table 1. Our experiments did not require a randomization protocol or need to assign animals or samples to specific treatment groups.

**AGM PSC isolation and ex vivo expansion.** E10.5–E13.5 AGM regions were dissected as previously described[18]. The dissected AGMs were transferred into collagenase type II (263 U ml⁻¹) and placed on a shaker at 37 °C for 15 min. The supernatant was passed through a 40-µm filter into a fresh tube and inactivated with 100% FCS. The cells were washed twice in PBS containing 2% FCS and plated in αMEM medium with 20% FCS and penicillin–streptomycin–glutamine[37], and cultured in an incubator at 37 °C and 5% $CO_2$ for 72 h. At the end of 72 h, the cells were washed in PBS to remove non-adherent cells and cultures were continued in fresh medium. The cells were passaged on reaching 80% confluency. After passaging, the cells were placed back in tissue culture flasks with αMEM + 20% FCS + penicillin–streptomycin–glutamine for bulk passaging.

**Long-term growth and serial clonogenicity of AGM PSCs.** AGM PSCs were expanded in bulk culture after plating 5,000–10,000 cells per T25 flask. The resulting cells were split every 8–12 d. Cumulative cell numbers were calculated and plotted (log₁₀ scale). To investigate single-cell serial clonogenicity, individual AGM PSC colonies were isolated using cloning cylinders ('O' rings) in the presence of Triple-E. The secondary, tertiary and quaternary colony formation of micro ($n = 54$), small ($n = 30$) and large ($n = 27$) AGM PSC colonies was evaluated by plating single cells from individual micro, small and large colonies (that were plated from secondary to tertiary and tertiary to quaternary colonies) into 48- or 96-well plates using a mouth pipette. The cells were cultured in αMEM medium with 20% FCS and penicillin–streptomycin–glutamine[37] at 37 °C and 5% $CO_2$ for 72 h. At the end of 72 h, the cells were washed in PBS to remove non-adherent cells and cultures were continued in fresh medium. Cell culture was ended at the end of day 12, and colonies were stained with crystal violet and counted as described[38]. Random-effects Poisson regression was used to compare counts of cells by colony size, alpha level and GFP levels with a random intercept for counts within the same dish. A linear mixed model was used to compare the growth curves of the two groups with random intercepts and slopes for each replicate.

**AGM pericyte culture.** E11.5 AGMs were dissected, and dissociated cells were washed and cultured as described[78] in an incubator at 37 °C and 5% $CO_2$ for 72 h. At the end of 72 h, the cells were washed in PBS to remove non-adherent cells and cultures were continued in fresh medium.

**Ex vivo post-implant embryo culture.** Harvested E10.0 embryos were cultured ex vivo for 36 h (ref. [70]) in the presence or absence of 40 µg ml⁻¹ APA5 (PDGFRA inhibitor)[79]. At the end of ex vivo culture the embryos were either fixed in 4% paraformaldehyde (PFA) for 15–20 min at room temperature for confocal microscopy imaging or AGM regions were dissected and dissociated as previously described[18] for CFU-C, CFU-F and long-term growth experiments. Fixed embryos were washed and embedded in optimal cutting temperature (OCT) compound by flash-freezing on dry ice, cut into 30-µm sections and then permeabilized with 0.03% Tween-20 in PBS (vol/vol) for 15 min at room temperature. The cells were washed once with PBS and then blocked with 10% donkey serum (vol/vol) in PBS for 1 h. The sections were subsequently incubated overnight at 4 °C with the primary antibodies in PBS containing 2% BSA (wt/vol), stained accordingly with secondary antibodies in 2% BSA and incubated for 1 h at 4 °C. Slides were mounted with ProLong Gold mounting medium (Invitrogen). Random-effects Poisson regression was used to compare counts of cells by colony size with a random intercept for counts within the same dish.

**Co-culture of OP9 and PDGFRA⁺ cells.** OP9 cells were cultured and expanded before co-culture following a previously established method[80]. E9.5 embryos were harvested from *UBC*–GFP⁺ pregnant mothers and PDGFRA⁺ cells were FAC-sorted. OP9 cells were expanded to form confluent layers in six-well plates. Single-cell suspensions of FAC-sorted GFP⁺PDGFRA⁺ cells from the E9.5 embryonic trunk were plated on top of the OP9 cells containing endothelium growth medium and cultured for 96 h (refs. [80,81]). The antibodies used are listed in Supplementary Table 2.

**In vivo vasculogenesis assay.** E11.5 AGMs were dissected, and dissociated cells were washed twice and stained with the appropriate antibodies. After washing with FACS buffer, the cells were FAC-sorted to obtain the desired cell population. The cells (250,000) were mixed with 100 µl Matrigel (BD Biosciences)[82] and injected subcutaneously into the nuchal and groin regions of C57BL/6 mice. At the end of 4 weeks, Matrigel plugs were harvested and fixed with 4% PFA (ProSciTech) in PBS

(wt/vol) for 15–20 min at room temperature. These tissues were embedded in OCT compound by flash-freezing on dry ice, cut into 30-µm sections or whole mounted for confocal imaging.

**Confocal microscopy.** Tissues were embedded in OCT compound by flash-freezing on dry ice, cut into 30-µm sections and permeabilized with 0.03% Tween-20 in PBS (vol/vol) for 15 min at room temperature. The cells were washed once with PBS and then blocked with 10% donkey serum (vol/vol) in PBS for 1 h. The sections were subsequently incubated overnight at 4 °C with primary antibodies in PBS containing 2% BSA (wt/vol), stained accordingly with secondary antibodies in 2% BSA and incubated for 1 h at 4 °C. Slides were mounted with ProLong Gold mounting medium (Invitrogen). The slides were analysed using either an L780 LSM Zeiss or a Leica SP8 DLS microscope. Three-dimensional rendering was performed using the Imaris software to provide improved spatial information in the *z* direction. Here we created three-dimensional isosurface renderings from confocal *z*-stacks of subcutaneously transplanted AGM PSCs in Matrigel for 3 weeks.

**PDGFRA⁺ cell depletion experiments.** *Pdgfra–cre*ᴱᴿᵀ² females were time-mated with *iDTR* males to generate *Pdgfra–cre*ᴱᴿᵀ²; *iDTR* embryos. When female mice were identified as pregnant, a single injection of tamoxifen (143 mg kg⁻¹) was administered at E9.5 and a single injection of diphtheria toxin (4 µg kg⁻¹) was administered intraperitoneally after 24 h. Embryos were harvested at E11.5 for analysis.

**In vitro lineage differentiation.** In vitro lineage differentiation was performed as described previously[83].

*Osteogenic differentiation.* Osteogenic differentiation was promoted by culturing cells for 21 d in either a six-well plate or a four-chamber slide containing Dulbecco's minimum essential medium (DMEM)-low glucose, 10% FCS, 100 µg ml⁻¹ penicillin plus 250 ng ml⁻¹ streptomycin, 200 mM L-glutamine and 0.1 µM dexamethasone, 10 mM β-glycerophosphate and 200 µM L-ascorbic acid 2-phosphate. The cells were stained with alizarin red.

*Chondrogenic differentiation.* Cells (1–2.5 × 10⁵) were plated in either a six-well plate or a four-well chamber slide containing serum-free DMEM-high glucose (DMEM-HG) medium, 100 µg ml⁻¹ penicillin, 250 ng ml⁻¹ streptomycin, 200 mM L-glutamine, 50 µg ml⁻¹ insulin–transferrin–selenious acid mix, 2 mM L-ascorbic acid 2-phosphate, 1 mM sodium pyruvate, 0.1 µM dexamethasone and 10 ng ml⁻¹ transforming growth factor β3 (LSBio). The medium was changed every 4 d for 28 d. Differentiated cells were stained for sulfated proteoglycans with 1% alcian blue.

*Adipogenic differentiation.* Cells were cultured for 7–10 d in DMEM-HG medium containing 10% FCS, 100 µg ml⁻¹ penicillin, 250 ng ml⁻¹ streptomycin, 200 mM L-glutamine plus 0.5 µM 1-methyl-3-isobutyl methylxanthine, 1 µM dexamethasone, 10 µg insulin and 200 µM indomethacin. The cells were fixed and stained with oil red O.

*Smooth-muscle differentiation.* Smooth-muscle differentiation was promoted by culturing the cells in the presence of 50 ng ml⁻¹ PDGF-BB (R&D Systems) made up with 5% FCS in DMEM-HG, 100 µg ml⁻¹ penicillin, 250 ng ml⁻¹ streptomycin and 200 mM L-glutamine. The cells were induced for 14 d; the medium was changed every 3–4 d.

*Endothelial differentiation.* Endothelial-cell differentiation was promoted by culturing cells in 5% FCS in Iscove's modified Dulbecco's medium containing 10 ng ml⁻¹ bFGF, 10 ng ml⁻¹ vascular endothelial growth factor, 100 µg ml⁻¹ penicillin, 250 ng ml⁻¹ streptomycin and 200 mM L-glutamine. For low-density lipoprotein uptake, acetylated apoprotein-low-density lipoprotein (AcLDL-Alexa Fluor 488 molecular probes) at a final concentration of 10 µg ml⁻¹ was added to the endothelial differentiation assays at the end of day 14. The cells were then cultured for a further 24 h. At the end of day 15, the cells were fixed and uptake was assessed by fluorescence yield. For Matrigel assays, AGM PSCs were plated on chamber slides containing Matrigel and cultured for 7 d. At the end of day 7, the tubes were fixed and stained.

*Cardiomyocyte differentiation.* To promote cardiomyocyte differentiation, cells were cultured for 4–5 d in normal MSC medium in 2% Matrigel-coated chamber slides or glass-bottomed Petri dishes. The cells were then differentiated in cardiomyogenic differentiation medium consisting of DMEM-low glucose: Medium 199 (4:1), 1.0 mg ml⁻¹ bovine insulin, 0.55 mg ml⁻¹ human transferrin, 0.5 µg ml⁻¹ sodium selenite, 50 mg ml⁻¹ BSA plus 0.47 µg ml⁻¹ linoleic acid, 1 × 10⁻⁴ M ascorbate phosphate, 1 × 10⁻⁹ M dexamethasone, 100 µg ml⁻¹ penicillin, 250 ng ml⁻¹ streptomycin, 200 mM L-glutamine and 10% FCS with 1 ng ml⁻¹ recombinant human neuregulin 1β2 for 14–21 d. The medium was changed every 3 d. The cells were stained for cardiac α-sarcomeric actinin. Images of beating cardiomyocytes were acquired on a Nikon Ti-E microscope with a

×20 phase objective (0.45 numerical aperture); 1,000 frames were acquired continuously with a 52 ms frame rate. Twelve-bit images were acquired with a 1,280 × 1,024 pixel array.

*Neuronal differentiation.* When cells were at 80% confluency, the culture medium was switched to DMEM-HG containing 100 μg ml⁻¹ penicillin, 250 ng ml⁻¹ streptomycin, 200 mM L-glutamine and 1 mM β-mercaptoethanol. The medium was changed every 3–4 d; the cells were cultured for 8–10 d.

*Hepatocyte differentiation.* When cells were at 80% confluency, the culture medium was switched to serum-free DMEM-HG containing 100 μg ml⁻¹ penicillin, 250 ng ml⁻¹ streptomycin, 200 mM L-glutamine, 20 ng ml⁻¹ EGF and 10 ng ml⁻¹ bFGF to inhibit cell proliferation for 2 d. After conditioning the cells, differentiation medium—consisting of DMEM-HG supplemented with 20 ng ml⁻¹ HGF and 10 ng ml⁻¹ bFGF—was added and the cells were incubated for 7 d. The cells were then cultured in DMEM-HG supplemented with 20 ng ml⁻¹ oncostatin M, 1 μmol l⁻¹ dexamethasone, 10 μl ml⁻¹ insulin–transferrin–selenious premix, and 100 μg ml⁻¹ penicillin and 250 ng ml⁻¹ streptomycin for 14 d.

**Immunohistochemistry.** Cells were washed with PBS for 10 min, fixed with 4% PFA in PBS (wt/vol) for 15–20 min and then permeabilized with 0.03% Tween-20 in PBS (vol/vol) for 15 min at room temperature. The cells were washed once with PBS and then blocked with 10% donkey serum (vol/vol) in PBS for 1 h. The cells were subsequently incubated overnight at 4 °C with primary antibodies in PBS containing 2% BSA (wt/vol), stained accordingly with secondary antibodies in 2% BSA and incubated for 1 h at 4 °C. Slides were mounted with ProLong Gold mounting medium. The antibodies used are listed in Supplementary Table 2.

**Long-bone immunohistochemistry.** Long bones (femur and tibia) were fixed in 4% PFA for 24 h and decalcified in 14% EDTA solution for a further 24–36 h. These bones were washed and embedded in OCT compound by flash-freezing on dry ice, cut into 10-μm sections and then permeabilized with 0.03% Tween-20 in PBS (vol/vol) for 15 min at room temperature. The cells were washed once with PBS and blocked with 10% donkey serum (vol/vol) in PBS for 1 h. The sections were subsequently incubated overnight at 4 °C with primary antibodies in PBS containing 2% BSA (wt/vol), stained accordingly with secondary antibodies in 2% BSA and incubated for 1 h at 4 °C. The antibodies used are listed in Supplementary Table 2.

**CFU-C assay.** Colony development was performed using M3434 medium (Stem Cell Technologies) according to the manufacturer's protocol. Colonies were scored after 7 d. Random-effects Poisson regression was used to compare counts of cells by different types of colonies with a random intercept for counts within the same dish.

**Long-term repopulation assay.** Donor tissues from *Pdgfra–cre*^ERT2^; *R26*–eYFP embryos and neonatal long bones—or *Pdgfra*–nGFP/tdTomato (ROSA26) AGM tissues or *UBC*–GFP⁺, *UBC*–GFP⁺*Mesp1*–DsRed⁺ and *UBC*–GFP⁺*Wnt1*–DsRed⁺ re-aggregates and co-aggregates—were isolated and transplanted via the tail vein into irradiated C57BL/6J adult recipients. The numbers of transplanted cells are reported throughout as the number present in one cultured AGM (for example, one dose is equal to 100% of the cells in one cultured AGM). Donor cells were co-injected with 20,000 wild-type bone-marrow cells. The number of HSCs was estimated using a limited dilution assay as described previously[84]. To assess donor chimerism following transplantation of sorted co-aggregate subpopulations, 20 co-aggregates were FAC-sorted for GFP⁺CD45⁺, GFP⁺CD45⁻ and DsRed⁺ PSC populations as mentioned in the experimental design. The sorted cells were each resuspended in 200 μl of 2% FCS. For tail-vein injections, 10 μl of each relevant cell suspension (approximately one co-aggregate equivalent) was used after top-up to 100 μl with 2% FCS. The antibodies used are listed in Supplementary Table 2.

**Phenotypic identification of HSCs.** Cell suspensions were stained using the appropriate combinations of monoclonal antibodies. Donor contribution was assessed by flow cytometry using endogenous GFP and DsRed. Mice demonstrating ≥4% donor-derived chimerism (contribution to both myeloid and lymphoid lineages) after a minimum of 4 months were considered to have been reconstituted. The antibodies used are listed in Supplementary Table 2.

**Flow cytometry and cell sorting.** Mononuclear staining was analysed on a LSRFortessa system (BD Biosciences). Cell sorting was performed on a BD Influx system (BD Biosciences). Antibodies are listed in Supplementary Table 2. The FACS data were analysed using FlowJo software.

**Co-aggregation assays.** Co-aggregates were made by reconstituting FAC-sorted endothelial cells from one AGM equivalent or 25,000 adult endothelial cells with 150,000 AGM PSCs. Dissociated cells were resuspended in 10 μl Iscove's modified Dulbecco's medium containing 20% fetal calf serum, 4 mM L-glutamine, 50 U ml⁻¹ penicillin–streptomycin, 0.1 mM β-mercaptoethanol, 100 ng ml⁻¹ IL-3, 100 ng ml⁻¹ SCF and 100 ng ml⁻¹ Flt3L. Co-aggregates were made by centrifugation in a yellow tip occluded by parafilm at 300*g* for 5 min and cultured on top of a 0.65-μm

Durapore filter (Millipore, cat. no. DVPP02500) at the gas–liquid interface as described[84]. Tissues were maintained in 5% CO₂ at 37 °C in a humidified incubator. The antibodies used are listed in Supplementary Table 2. The growth factors are listed in Supplementary Table 3.

**Inhibitor assays.** NOTCH, BMP or WNT inhibitors were added to the re-aggregate/co-aggregate culture media and to the cell mixture (while generating aggregates), and cultured for 96 h (from 0 h and kept for 96 h). At the end of 96 h, the co-aggregates and re-aggregates were dissociated into single-cell suspensions to perform CFU-C and transplant assays. Inhibitors were reconstituted according to the manufacturer's instructions. Random-effects Poisson regression was used to compare the effect of inhibitors on the counts of CFU-C colonies and chimerism with a random intercept for counts within the same dish. Inhibitors and cytokines are listed in Supplementary Table 4.

**Adult endothelial cell isolation.** Mononuclear cells were isolated from dissected hearts, lung, aorta and inferior vena cava of 12–16-week-old female mice by mincing the tissues before digestion in 263 U ml⁻¹ collagenase type II (heart, aorta and inferior vena cava) and 263 U ml⁻¹ collagenase type I (lung) in PBS at 37 °C for 30 min, with mechanical trituration at 10-min intervals. Myocyte debris were removed with a 40-μm filter. Dead cells were removed using a MACS dead cell removal kit before incubation with fluorophore-conjugated antibodies (Supplementary Table 2) to FAC-sort endothelial cells.

**RNA sequencing, PCA and hierarchical clustering.** RNA libraries were prepared using an Illumina TruSeq RNA library preparation kit v2 and 10 nM complementary DNA was used as input for high-throughput sequencing on an Illumina HiSeqX (150-bp paired-end reads). Raw sequencing reads were filtered for adaptors: 12–16 reads in which more than 10% of bases were unknown and reads in which more than 50% of bases were low quality (base quality < 20). The resultant high-quality reads were aligned to the mouse genome (mm10) using the software STAR (v2.5.0b)[85] with standard parameters. We mapped an average of 50,727,988 reads per sample and the average alignment rate was 94.7%. Gene expression levels were quantified using HTSeq (v0.9)[86]. The expression levels were trimmed mean of M values-normalized using the software package EdgeR (v3.5) in the R statistical analysis software (v3.3.3)[87,88]. Genome-wide expression profiles were analysed using PCA analysis[89,90]. The PCA algorithm is a dimension-reduction technique that identifies directions (called principal components) along which gene expression measures are most variant. The principal components are linear combinations of the original gene expression measures and allow visualization of genome-wide expression profiles in two or more dimensions. Hierarchical clustering with average linkage and Euclidean distance was performed using the Partek Genomics Suite (v6.6). software and are summarized in Supplementary Table 5.

**Single-cell RNA sequencing and data analysis.** Endothelial-cell single-cell RNA sequencing of freshly isolated adult cardiac endothelium (0 h) and FAC-sorted adult cardiac endothelium following co-aggregation with *MesP1*^der^ PSCs for 24, 48, 72 and 96 h was performed in biological triplicates. The 10X Genomics Chromium Single Cell 3′ platform (v3 chemistry) was used for sequencing and Illumina HiSeq 4000 Cell Ranger (v3.1.0) was used to process raw datasets including quality control, the extraction of gene expression matrices and the aggregation of gene expression matrices from different sequencing runs with batch-effect removal. High-quality transcriptome expression (that is, singlets; ≥5,000 unique molecular identifiers) profiles were extracted from 9,562 single cells. To analyse the filtered gene expression counts, we wrote custom Python 3.9 scripts (available at: https://github.com/iosonofabio/scpaper_Vashe). Briefly, the raw-count matrix was converted to gene names, summing over all Ensembl IDs with the same gene name, and normalized to counts per ten thousand unique molecular identifiers (CPTT). We then used scanpy (v1.8.2; https://scanpy.readthedocs.io) to log-transform the counts, calculate overdispersed features, perform PCA and UMAP embedding (https://arxiv.org/abs/1802.03426), compute a similarity graph with ten neighbours and cluster with the Leiden algorithm[63]. We next used singlet (https://singlet.readthedocs.io) to make dot plots with a threshold of 0.5 CPTT and UMAP projections by cluster, and log-transformed gene expression and cumulative distributions for the expression of selected genes within each cluster. Clusters were numbered from the highest *Cdh5* expressor to the highest *Runx1* expressor. Pseudotime analysis was performed using scanpy (https://scanpy.readthedocs.io/en/stable/api/scanpy.tl.dpt.html). Both branched and unbranched pseudotime were tested and yielded similar results (unbranched pseudotime was used for Fig. 6d).

**Statistical analysis.** All data are presented as the mean ± s.d. (as indicated in the figure legends). The *P* values were all for two-sided tests. Data presented in the figures reflect multiple independent experiments performed on different days using different embryos or mice. Unless otherwise mentioned, the data presented in the figure panels are based on at least three independent experiments. Random-effects Poisson regression was used to compare counts of cells by colonies with a random intercept for counts within the same dish. ANOVA was used to compare donor chimerism in transplants. A linear mixed model was

used to compare the growth curves of the two groups with random intercepts and slopes for each replicate. $P > 0.05$ was considered not significant; $*P < 0.05$; $**P < 0.01$; $***P < 0.005$. In all the figures, $n$ refers to the number of mice, embryos or replicates. All statistical analyses were performed using SAS v9.4 (SAS Institute; 2016). SAS statistical analyses and codes are available as Supplementary Information.

No animals were excluded from analyses. Sample sizes were selected based on previous experiments. Investigators were blinded to group allocation during data collection and analysis. Unless otherwise indicated, results are from three independent biological replicates to guarantee reproducibility of findings.

**Reporting summary.** Further information on research design is available in the Nature Research Reporting Summary linked to this article.

## Data availability

Bulk and single-cell RNA-sequencing data were aligned to mm10 (https://www.ncbi.nlm.nih.gov/assembly/GCF_000001635.20/) and have been deposited in Gene Expression Omnibus under GSE163757 and GSE114464, respectively. These data are publicly available. All other data supporting the findings of this study are available from the corresponding authors on reasonable request. Source data are provided with this paper.

## Code availability

Custom scripts for single-cell data analysis are available at: https://github.com/iosonofabio/scpaper_Vashe. Statistical analyses and codes are available as an accompanying file (Supplementary Information).

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

## Acknowledgements

We thank W. D. Richardson (University College London) and K. M. Young (University of Tasmania) for providing *Pdgfra–cre*^ERT2 reporter mice as well as A. Waisman (Johannes Gutenberg University Mainz) for providing iDTR mice. We thank R. P. Harvey (Victor Chang Cardiac Research Institute, Sydney) for transferring *Pdgfra–nGFP*, *Wnt1–cre* and *Mesp1–cre* mice. We thank the staff at the UNSW Sydney Biological Resource Centre for maintaining mouse lines and the Mark Wainwright Analytical Centre for assistance with flow cytometry, confocal microscopy and image processing. This work was funded by the National Health and Medical Research Council of Australia (J.E.P, grant nos 510100, 568668, 630497 and 1102589; and V.C., grant no. 1061593), Australian Research Council (J.E.P, grant no. DP0984701), a Faulty of Medicine, UNSW grant (V.C.) and a Mark Wainwright Analytical Centre, UNSW grant (V.C.). B.G. was supported by Wellcome (grant no. 206328/Z/17/Z), MRC (grant no. MR/S036113/1), Blood Cancer UK (grant no. 18002) and core funding by Wellcome to the Cambridge Stem Cell Institute.

## Author contributions

V.C. and J.E.P. designed the study with valuable discussions from S.T. throughout. V.C. performed most of the experiments and analysed the data for the manuscript. P.R., S.J., Y.C.K., K.K., Q.Q., R.A.O., A.U., B.L., C.B. and C.P. performed experiments. F.Z., Y.H., D.C., D.R.C. and D.B. performed bioinformatics analysis on the sequencing data. J.O. reviewed and performed all statistical analyses. R.B., S.M.-F., G.E., W.W. and S.T. contributed valuable tools and protocols. V.C., F.Z., B.G., S.T. and J.E.P. interpreted results. V.C. and J.E.P. wrote the manuscript. All authors have read and approved the manuscript.

## Competing interests

The authors declare no competing interests.

## Additional information

**Extended data** is available for this paper at https://doi.org/10.1038/s41556-022-00955-3.

**Correspondence and requests for materials** should be addressed to Vashe Chandrakanthan or John E. Pimanda.

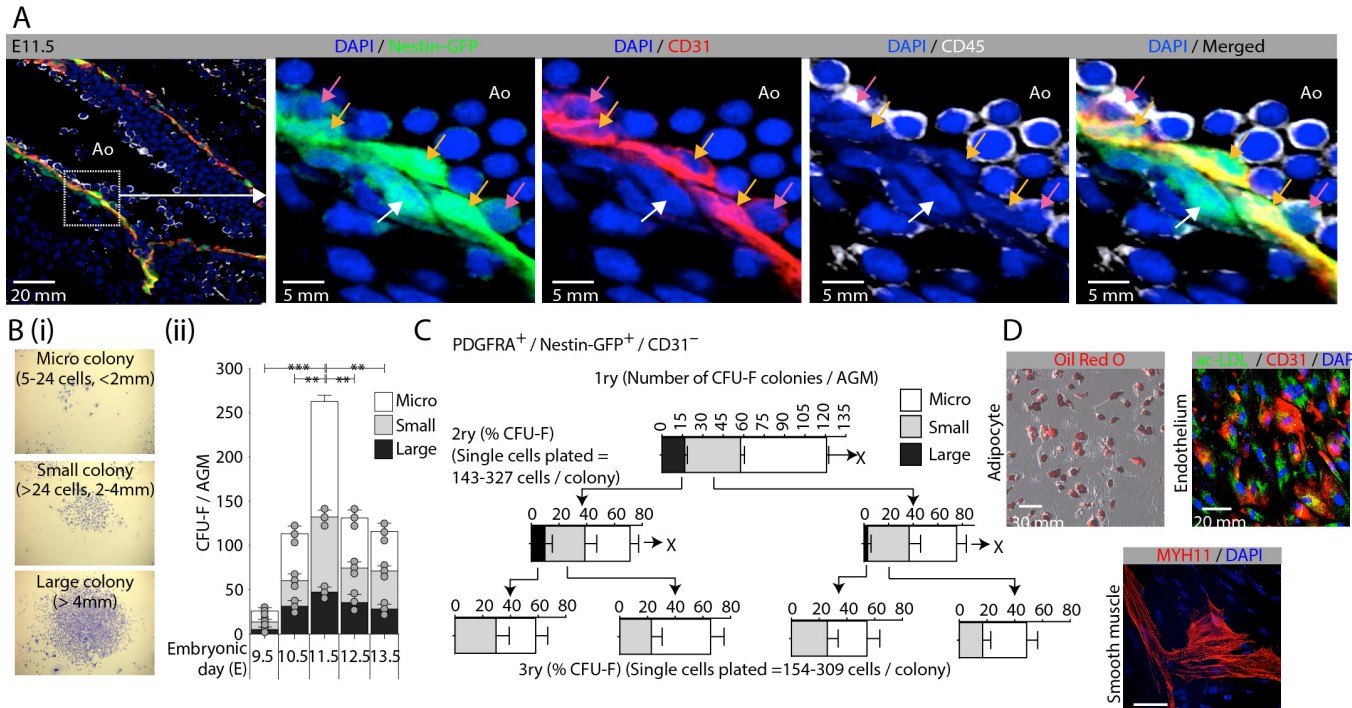

**Extended Data Fig. 1 | Colony forming and differentiation potential of E11.5 AGM cells expressing PDGFRA and *Nestin*-GFP.** (A) Confocal images of the ventral surface of an E11.5 dorsal aorta showing *Nestin*-GFP (green) expression in CD31+ endothelial (orange arrow), CD31- perivascular mesenchymal (white arrow) and budding CD45+ blood cells (pink arrow). (B) (i) CFU-F classification based on size and cell number. (ii) CFU-Fs in E9.5, 10.5, 11.5, 12.5 and E13.5 AGM (n = 3 /timepoint). (C) Single cell clonal analysis of CD31-; PDGFRA+; *Nestin*-GFP+ CFU-Fs. (D) In vitro differentiation of CD31-; PDGFRA+; *Nestin*-GFP+ cells (n = 3). Ao: aortic lumen; CFU-F: colony-forming unit–fibroblast; colony sizes: micro colonies (<2 mm, 2–24 cells), small colonies (2–4 mm, >25 cells) and large colonies (>4 mm, >100 cells). CFU-F colony numbers in (F) were representative of n = 3 (1ry plating) and n = 7-12 (2–4ry plating). ** p < 0.01, *** p < 0.005; random effects Poisson regression was used to compare colony counts in B (ii). Data were derived from biologically independent samples and experiments. Data represent mean ± S.D. The precise p values are listed in the source data.

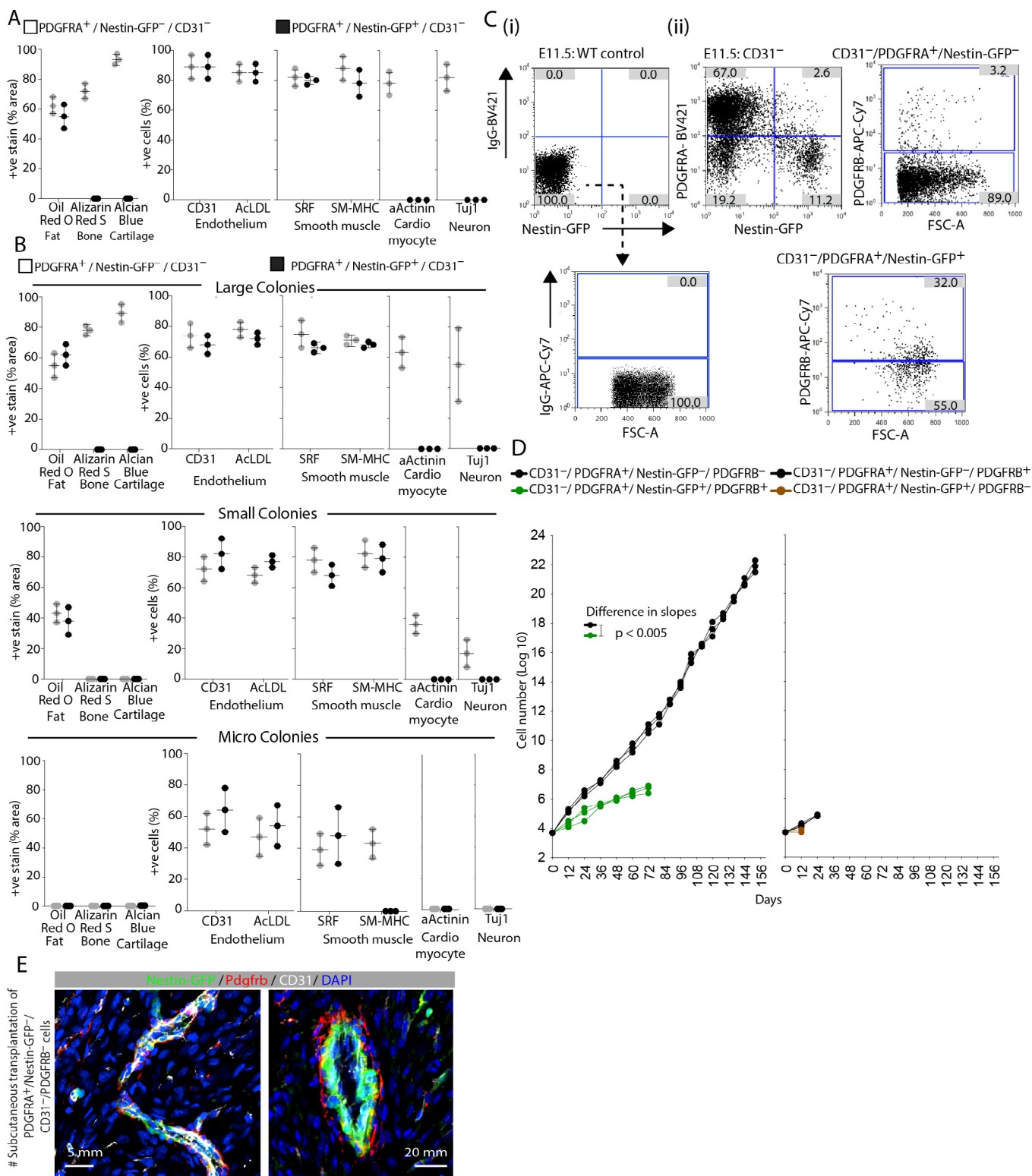

**Extended Data Fig. 2 | See next page for caption.**

**Extended Data Fig. 2 | Comparative differentiation and long-term growth potential of E11.5 AGM cells based on PDGFRA, PDGFRB and *Nestin*-GFP expression.** (A) Dot plots showing % (percentage) conversion to marker-positive cells following differentiation induction of bulk cultured CD31⁻; PDGFRA⁺; *Nestin*-GFP⁻ and CD31⁻; PDGFRA⁺; *Nestin*-GFP⁺ CFU-F cells (n = 3). (B) Dot plots showing % (percentage) conversion to marker-positive cells following differentiation induction of CD31⁻; PDGFRA⁺; *Nestin*-GFP⁻ and CD31⁻; PDGFRA⁺; *Nestin*-GFP⁺ single cell derived large-, small- and micro- CFU-F colonies (n = 3). (C) (i) Isotype staining controls. (ii) Percentage of E11.5 *Nestin*-GFP⁺ AGM CD31⁻; PDGFRA⁺; *Nestin*-GFP⁻ and CD31⁻; PDGFRA⁺; *Nestin*-GFP⁺ cells that also express PDGFRB. (D) Long-term growth of E11.5 AGM-derived CFU-Fs based on CD31, PDGFRA, *Nestin*-GFP and PDGFRB expression. (E) Confocal images showing vessel-like structures (left; longitudinal and right; cross section) lined by *Nestin*-GFP⁺ endothelial cells with surrounding PDGFRB⁺ pericytes. These images were taken from tissues harvested at 3 weeks after subcutaneous transplantation of a Matrigel plug loaded with PDGFRA⁺ *Nestin*-GFP⁻CD31⁻PDGFRB⁻ FAC-sorted cells from E11.5 *Nestin*-GFP⁺ AGMs. Colony sizes: micro colonies (<2 mm, 2–24 cells), small colonies (2–4 mm, >25 cells) and large colonies (>4 mm, >100 cells). * p < 0.05, ** p < 0.01, *** p < 0.005; two-sided t-tests were used to compare staining (A and B), and a linear mixed model was used to compare the growth curves (D). Data were derived from biologically independent samples and experiments. Data represent mean ± SD. The precise p values are listed in the source data.

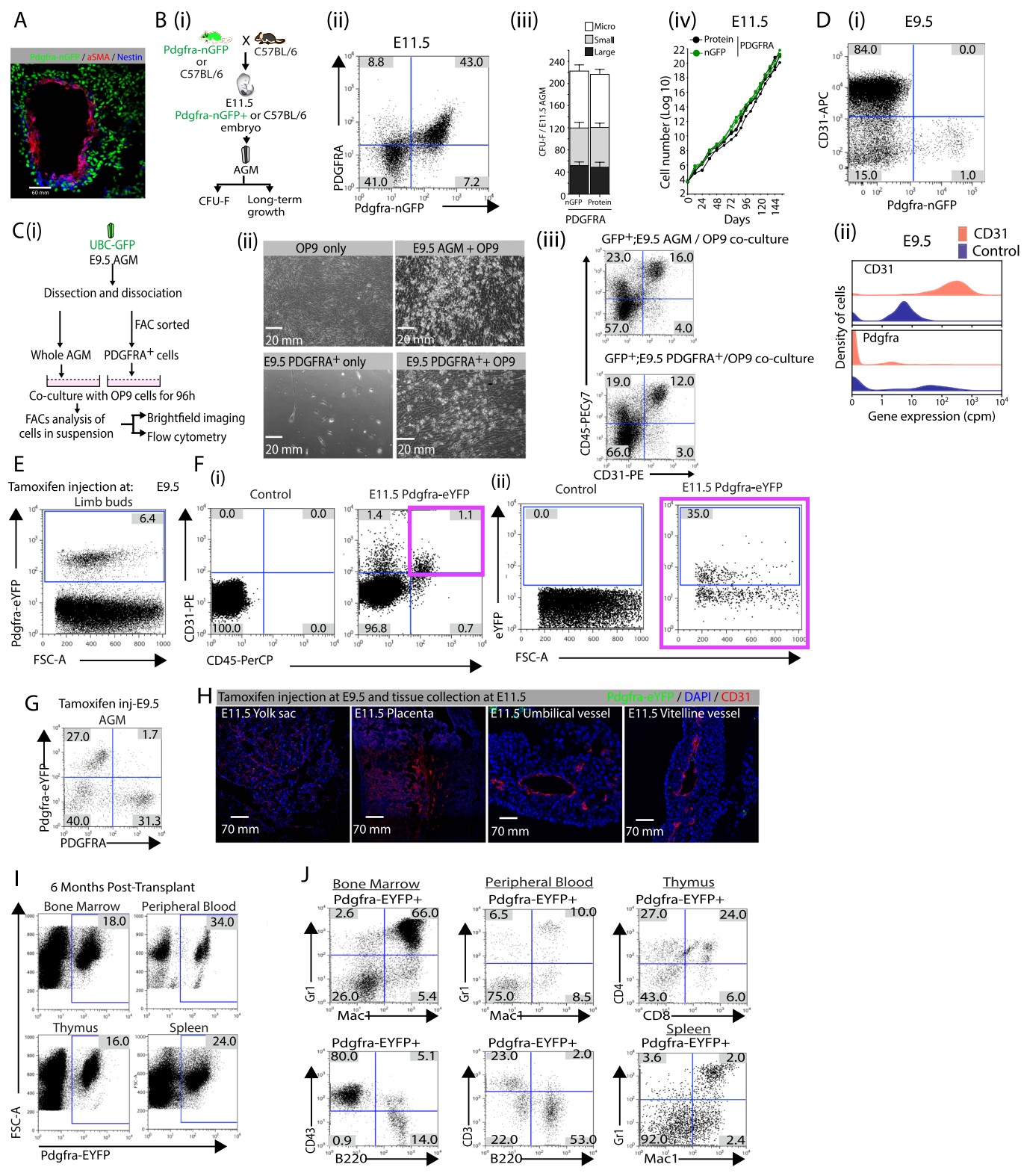

**Extended Data Fig. 3 | See next page for caption.**

**Extended Data Fig. 3 | Characterization of PDGFRA+ cell contributions to the developing aorta and LT-HSCs.** (A) Spatial distribution of *Pdgfra*-nGFP, NESTIN and aSMA expressing cells in a *Pdgfra*-nGFP E11.5 AGM. (B) (i) Schematic outline of experiments using E11.5 wild-type or *Pdgfra*-GFP+ AGM to evaluate comparability of PDGFRA protein and *Pdgfra*-nGFP positivity. (ii) Flow cytometry of E11.5 AGMs harvested from *Pdgfra*-nGFP embryos. (iii) Comparison of CFU-F potential of cells expressing *Pdgfra*-nGFP and cells expressing PDGFRA protein in the E11.5 AGM (n = 5). (iv) Long-term growth of *Pdgfra*-nGFP and PDGFRA protein expressing CFU-Fs (n = 3). (C) (i) Schematic outline of AGM/PSC OP9 co-culture experiments. (ii) Bright-field images of cultures at 96 h. (iii) Flow cytometry analysis showing CD45/CD31 expression in GFP+ cells at 96 h. (D) (i) Flow cytometry of E9.5 *Pdgfra*-nGFP+ AGM for CD31 and GFP expression. (ii) CD31 and Pdgfra expression levels in single CD45−, CD31+, CD144+ endothelial and single CD45−, CD31−, CD144− control cells sorted from the E9.5 embryo proper excluding head, limb buds, heart and visceral bud[44]. (E) Flow cytometry of limb buds harvested from E11.5 *Pdgfra*-cre[ERT2] *R26R*-eYFP embryos after a single injection of tamoxifen at E9.5. (F) (i) Flow cytometry of E11.5 AGM harvested from *Pdgfra*-cre[ERT2] *R26R*-eYFP embryos after a single injection of tamoxifen at E9.5. (ii) eYFP expression in CD45+ CD31+ cells in (i). (G) Flow cytometry of AGMs harvested from E11.5 *Pdgfra*-cre[ERT2] *R26R*-eYFP embryos after a single injection of tamoxifen at E9.5. (H) Confocal images of E11.5 *Pdgfra*-cre[ERT2] *R26R*-eYFP embryo yolk sac, placenta, umbilical and vitelline vessels after a single injection of tamoxifen at E9.5. (I) Flow cytometry analysis of donor eYFP+ cells (from neonatal bone marrow following *cre*-activation at E9.5) in various tissues in primary transplants. (J) Flow cytometry analysis of donor eYFP+ cell (from neonatal bone marrow following *cre*-activation at E9.5) contributions to various blood cell fractions in the bone marrow, peripheral blood, thymus and spleens in recipients 6 months post-transplantation. Colony sizes: micro colonies (<2 mm, 2–24 cells), small colonies (2–4 mm, >25 cells) and large colonies (>4 mm, >100 cells). FSC-A: forward scatter area. A linear mixed model was used to compare the growth curves (B(iv)). Data were derived from biologically independent samples, animals and experiments.

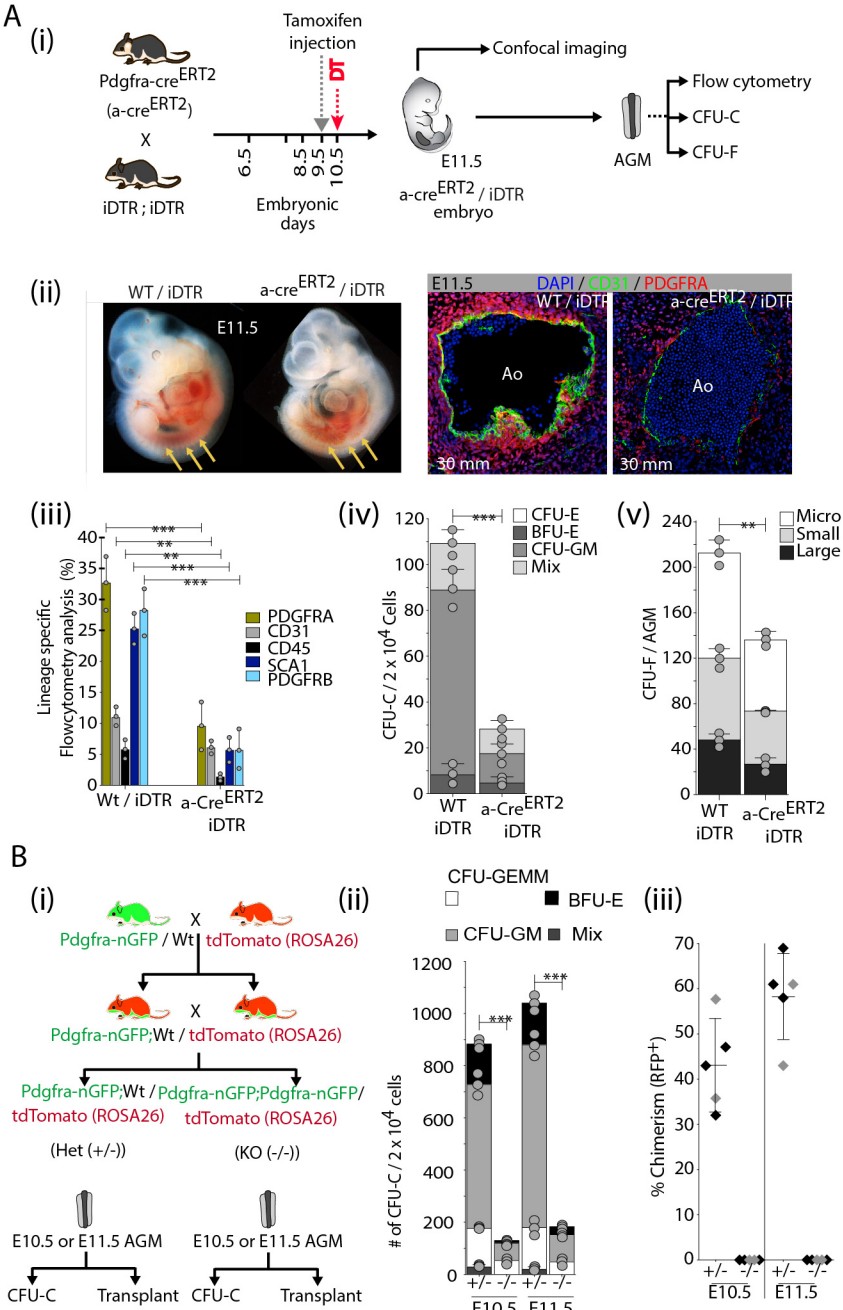

**Extended Data Fig. 4 | PDGFRA+ expression is required for LT-HSC generation in the E11.5 AGM.** (A) (i) Schematic showing the experimental strategy used to ablate PDGFRA+ cells. (ii) Phenotypic changes in E11.5 wild-type – and PDGFRA+ – cell-ablated whole embryos (arrows indicate the AGM region). Confocal images of control (Wt / iDTR) (left) and *Pdgfra-cre^ERT2* / iDTR (right) E11.5 AGMs following tamoxifen induction at E9.5 and diphtheria toxin treatment at E10.5. (iii) Flow cytometry of control (Wt / iDTR) and *Pdgfra-cre^ERT2* / iDTR E11.5 AGM (following tamoxifen induction at E9.5 and diphtheria toxin treatment at E10.5) to quantify numbers of blood (CD45), endothelial (CD31), pericyte (PDGFRB) and CFU-Fs (PDGFRA). (iv) CFU-C quantification in E11.5 AGM (n=14) after tamoxifen-induced PDGFRA+ cell ablation. (v) CFU-Fs in E11.5 AGM (n=12) after tamoxifen-induced PDGFRA+ cell ablation. (B) (i) Schematic showing the experimental strategy used to generate *Pdgfra*-nGFP knockout (KO) embryos. (ii) CFU-Cs in *Pdgfra* knockout E10.5 and E11.5 AGM (n=11). (iii) Flow cytometry analysis of donor tdTomato+ cells (from *Pdgfra* knockout and wild-type AGM) in peripheral blood of recipient mice at 6 months post bone marrow transplantation. CFU-F: colony-forming unit–fibroblast; colony sizes: micro colonies (<2 mm, 2–24 cells), small colonies (2–4 mm, >25 cells) and large colonies (>4 mm, >100 cells). FSC-A: forward scatter area. Ao: aortic lumen; CFU-C: colony-forming unit–culture; CFU-E: colony-forming unit–erythroid; BFU-E: burst-forming unit–erythroid; CFU-GM colony-forming unit–granulocyte/macrophage; Mix: CFU-C with indistinct boundary, possibly co-localization of separate CFU-C or an early split from a single CFU-C. *Pdgfra* KI/KI (null): –/– and *Pdgfra* KI/ + (heterozygote): +/–. * p < 0.05, ** p < 0.01, *** p < 0.005; two-sided t-tests were used to compare marker positivity (A(iii)), and random effects Poisson regression was used to compare colony counts (A(iv), (v)) and B(ii)). Data were derived from biologically independent samples, animals and experiments. Data represent mean ± S.D. The precise p values are listed in the source data.

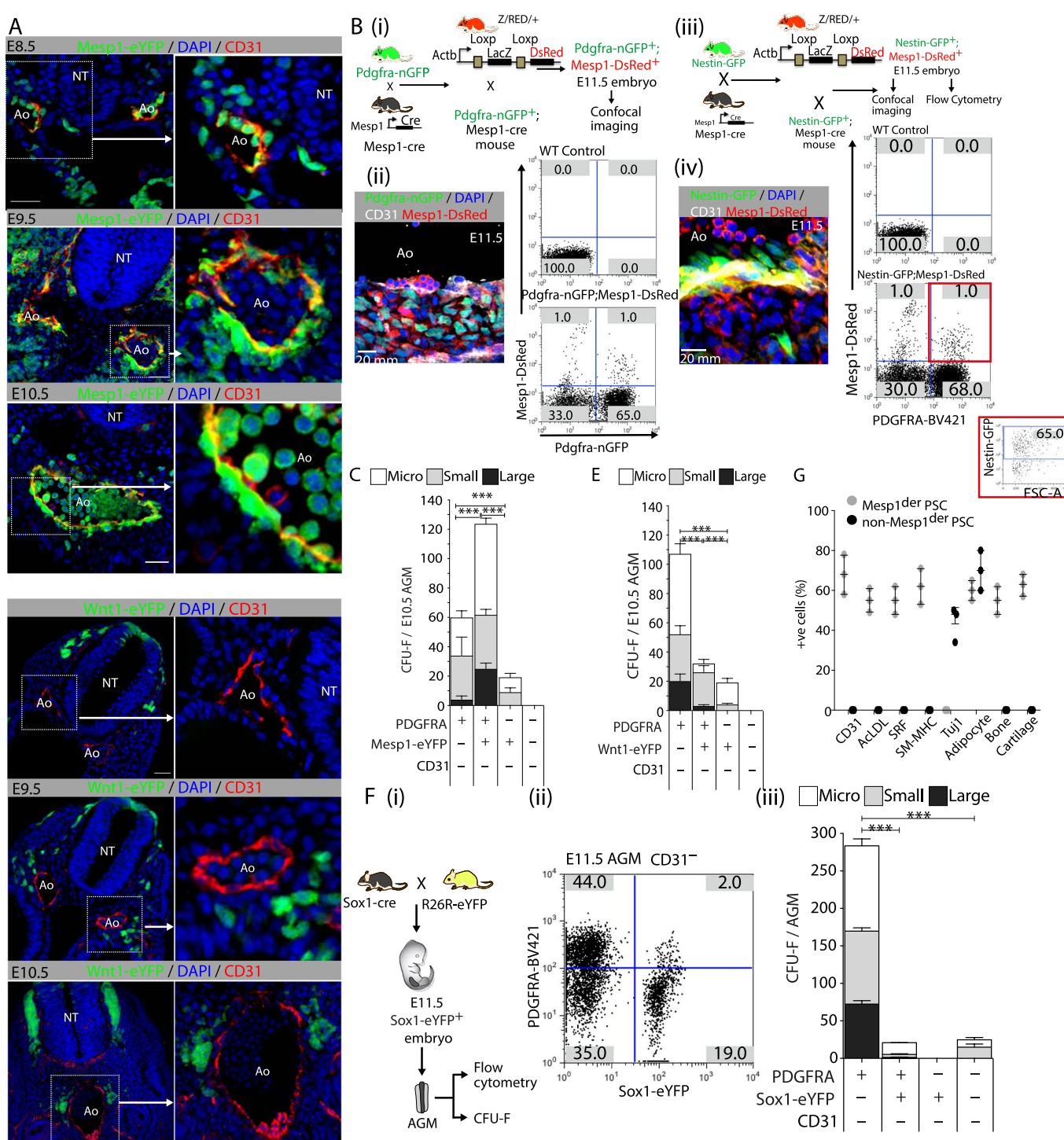

**Extended Data Fig. 5 | See next page for caption.**

**Extended Data Fig. 5 | Germ layer contributions to PDGFRA$^+$ PSCs in the AGM.** (A) Confocal images of E8.5, E9.5 and E10.5 AGMs in *Mesp1*-eYFP embryos. (B) (i) A schematic outline of the genetic cross used to generate and harvest *Mesp1*-DsRed; *Pdgfra*-nGFP embryos (n = 5) at E11.5. (ii) Confocal image of the ventral surface of a E11.5 dorsal aorta in a *Pdgfra*-nGFP/*Mesp1*-DsRed double transgenic embryo showing DsRed$^+$ blood and endothelium and nGFP$^+$/ DsRed$^+$ stromal cells (left); flow cytometry analysis of *Pdgfra*-nGFP/*Mesp1*-DsRed double transgenic E11.5 AGM (right). (iii) A schematic outline of the genetic cross used to generate and harvest *Mesp1*-DsRed; *Nestin*-nGFP embryos (n = 5) at E11.5. (iv) Confocal image of the E11.5 dorsal aorta in a *Nestin*-nGFP/*Mesp1*-DsRed double transgenic embryo showing DsRed$^+$ blood and peri-aortic stromal cells and nGFP$^+$/ DsRed$^+$ endothelial and sub-endothelial stromal cells (right); flow cytometry analysis of *Nestin*-nGFP/*Mesp1*-DsRed double transgenic E11.5 AGM (right). (C) CFU-Fs in cell fractions sorted from *Mesp1*-eYFP$^+$ AGMs (n = 9) at E10.5. (D) Confocal images of E8.5, E9.5 and E10.5 AGMs in *Wnt1*-eYFP embryos. (E) CFU-Fs in cell fractions sorted from *Wnt1*-eYFP$^+$ AGMs (n = 12) at E10.5. (F) (i) A schematic outline of the genetic cross used to generate and harvest *Sox1*-eYFP embryos (n = 12) at E11.5. (ii) Flow cytometry showing the percentage of *Sox1*-eYFP cells in the E11.5 AGM. (iii) CFU-Fs in *Sox1*-eYFP E11.5 AGM. (G) In vitro differentiation potential of *Mesp1*$^{der}$ PSCs and non-*Mesp1*$^{der}$ PSCs. Ao: aortic lumen; DAPI: 4′,6-diamidino-2-phenylindole dihydrochloride; BV421: Brilliant Violet 421; CFU-F: colony-forming unit–fibroblast; colony sizes: micro colonies (<2 mm, 2–24 cells), small colonies (2–4 mm, >25 cells) and large colonies (>4 mm, >100 cells). * p < 0.05, ** p < 0.01, *** p < 0.005; random effects Poisson regression was used to compare colony counts (C, E, F(iii)), and a two-sided t-test was used to compare means in (G). Data were derived from biologically independent samples, animals and experiments. Data represent mean ± S.D. The precise p values are listed in the source data.

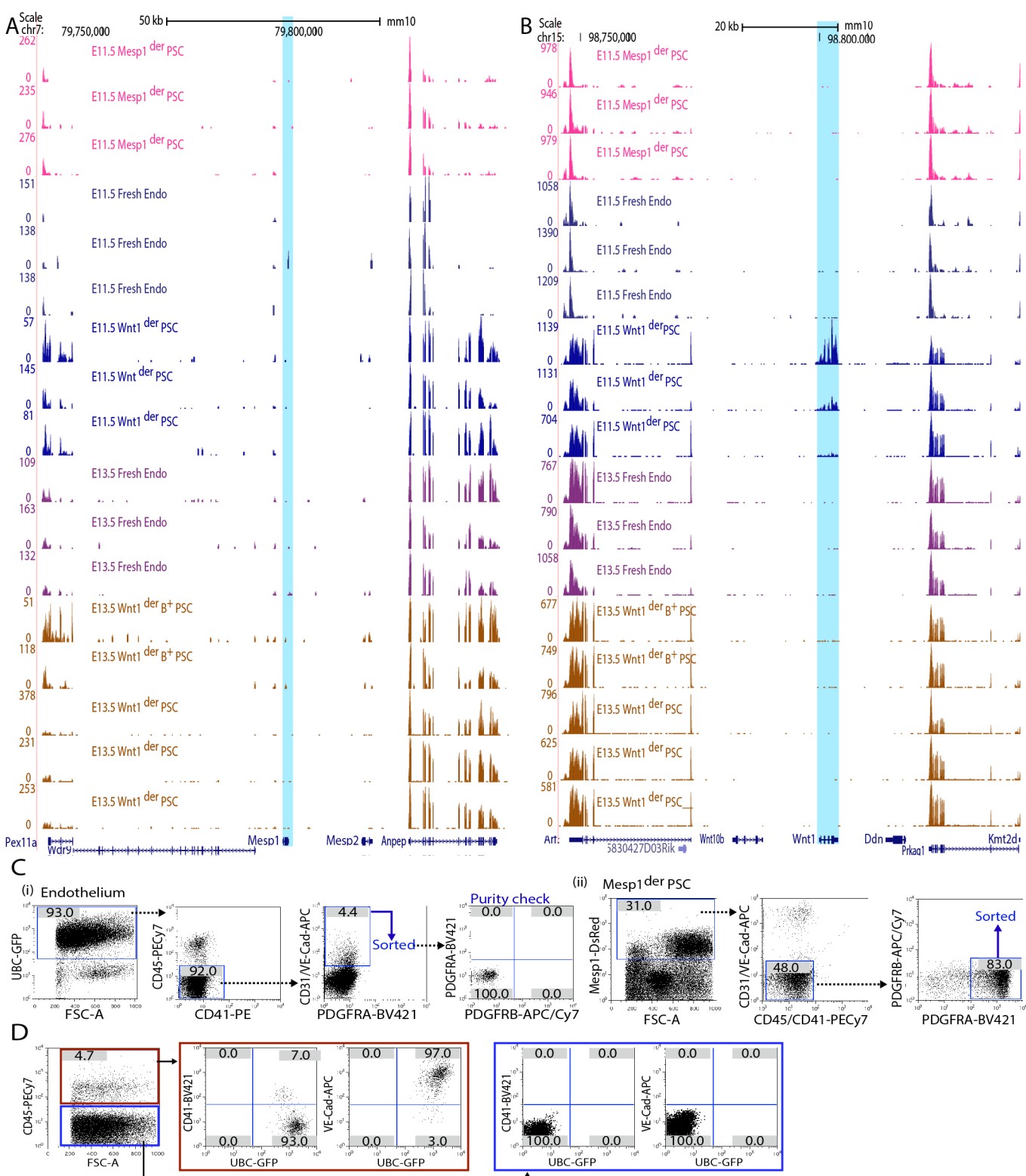

**Extended Data Fig. 6 | Evaluation of *Mesp1* and *Wnt1* transcripts in E11.5 and E13.5 AGM cell subsets.** (A) Evaluation of *Mesp1* expression in AGM cell subsets. RNA sequencing traces mapped to the *Mesp1* locus and its flanking genes. *Mesp1*der PSCs: *Mesp1*-eYFP+/PDGFRA+/PDGFRB−/CD31−/VE-Cad−/CD41−/CD45−, *Wnt1*der PSCs: *Wnt1*-eYFP+/ PDGFRA+/PDGFRB−/CD31−/VE-Cad−/CD41−/CD45−, *Wnt1*der B+ PSCs: *Wnt1*−eYFP+/ PDGFRA+/PDGFRB+/ CD31−/VE-Cad−/CD41−/CD45−and Endo: endothelial cells; PDGFRA−/PDGFRB−/CD31+/VE-Cad+/CD41−/CD45−. (B) Evaluation of *Wnt1* expression in AGM cell subsets. RNA sequencing traces mapped to the *Wnt1* locus and its flanking genes. (C) (i) Gating strategy to FAC sort (i) endothelium and (ii) *Mesp1*-DsRed PSCs. (D) Flow cytometry of a co-aggregate (UBC-GFP E13.5 AGM endothelial cells + E11.5 *Mesp1*-DsRed+ PSC) at 96 h.

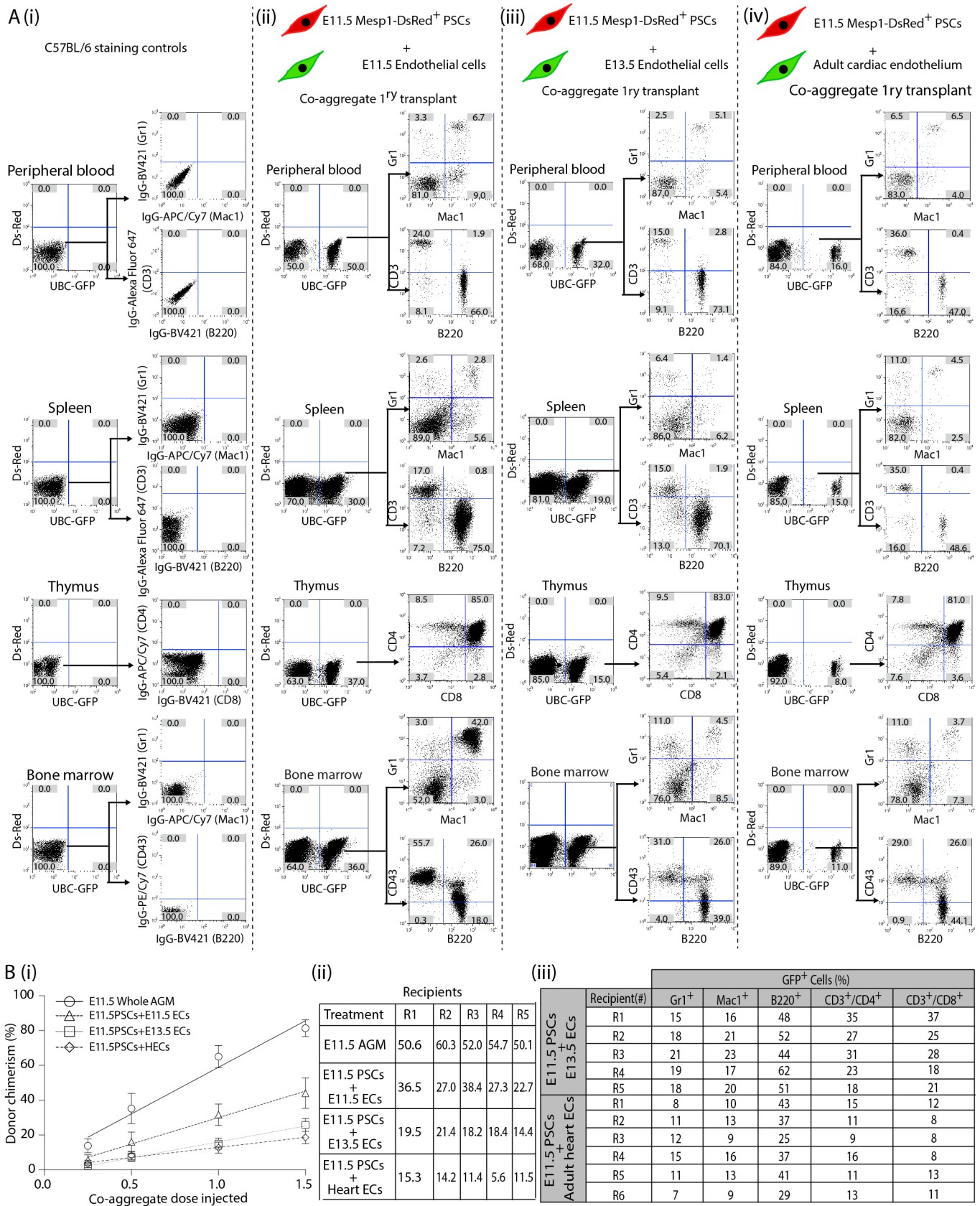

**Extended Data Fig. 7 | See next page for caption.**

**Extended Data Fig. 7 | Multilineage reconstitution from embryonic and adult heart endothelial cell-derived LT-HSCs.** (A) (i) Staining controls using adult C57BL/6 wild-type peripheral blood, spleen, thymus and bone marrow. (ii) Flow cytometry using hematopoietic tissues from recipients 6 months after transplantation with E11.5 endothelium and E11.5 *Mesp1*[der] PSC co-aggregates. (iii) Flow cytometry using hematopoietic tissues from recipients 6 months after transplantation with E13.5 endothelium and E11.5 *Mesp1*[der] PSC co-aggregates. (iv) Flow cytometry using hematopoietic tissues from recipients 6 months after transplantation with adult cardiac endothelium and E11.5 *Mesp1*[der] PSC co-aggregates. (B) (i) Relative donor chimerism and HSC numbers from whole AGM re-aggregates and PSC/endothelial co-aggregates (n = 5). (ii) Table summarizing HSC numbers calculated from serial dilutions/ transplantation. (iii) Table summarizing the contribution of donor cells to various blood lineages in recipient mice. The number of HSCs was estimated using a limited dilution assay as described[18] and calculated in Graph Pad Prism v8.4.3 using nonlinear regression. Data were derived from biologically independent samples, animals and experiments. Data represent mean ± S.D.

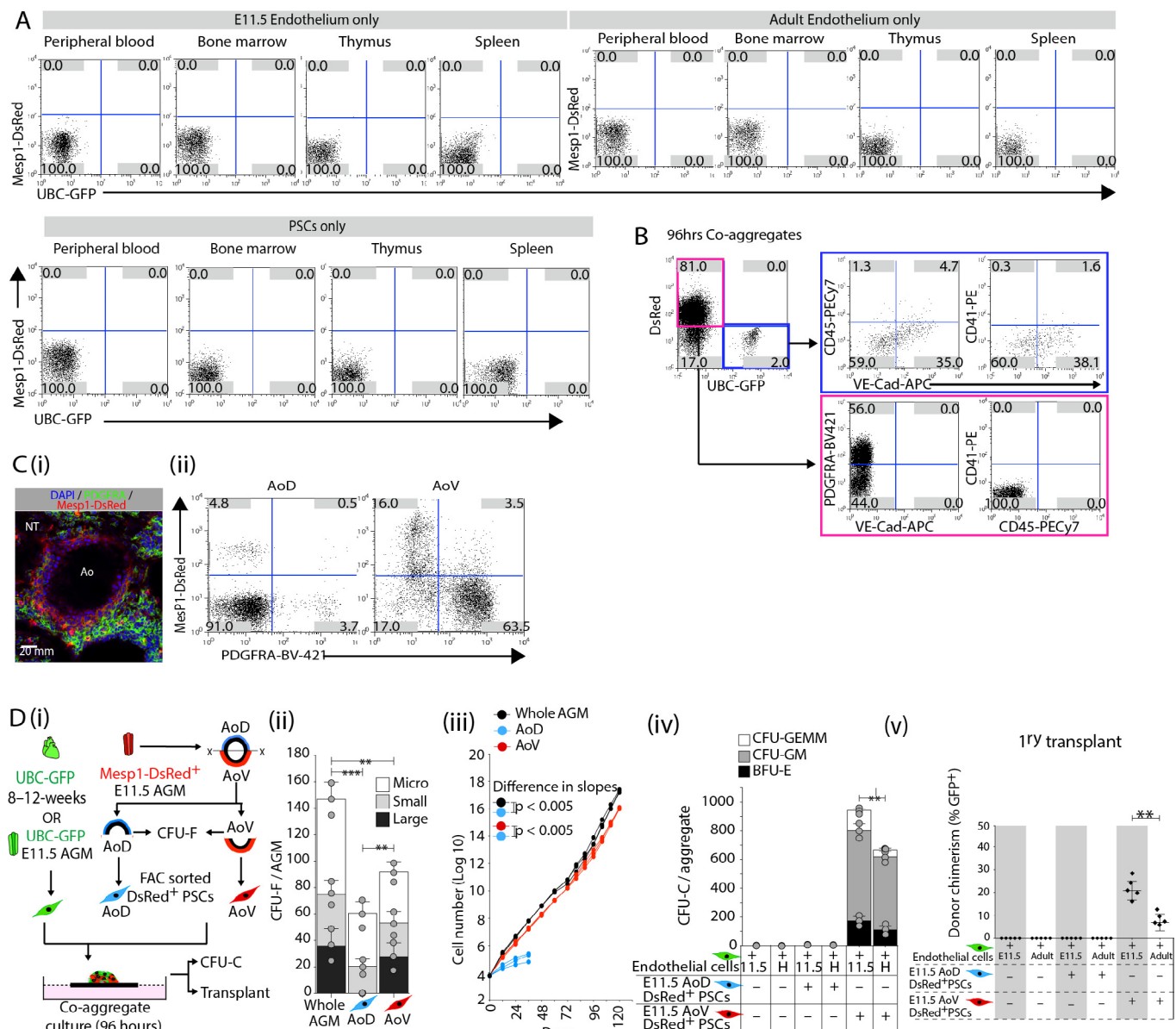

**Extended Data Fig. 8 | *Mesp1*der PSC situated in the ventral region of the dorsal aorta have long-term growth and induce hematopoiesis.** (A) Flow cytometry using hematopoietic tissues from recipients 6 months after transplantation with re-aggregates of E11.5 endothelium or adult cardiac endothelium or E11.5 PSCs. (B) Flow cytometry of pooled co-aggregates (UBC-GFP adult heart endothelial cells + E11.5 *Mesp1*-DsRed⁺ PSC) at 96 h. (C) (i) Confocal image showing PDGFRA expression in the E11.5 dorsal aorta of a *Mesp1*-DsRed embryo. (ii) Flow cytometry of the ventral and dorsal halves of the E11.5 dorsal aorta in *Mesp1*-DsRed embryos (n = 5). (D) (i) Schematic outlining the process of harvesting endothelial cells from E11.5 AGM or adult heart and *Mesp1*der PSCs from the dorsal or ventral halves of the E11.5 aorta for CFU-F assays and co-aggregate cultures. (ii) CFU-F activity of *Mesp1*der PSCs from the whole AGM or dorsal or ventral halves of the E11.5 aorta. (iii) Long-term replating of *Mesp1*der PSCs from the whole AGM or dorsal or ventral halves of the E11.5 aorta. (iv) CFU-C potential of embryonic or adult endothelium co-aggregated with E11.5 *Mesp1*der PSCs from either the dorsal or ventral halves of the aorta. (v) Percentage of GFP⁺ cells in peripheral blood of irradiated recipients 6 months after transplantation of co-aggregates (one co-aggregate for each adult irradiated recipient). CFU-C: colony-forming unit–culture; BFU-E: burst-forming unit–erythroid; CFU-GM: colony-forming unit– granulocyte/macrophage; CFU-GEMM: colony-forming unit–granulocyte/erythrocyte/macrophage/megakaryocyte. * p < 0.05, ** p < 0.01, *** p < 0.005; random effects Poisson regression was used to compare colony counts (D(ii), (iv)), ANOVA was used to compare donor chimerism (D(v)) and a linear mixed model was used to compare the growth curves (D(iii)). Data were derived from biologically independent samples, animals and experiments. Data represent mean ± S.D. The precise p values are listed in the source data.

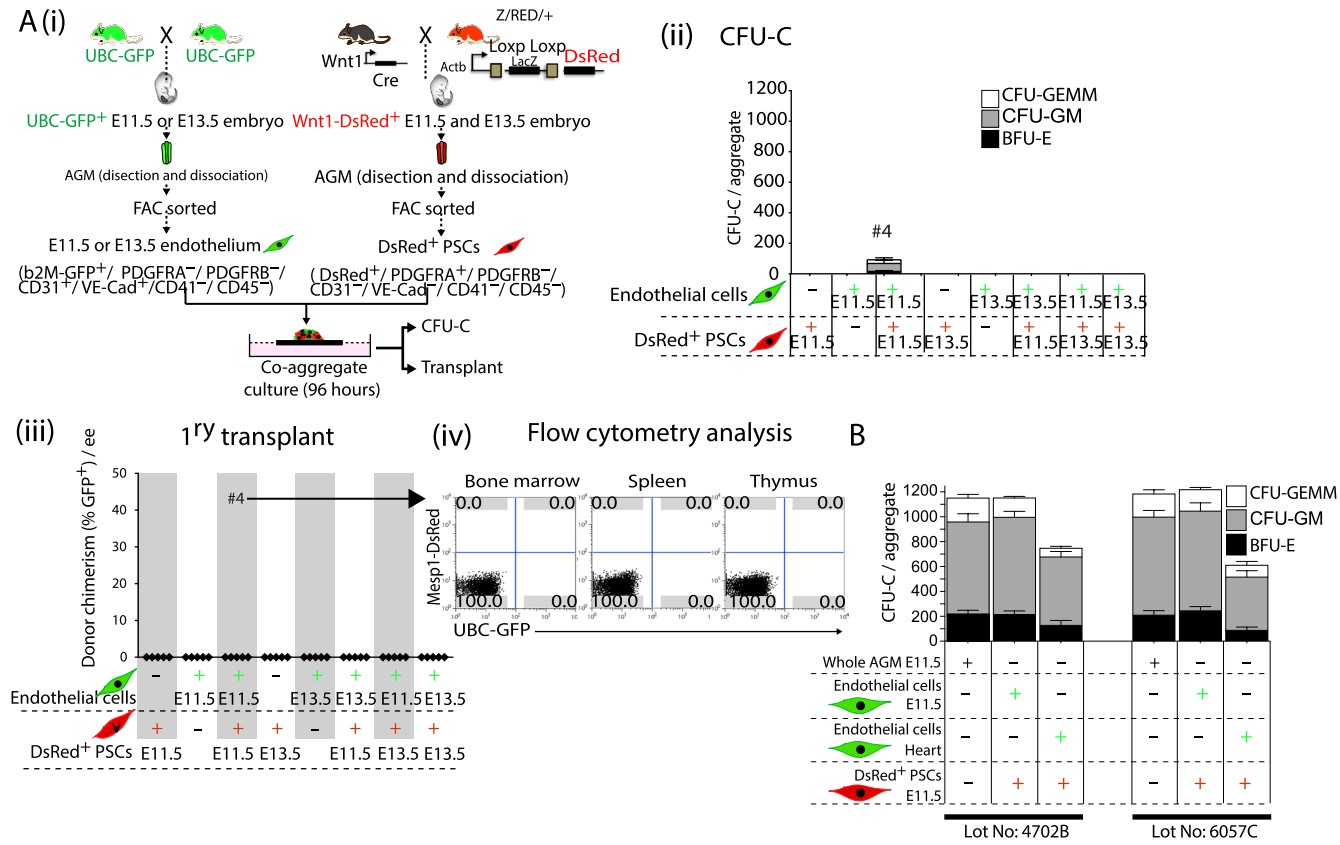

**Extended Data Fig. 9 | *Wnt1^der* PSCs do not induce EHT.** (A) (i) Schematic diagram showing the experimental strategy used to generate cells for co-aggregation of endothelial cells with *Wnt1^der* PSCs. (ii) CFU-C assays from 96-h co-aggregate cultures (n = 5–7 co-aggregates per culture condition). (iii) Percentage of GFP+ cells in peripheral blood of recipients (n = 5 co-aggregates per culture condition). One co-aggregate per recipient mouse and each point represents an individual recipient. (iv) Flow cytometry analysis of bone marrow, spleen and thymus for GFP+ cells. (B) CFU-C generation was also comparable across different lots of FCS. PSC: PDGFRA + stromal cells; AGM: aorta gonad mesonephros; EC: endothelial cell; HEC: adult heart endothelial cells. ANOVA was used to compare colony counts and donor chimerism. Data were derived from biologically independent samples, animals and experiments. Data represent mean ± S.D.

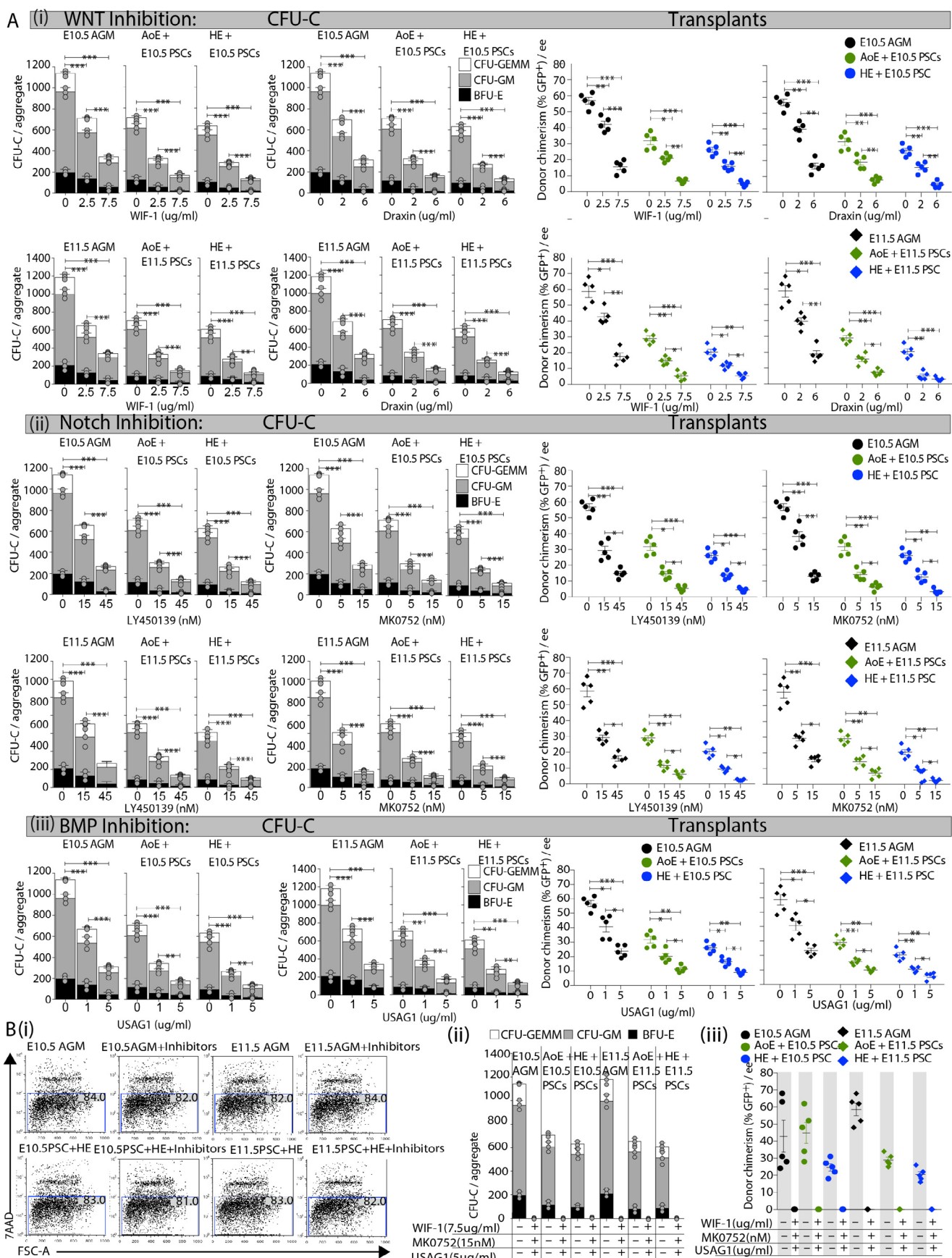

**Extended Data Fig. 10 | See next page for caption.**

**Extended Data Fig. 10 | Conversion of adult endothelial cells into HSC producing hemogenic endothelium requires WNT, NOTCH and BMP signalling.**
(A) PSCs (150,000 cells) from E10.5 and E11.5 embryos were co-aggregated with single cell suspensions from adult heart endothelium (25,000 cells), adult aortic endothelium (25,000 cells) or re-aggregated whole AGM and cultured for 96 h in the presence or absence of stated inhibitors (n = 3). The PSCs were DsRed$^+$ (*Mesp1*-DsRed$^+$/PDGFRA$^+$/PDGFRB$^-$/CD31$^-$/VECad$^-$/ CD41$^-$/CD45$^-$); the endothelial cells were GFP$^+$ (*UBC*-GFP$^+$/PDGFRA$^-$/ PDGFRB$^-$/CD31$^+$/VECad$^+$/CD41$^-$/CD45$^-$). CFU-C potential (left) and donor chimerism at 6 months following bone marrow transplantation (right; one aggregate per irradiated adult recipient) of E10.5 or E11.5 AGM re-aggregates and adult aortic endothelium (AoE) or adult heart endothelium (HE) co-aggregated with E10.5 or E11.5 *Mesp1*-DsRed$^+$ PSCs in the presence of increasing concentrations of (i) WNT inhibitors, (ii) NOTCH inhibitors and (iii) a BMP inhibitor. (B) (i) Flow cytometry showing % of viable (7AAD$^-$) cells in co-aggregates and re-aggregates with and without inhibitors at 96 h (n = 3). (ii) CFU-C potential of E10.5 or E11.5 AGM re-aggregates or adult aortic endothelium (AoE) and adult heart endothelium (HE) co-aggregated with E10.5 or E11.5 *Mesp1*-DsRed$^+$ PSCs in the presence of all three (WNT, NOTCH and BMP) inhibitors (n = 3). (iii) Donor chimerism at 6months following bone marrow transplantation (one aggregate per irradiated adult recipient) of re-aggregates and co-aggregates (n = 3). CFU-C: colony-forming unit–culture; BFU-E: burst-forming unit–erythroid; CFU-GM: colony-forming unit–granulocyte/macrophage; CFU-GEMM: colony-forming unit–granulocyte/erythrocyte/macrophage/megakaryocyte. * $p < 0.05$, ** $p < 0.01$, *** $p < 0.005$; random effects Poisson regression was used to compare colony counts (A(i), (ii), (iii)), and ANOVA was used to compare donor chimerism (A(i), (ii), and (iii)). Data were derived from biologically independent samples, animals and experiments. Data represent mean ± S.D. The precise p values are listed in the source data.

| | |
|---|---|

# Reporting Summary

## Statistics

For all statistical analyses, confirm that the following items are present in the figure legend, table legend, main text, or Methods section.

| n/a | Confirmed | |
|---|---|---|
| ☐ | ☒ | The exact sample size (*n*) for each experimental group/condition, given as a discrete number and unit of measurement |
| ☐ | ☒ | A statement on whether measurements were taken from distinct samples or whether the same sample was measured repeatedly |
| ☐ | ☒ | The statistical test(s) used AND whether they are one- or two-sided<br>*Only common tests should be described solely by name; describe more complex techniques in the Methods section.* |
| ☒ | ☐ | A description of all covariates tested |
| ☐ | ☒ | A description of any assumptions or corrections, such as tests of normality and adjustment for multiple comparisons |
| ☐ | ☒ | A full description of the statistical parameters including central tendency (e.g. means) or other basic estimates (e.g. regression coefficient) AND variation (e.g. standard deviation) or associated estimates of uncertainty (e.g. confidence intervals) |
| ☐ | ☒ | For null hypothesis testing, the test statistic (e.g. *F*, *t*, *r*) with confidence intervals, effect sizes, degrees of freedom and *P* value noted<br>*Give P values as exact values whenever suitable.* |
| ☒ | ☐ | For Bayesian analysis, information on the choice of priors and Markov chain Monte Carlo settings |
| ☐ | ☒ | For hierarchical and complex designs, identification of the appropriate level for tests and full reporting of outcomes |
| ☒ | ☐ | Estimates of effect sizes (e.g. Cohen's *d*, Pearson's *r*), indicating how they were calculated |

*Our web collection on statistics for biologists contains articles on many of the points above.*

## Software and code

Policy information about availability of computer code

| Data collection | L780 LSM Zeiss, Leica SP8 DLS, Imaris software (v9.1), The BD LSRFortessa™ SORP X-20, BD Influx™, Illumina HiSeqX, 10X Genomics Chromium™ Single Cell 3' platform (v3 chemistry) |
|---|---|
| Data analysis | 1. Bulk RNA sequencing - reads were aligned to the mouse genome (mm10) (https://www.ncbi.nlm.nih.gov/assembly/GCF_000001635.20/) using the software STAR (v2.5.0b). Gene expression levels were quantified using HTSeq (v0.9). Expression levels were TMM-normalized using the software package EdgeR (v3.5) in the R statistical analysis software (v3.3.3). Hierarchical clustering with average linkage and Euclidean distance was performed using the Partek Genomics Suite (v 6.6).<br><br>2. Single cell RNA sequencing – To analyze the filtered gene expression counts, we wrote custom Python 3.9 scripts, available at: https://github.com/iosonofabio/scpaper_Vashe. We then used scanpy (v1.8.2) (https://scanpy.readthedocs.io) to log the counts, calculate overdispersed features, perform PCA and UMAP embedding (https://arxiv.org/abs/1802.03426), compute a similarity graph with 10 neighbors, and cluster with the Leiden algorithm then used singlet (https://singlet.readthedocs.io) to make dot plots with a threshold of 0.5 cptt and UMAP projections by cluster and logged gene expression. Pseudotime analysis was performed using scanpy (https://scanpy.readthedocs.io/en/stable/api/scanpy.tl.dpt.html).<br><br>3. Cell Ranger (v3.1.0) was used to process raw datasets including – quality control, the extraction of gene expression matrices, and the aggregation of gene expression matrices from different sequencing runs with batch effect removal.<br><br>4. Statistical data analysis – All statistical analyses were performed using SAS v9.4 (SAS version 9.4, SAS Institute (2016), Cary, NC). SAS statistical analysis and codes are available as Chandrakanthan et al_Statistics_SAS Output.<br><br>5. Flow cytometry- FlowJo™ v10.5+, BD FACSDiva™ v6.0 |

6. GraphPad Prism v8+

For manuscripts utilizing custom algorithms or software that are central to the research but not yet described in published literature, software must be made available to editors and reviewers. We strongly encourage code deposition in a community repository (e.g. GitHub). See the Nature Portfolio guidelines for submitting code & software for further information.

## Data

Policy information about availability of data

All manuscripts must include a data availability statement. This statement should provide the following information, where applicable:
- Accession codes, unique identifiers, or web links for publicly available datasets
- A description of any restrictions on data availability
- For clinical datasets or third party data, please ensure that the statement adheres to our policy

Bulk and single cell RNA-sequencing data were aligned to mm10 (https://www.ncbi.nlm.nih.gov/assembly/GCF_000001635.20/) and have been deposited in Gene Expression Omnibus under GSE163757 and GSE114464 respectively. These data are publicly available. Source data are provided with this paper. All other data supporting the findings of this study are available from the corresponding authors on reasonable request.

# Field-specific reporting

Please select the one below that is the best fit for your research. If you are not sure, read the appropriate sections before making your selection.

☒ Life sciences  ☐ Behavioural & social sciences  ☐ Ecological, evolutionary & environmental sciences

For a reference copy of the document with all sections, see nature.com/documents/nr-reporting-summary-flat.pdf

# Life sciences study design

All studies must disclose on these points even when the disclosure is negative.

| | |
|---|---|
| Sample size | No sample size calculation was performed. Sample sizes were selected on the basis of previous experiments (Reference #80 (Medvinsky et al, 2008), #84 (Taoudi et al., 2008)). Standard practice in molecular and cell biology involves at least three independent biological replicates for each experiment and the number performed for each are mentioned in the figure legends. Data presented in the figures reflect multiple independent experiments (independent cultures and passages were used for each repeat and were obtained on different days) with orthogonal validation where possible. |
| Data exclusions | No data were excluded from analysis. |
| Replication | Data presented in the figures reflect multiple independent experiments, performed on different days using different animals. All experiments were reproduced at least three times. |
| Randomization | Randomization does not apply to cell based experiments, in which large numbers of cells from a given source were partitioned among experimental conditions. Tissues from multiple embryos and adult (12-16 week, female) mice were pooled and then divided as embryo equivalents or defined cell numbers for downstream experiments. |
| Blinding | Investigators were blinded to group allocation during data collection and analysis. |

# Reporting for specific materials, systems and methods

We require information from authors about some types of materials, experimental systems and methods used in many studies. Here, indicate whether each material, system or method listed is relevant to your study. If you are not sure if a list item applies to your research, read the appropriate section before selecting a response.

## Materials & experimental systems

| n/a | Involved in the study |
|---|---|
| ☐ ☒ | Antibodies |
| ☒ ☐ | Eukaryotic cell lines |
| ☒ ☐ | Palaeontology and archaeology |
| ☐ ☒ | Animals and other organisms |
| ☒ ☐ | Human research participants |
| ☒ ☐ | Clinical data |
| ☒ ☐ | Dual use research of concern |

## Methods

| n/a | Involved in the study |
|---|---|
| ☒ ☐ | ChIP-seq |
| ☐ ☒ | Flow cytometry |
| ☒ ☐ | MRI-based neuroimaging |

# Antibodies

**Antibodies used**

A list of antibodies and their source are included in the supplemental information file as a resource table under Reagents or Resource Antibodies (extended data)
REAGENT or RESOURCE SOURCE IDENTIFIER
Antibodies
AcLDL-Alexa Fluor 488 Invitrogen L23380
Anti-GFP Invitrogen A10263
Anti-GFP-Alex-488 Invitrogen A21311
Anti-GFP-Alex-647 Invitrogen A31852
αSarcomaric actinin Sigma 051M4773
B220-BV421 Biolegend 103239
Calponin Abcam ab46794
CD146 Biolegend 134701
CD31 Abcam ab28364
CD31-APC eBioscience 17-0453-82
CD31-PE BD Biosciences 553373
CD31-PE/Cy7 BD Biosciences 561410
CD3-Alexa Fluor 647 Biolegend 100209
CD41- PE/Cy7 Biolegend 133915
CD43-PE/Cy7 Biolegend 143210
CD45 eBioscience 14-0451-82
CD45-eFluor eBioscience 48-0451-82
CD45-FITC Biolegend 103107
CD4-APC/CY7 Biolegend 100414
CD8-BV421 Biolegend 100737
cKIT BD Bioscience 103101
cKIT-APC Biolegend 105811
Gr1-BV421 Biolegend 108433
Hnf4α Santa Cruz Biotechnology sc-6556
Mac1-APCCy7 Biolegend 101226
Mouse Lineage Panel BD Pharmigen 557791
Myh11 Thermo Fisher Scientific MA5-11971
Nestin Millipore MAB353
PDGFRA BD Pharmigen 558774
PDGFRA (APA5) Biolegend 135901
PDGFRA-APC Biolegend 135908
PDGFRA-BV421 Biolegend 562774
PDGFRB Biolegend 136002
PDGFRB (APB5) Biolegend 136002
Purified Anti-YFP Biovision 3991-100
Sca1-PE/Cy7 Biolegend 108114
Serum response factor Abcam ab53147
Tuj1 Santa Cruz Biotechnology sc-80005
VE-Cadherin Santa Cruz Biotechnology sc-9989
VE-Cadherin-PE Biolegend 138105

**Validation**

REAGENT or RESOURCE SOURCE IDENTIFIER DILUTION REFERENCE (PMID)
AcLDL-Alexa Fluor 488 Invitrogen L23380 5ug/ml PMID:20453163, 21885849
Anti-GFP-Biotin Invitrogen A10263 1;400 PMID: 25347465
Anti-GFP-Alex-488 Invitrogen A21311 1;500 PMID: 33907215
Anti-GFP-Alex-647 Invitrogen A31852 1;500 PMID: 28815216
αSarcomaric actinin Sigma 051M4773 1;500 PMID: 21084676
B220-BV421 Biolegend 103239 1;200 PMID: 24719463
Calponin Abcam ab46794 1;300 PMID: 28077619
CD146 Biolegend 134701 1;400 PMID: 24067916
CD31 Abcam ab28364 1;300 PMID: 29208669
CD31-APC eBioscience 17-0453-82 1;200 PMID: 27022143
CD31-PE BD Biosciences 553373 1;350 PMID: 7956830
CD31-PE/Cy7 BD Biosciences 561410 1;350 PMID: 7956830
CD3-Alexa Fluor 647 Biolegend 100209 1;200 PMID: 29466757
CD41- PE/Cy7 Biolegend 133915 1;300 PMID: 27183606
CD41-PE Biolegend 133906 1;100 PMID: 26193121
CD41-BV421 Biolegend 133911 1;150 PMID: 25840412
CD43-PE/Cy7 Biolegend 143210 1;300 PMID: 15778363
CD45 eBioscience 14-0451-82 1;300 PMID: 26347471
CD45-eFluor eBioscience 48-0451-82 1;150 PMID: 27525437
CD45-FITC Biolegend 103107 1;200 PMID: 16709810
CD45-PE/Cy7 Biolegend 103114 1;300 PMID: 16709810
CD4-APC/CY7 Biolegend 100414 1;200 PMID: 23851361
CD8-BV421 Biolegend 100737 1;150 PMID: 16116223
cKIT BD Bioscience 103101 1;300 PMID: 7508684
cKIT-APC Biolegend 105811 1;300 PMID: 20512127
Gr1-BV421 Biolegend 108433 1;150 PMID: 16142239

Hnf4α Santa Cruz Biotechnology sc-6556 1;400 PMID: 32880442
Mac1-APCCy7 Biolegend 101226 1;400 PMID: 24431111
Mouse Lineage Panel BD Pharmigen 557791 1;400 PMID: 9169840
Myh11 Thermo Fisher Scientific MA5-11971 1;400 PMID: 32102389
Nestin Millipore MAB353 1;300 PMID: 25683249
PDGFRA BD Pharmigen 558774 1;400 PMID: 8875964
PDGFRA (APA5) Biolegend 135901 1;300 PMID: 8875964
PDGFRA-APC Biolegend 135908 1;200 PMID: 26056396
PDGFRA-BV421 Biolegend 562774 1;200 PMID: 26056396
PDGFRB Biolegend 136002 1;350 PMID: 29861387
PDGFRB (APB5) Biolegend 136002 1;200 PMID: 11413086
Purified Anti-YFP Biovision 3991-100 1;250 PMID: 31387989
Sca1-PE/Cy7 Biolegend 108114 1;300 PMID: 19443245
Serum response factor Abcam ab53147 1;250 PMID: 27323859
Tuj1 Santa Cruz Biotechnology sc-80005 1;300 PMID: 31619962
VE-Cadherin Santa Cruz Biotechnology sc-9989 1;300 PMID: 31863691
VE-Cadherin-PE Biolegend 138105 1;300 PMID: 11156369
VE-Cadherin-APC Biolegend 138012 1;200 PMID: 11156369

# Animals and other organisms

Policy information about studies involving animals; ARRIVE guidelines recommended for reporting animal research

| Laboratory animals | 1. C57BL/6J- 12-16 weeks old Males and females used for time mating to get age appropriate embryos. Females were used for transplant experiments. 3-4 females used to get age appropriate embryos and 3-5 females used for each transplant experiments (please see figures for exact numbers).
2. Pdgfratm11(EGFP)Sor- 12-16 weeks old Males and females used for time mating to get age appropriate embryos. 4-6 females used to get age appropriate embryos per experiment.
3. Gt(ROSA)26Sortm(EYFP)Cos/J-This is a reporter mouse strain. 12-16 weeks old Males or females used for time mating with male or female Cre-lines to get  age appropriate embryos. 4-8 females used to get age appropriate embryos per experiment
Tg(CAG-DsRed*MST)1Nagy- This is a reporter mouse strain. Males or females used for time mating with male or female Mesp1 or Wnt1 Cre-lines to get  age appropriate embryos. 4-8 females used to get age appropriate embryos per experiment.
C57BL/6Tg(UBC-GFP)30Scha/J- 12-16 weeks old Males and females used for time mating to get age appropriate embryos. 12-16 weeks old Females were used to harvest heart, lung, aorta and IVC endothelial cells. 3 females used to get age appropriate embryos per experiment.
Mesp1tm2(cre)Ysa- 12-16 weeks old Males and females used for time mating with reporter mouse lines to get age appropriate embryos.4-8 females used to get age appropriate embryos per experiment.
Tg(Wnt1-cre)11Rth- 12-16 weeks old Males and females used for time mating with reporter mouse lines to get age appropriate embryos.4-8 females used to get age appropriate embryos per experiment.
Sox1tm1(cre)Take- 12-16 weeks old Males and females used for time mating with reporter mouse lines to get age appropriate embryos.4-8 females used to get age appropriate embryos per experiment.
Tg(Pdgfra-cre/ERT2)1Wdr- 12-16 weeks old Males and females used for time mating with reporter mouse lines to get age appropriate embryos for lineage tracing studies. 6-8 females used to get age appropriate embryo.
Tg(Nes-EGFP)33Enik-12-16 weeks old Males and females used for time mating to get age appropriate embryos. 4-6 females used to get age appropriate embryos per experiment.
Gt(ROSA)26Sortm1(HBEGF)Awai-12-16 weeks old Males and females used for time mating with reporter mouse lines to get age appropriate embryos for lineage tracing studies. 6-8 females used to get age appropriate embryos

Animals were housed and bred at the Biological Resources Centre at UNSW, a specific pathogen free, PC2 facility with semi-natural light cycle of 12:12 hours light: dark, and regulated air quality, ventilation (15 ACH), humidity (55%) and temperature (22C)- see methods. |
| Wild animals | Study did not involve wild animals. |
| Field-collected samples | Study did not involve samples collected from the field. |
| Ethics oversight | All animal experiments were approved by the Animal Ethics Committee of UNSW Sydney, Sydney, NSW, Australia |

Note that full information on the approval of the study protocol must also be provided in the manuscript.

# Flow Cytometry

## Plots

Confirm that:

☒ The axis labels state the marker and fluorochrome used (e.g. CD4-FITC).

☒ The axis scales are clearly visible. Include numbers along axes only for bottom left plot of group (a 'group' is an analysis of identical markers).

☒ All plots are contour plots with outliers or pseudocolor plots.

☒ A numerical value for number of cells or percentage (with statistics) is provided.

## Methodology

Sample preparation

Sample preparation was in accordance with published protocols (references are included in the methods section).

Instrument

Mononuclear staining was analysed on a BD LSRFortessa (BD Biosciences). Cell sorts were performed on a BD Influx (BD Biosciences).

Software

BD FACSDiva™ Software v6 (BD Biosciences) and FACS data were analysed using FlowJo v10.5+ software (TreeStar).

Cell population abundance

We sorted PSCs and endothelial cells separately.

Gating strategy

First gated for FSC-A/FSC-H and eliminated all doublets and then gated for FSC-A/SSC-A to select mononuclear cells. From there we used relevant fluorochromes to gate and select respective cell populations for our experiments. Experimental details are listed in the main and supplemental information files (including controls).

☒ Tick this box to confirm that a figure exemplifying the gating strategy is provided in the Supplementary Information.

