## [Peer Review File · Nature Cell Biology]

Peer Review Information

Journal: Nature Cell Biology

Manuscript Title: Mesoderm-Derived PDGFRA+ Cells Regulate the Emergence of Hematopoietic Stem Cells in the Dorsal Aorta.

Corresponding author name(s): John Pimanda

Reviewer Comments & Decisions:

Decision Letter, initial version:

Dear Professor Pimanda,

Your manuscript, "Mesoderm-Derived PDGFRA+ Cells Regulate the Emergence of Hematopoietic Stem Cells in the Dorsal Aorta.", has now been seen by 3 referees, who are experts in hematopoiesis, DA, AGM, endothelium (referee 1); hematopoiesis, HE, vascularization, development (referee 2); and hematopoiesis, AGM, endothelium (referee 3). As you will see from their comments (attached below) they find this work of potential interest, but have raised substantial concerns, which in our view would need to be addressed with considerable revisions before we can consider publication in Nature Cell Biology.

Nature Cell Biology editors discuss the referee reports in detail within the editorial team, including the chief editor, to identify key referee points that should be addressed with priority, and requests that are overruled as being beyond the scope of the current study. To guide the scope of the revisions, I have listed these points below. We are committed to providing a fair and constructive peer-review process, so please feel free to contact me if you would like to discuss any of the referee comments further.

In particular, it would be essential to:

A) Acknowledge and discussion related literature:

Reviewer 2

"Previous studies by Ding <https://doi.org/10.1002/dvdy.23923> with knockout animals revealed that loss of PDGFRA had no effect on fetal liver HSCs. However, present study has found a profound effect of Pdgfra knockout or inhibition with APA5 antibody on HSC and blood production. How this could be explained?"

"Another important difference between findings presented in this manuscript and the Ding paper is the developmental window at which contribution of PDGFRa+ cells to blood cells was observed. Ding found no contribution of PDGFRa+ cells to HSCs when tamoxifen was injected at E9.5. In contrast, this study revealed a substantial labeling blood cells following tamoxifen treatment at E9.5. This should be discussed."

Reviewer 3

"The authors show E9.5 labeled Pdgfra+ cells contribute to endothelial cells and HSC in the dorsal aorta. Ding, G. et al., (doi: 10.1002/dvdy.23923) showed by lineage tracing that Pdgfra+ mesoderm labeled at E7.5-E8 contributes to endothelial cells in the dorsal aorta and to adult hematopoietic cells, including KLS cells. The authors should acknowledge these findings in this section and add their comments on how their results add to the previous report."

"The previous findings of Saga, Y. et al, (doi: 10.1016/s1050-1738(01)00069-x.) showing contribution of Mesp1 mesoderm to the dorsal aorta should be acknowledged."

B) Improve quantification and statistical analysis:

Reviewer 1

"The authors should quantitate the total number of CFU-F present in the AGM region over the stages E9.5-13.5. It is important to know the total number present in a tissue (at each embryonic stage) and then compare this number to the total number recovered in the various sorted cell populations. Only then can the authors claim (page 6 line 20/21) "Taken together, these data show that PDGFRFA marks all CFU-Fs in the E11.5 AGM...."

"The authors show numerous growth curves (example Fig. 1E) but average results of 3 studies. All of the individual growth curves for n=3 should be shown to assess the variance among the individual groups."

"There is no indication how many times the clonal analyses (Fig. 1F) were performed."

Reviewer 3

"...Several results show no statistical analysis, or they lack the reference that no statistical significance was observed..."

C) Provide sufficient experimental details as requested by Reviewer 3:

"The authors should include information of how many embryo equivalents per recipient are used for direct transplantation of AGM cells (Figure 2)."

"The authors' findings that co-aggregation of DA endothelium and, particularly adult endothelium with embryo-derived Mesp1der PSC instructs EHT and HSC emergence are very interesting. There are experimental details and inclusion of more data that can clarify/strengthen the findings..."

2D) All other referee concerns pertaining to strengthening existing data, providing controls, methodological details, clarifications and textual changes as applicable should also be addressed.

E) Finally please pay close attention to our guidelines on statistical and methodological reporting (listed below) as failure to do so may delay the reconsideration of the revised manuscript. In particular please provide:

- a Supplementary Figure including unprocessed images of all gels/blots in the form of a multi-page pdf file. Please ensure that blots/gels are labeled and the sections presented in the figures are clearly indicated.
- a Supplementary Table including all numerical source data in Excel format, with data for different figures provided as different sheets within a single Excel file. The file should include source data giving rise to graphical representations and statistical descriptions in the paper and for all instances where the figures present representative experiments of multiple independent repeats, the source data of all repeats should be provided.

We would be happy to consider a revised manuscript that would satisfactorily address these points, unless a similar paper is published elsewhere, or is accepted for publication in Nature Cell Biology in the meantime.

- ensure that it conforms to our format instructions and publication policies (see below and <https://www.nature.com/nature/for-authors>).
- provide a point-by-point rebuttal to the full referee reports verbatim, as provided at the end of this letter.
- provide the completed Reporting Summary (found here <https://www.nature.com/documents/nr-reporting-summary.pdf>). This is essential for reconsideration of the manuscript will be available to editors and referees in the event of peer review. For more information see <http://www.nature.com/authors/policies/availability.html> or contact me.

When submitting the revised version of your manuscript, please pay close attention to our [href="https://www.nature.com/nature-research/editorial-policies/image-integrity">Digital Image Integrity Guidelines](https://www.nature.com/nature-research/editorial-policies/image-integrity). and to the following points below:

Nature Cell Biology is committed to improving transparency in authorship. As part of our efforts in this direction, we are now requesting that all authors identified as 'corresponding author' on published papers create and link their Open Researcher and Contributor Identifier (ORCID) with their account on the Manuscript Tracking System (MTS), prior to acceptance. ORCID helps the scientific community achieve unambiguous attribution of all scholarly contributions. You can create and link your ORCID from the home page of the MTS by clicking on 'Modify my Springer Nature account'. For more information please visit www.springernature.com/orcid.

This journal strongly supports public availability of data. Please place the data used in your paper into a public data repository, or alternatively, present the data as Supplementary Information. If data can only be shared on request, please explain why in your Data Availability Statement, and also in the correspondence with your editor. Please note that for some data types, deposition in a public repository is mandatory - more information on our data deposition policies and available repositories appears below.

[REDACTED]

We would like to receive a revised submission within six months.

We hope that you will find our referees' comments, and editorial guidance helpful. Please do not hesitate to contact me if there is anything you would like to discuss.

Best wishes,

Zhe Wang

Zhe Wang, PhD
Senior Editor
Nature Cell Biology

Tel: +44 (0) 207 843 4924
email: zhe.wang@nature.com

Reviewers' Comments:

Reviewer #1:

Remarks to the Author:

The authors investigated whether cells with mesenchymal stem cell-like activity, which provide an essential niche for long-term HSCs (LT-HSCs) in the bone marrow, reside in the aorta6 gonad-mesonephros (AGM) and contribute to the structural development of the dorsal aorta and endothelial to hematopoietic transition (EHT). Using transgenic mice, they identify a lineage hierarchy for AGM stromal cells and traced E10.5/E11.5 aortic endothelium and HSCs to mesoderm derived (Mesp1) PDGFRA+ stromal cells (Mesp1der PSCs). They determined that these cells dominate the sub-endothelial and ventral stroma in the E10.5–E11.5 AGM but by E13.5 were replaced by neural crest (Wnt1) derived PDGFRA+ stromal cells (Wnt1der PSCs). Co-aggregating non-hemogenic embryonic and adult endothelial cells with Mesp1der PSCs but not with Wnt1der 13 PSCs resulted in activation of a hematopoietic transcriptional program in embryonic and adult endothelial cells accompanied by EHT and generation of LT-HSCs. Dose-dependent inhibition of PDGFRA signaling or BMP, WNT, NOTCH signaling interrupted this reprogramming event. The authors postulate that such endothelial and PSC interactions may prove beneficial in translating generation of HSC in vitro.

This is novel insights into an area that has been of interest to developmental and stem cell scientists for many years. They directly contributed to new knowledge about the origins of stromal cells that regulate the endothelial cell behavior in becoming hemogenic endothelium with HSC emergence. The methods used are state of the art and appropriate controls are used throughout. The flow of studies is easy to follow and the salient points summarized at key steps. The conclusions are supported by the data presented.

A key deficiency in the presentation of the results are the sporadic use of statistical tests. To bolster confidence, the authors should statistically analyze all quantitative data for level of significance.

There are some questions and points to consider:

- 1) The authors should quantitate the total number of CFU-F present in the AGM region over the stages E9.5-13.5. It is important to know the total number present in a tissue (at each embryonic stage) and then compare this number to the total number recovered in the various sorted cell populations. Only then can the authors claim (page 6 line 20/21) "Taken together, these data show that PDGRFA marks all CFU-Fs in the E11.5 AGM...."
- 2) The authors show numerous growth curves (example Fig. 1E) but average results of 3 studies. All of the individual growth curves for n=3 should be shown to assess the variance among the individual groups.
- 3) There is no indication how many times the clonal analyses (Fig. 1F) were performed.
- 4) Fig. 1J and Movie 2 do not convince this reviewer that the cells formed lumenized vessel structures in vivo. One would need to see intraluminal red blood cells or evidence of systemically infused fluorescent dyes to be sure these are vessels.

5- 5) Fig. 2C(ii) does not indicate which are primary or secondary recipient mice.
- 6) The doses used in combination in Ext. data Fig. 6C (i-ii) were near dose limiting when used as single agents. One would question whether the inhibition of CFU-C and LT-HSC activity is because all of the cells died. Some evidence of cell viability at those combined doses (or lowered combined doses that assure cell viability) are required.

Reviewer #2:

Remarks to the Author:

In this manuscript, Chandrakanthan et al. investigated the origin and properties of stromal cells in AGM region. They show that periaortic stromal cells in this location are derived from *Mesp1*⁺ progenitors (*Mesp1*^{der}) but later replaced by stromal cells derived from neural crest (*Wnt1*^{der}). Coaggregation of *Mesp1*^{der} or *Wnt1*^{der} stromal cells with endothelial cells revealed that aggregates of *Mesp1*^{der} stromal cells with AGM endothelium including nonhemogenic E13.5 endothelium and endothelium from hearts and aorta of 8-12 weeks old animals produced HSCs with robust engraftment potential. This effect was observed only with E10.5 or E11.5 *Mesp1*^{der} stromal cells but not with E13.5 *Wnt1*^{der} stromal cells. In addition, authors demonstrated that treatment of endothelial/stromal aggregates or ex vivo cultured E10.5 embryos with PDGFR α inhibitor reduced blood and HSC formation. The differences in *Mesp1*^{der} stromal cells from ventral and dorsal fraction of were also revealed. Overall, these studies are well executed and make a significant novel contribution to our understanding of HSC development.

1. Previous studies by Ding <https://doi.org/10.1002/dvdy.23923> with knockout animals revealed that loss of PDGFR α had no effect on fetal liver HSCs. However, present study has found a profound effect of Pdgfra knockout or inhibition with APA5 antibody on HSC and blood production. How this could be explained?
2. Another important difference between findings presented in this manuscript and the Ding paper is the developmental window at which contribution of PDGFR α ⁺ cells to blood cells was observed. Ding found no contribution of PDGFR α ⁺ cells to HSCs when tamoxifen was injected at E9.5. In contrast, this study revealed a substantial labeling blood cells following tamoxifen treatment at E9.5. This should be discussed.
3. PDGFR α -positive cells in the AGM region of E9.5 embryos could still contain multipotential precursors, including Flk1⁺ mesodermal precursors which are known to contribute to HSC in the embryo proper. What is the proportion of FLK1⁺ cells within PDGFR α ⁺ cells at E9.5 in the mouse model used in this study?
4. Tracing experiments demonstrated that blood and hemogenic endothelium originate from PDGFR α ⁺ cells found in periaortic mesenchyme. Any evidence that these cells possess hematopoietic potential in vitro, for example in coculture with OP9?
5. How embryos were exposed to APA5 and APB5 antibodies. Were they injected with antibodies or

6antibodies simply added to ex vivo cultures? In the later scenario, any evidence that antibodies can reach AGM by diffusion?

6. Page 11 lines 20-23. *Pdgfra*-nGFP+/*Mesp1*-DsRed mice are introduced here for the first time. However, it is unclear whether *Mesp1*-DsRed is knock-in reporter or it traces *Mesp1*- Cre recombination. If *Mesp1*-DsRed is knock-in reporter, why DsRed signal is seen in blood and endothelial cells. Expression of *Mesp1* is downregulated in endothelial and blood cells.

7. Transcriptional analysis of adult heart and aortic endothelium before and after coaggregation would be very interesting.

8. Please spell out PSC abbreviation.

Reviewer #3:

Remarks to the Author:

Chandrakanthan, V. et al. investigated the origin and contribution of *Pdgfra*+ cells to hematopoietic stem cell (HSC) development in the dorsal aorta. The study shows *Pdgfra*+ cells from mesodermal origin contribute to endothelial cells and HSC in the dorsal aorta, and to sub-endothelial mesenchyme at the time HSC emerge. This mesoderm-derived mesenchyme supports endothelial-to-hematopoietic transition (EHT) from hemogenic endothelium and is later replaced by neural crest-derived *Pdgfra*+ cells that lack this ability, coinciding with the cessation of hematopoiesis in the DA. Mesoderm-derived *Pdgfra*+ cells not only enable EHT from hemogenic endothelium but also from non-hemogenic embryonic and adult endothelial cells, in a process dependent on *Pdgfra* signaling and BMP, WNT and Notch pathways.

The manuscript includes interesting, novel findings and builds upon previous reports, such as the contribution of *Mesp1*-mesoderm and *Pdgfra*+ cells to the dorsal aorta and to adult hematopoiesis, with transplantation assays that bring more robustness to the previous reports (references below) that were based on lineage tracing. The findings that *Mesp1*-derived PSC can instruct adult EC to undergo EHT and generate LT-HSC in vitro are interesting and significant, though this reviewer suggests the inclusion of more experimental details and the performance of experiments that might confer more robustness to the findings. Several results show no statistical analysis, or they lack the reference that no statistical significance was observed. Overall, the manuscript is well written and presents high-quality and significant data, but the clarification of some points and the performance of additional experiments, suggested below, may confer more robustness and validity to the findings and conclusions of the manuscript.

1. The authors show E9.5 labeled *Pdgfra*+ cells contribute to endothelial cells and HSC in the dorsal aorta. Ding, G. et al., (doi: 10.1002/dvdy.23923) showed by lineage tracing that *Pdgfra*+ mesoderm labeled at E7.5-E8 contributes to endothelial cells in the dorsal aorta and to adult hematopoietic cells, including KLS cells. The authors should acknowledge these findings in this section and add their comments on how their results add to the previous report.

72. Rybtsov, S. et al. (doi.org/10.1084/jem.20102419) showed that at E11.5 cells with long-term repopulating ability reside within the VE-Cadherin+CD45+ population. Are all CD31+CD45+ labeled with Pdgfra-eYFP at E11.5 (Figure2)?

3. The authors should include information of how many embryo equivalents per recipient are used for direct transplantation of AGM cells (Figure 2).

4. The previous findings of Saga, Y. et al, (doi: 10.1016/s1050-1738(01)00069-x.) showing contribution of Mesp1 mesoderm to the dorsal aorta should be acknowledged.

5. Based on the confocal image results in Figure3 it appears Mesp-1 derived cells, though more abundant ventrally, are also present dorsally. The transplants performed assessing the role of Mesp1-DsRed+Pdgfra+ isolated from dorsal and ventral aorta suggest PSC cells are located both dorsally and ventrally. Do the authors see differences in numbers and a preferential localization in the ventral aorta?

6. Most Pdgfra+ cells present in the aorta at E11.5 are not derived from Mesp1-mesoderm or Wnt1 neural crest. Are Mesp1-derived PSC more limited in differentiation potential compared to Mesp1neg PSC? Assessment of Nestin expression in Mesp1der PSC should be shown.

7. The authors' findings that co-aggregation of DA endothelium and, particularly adult endothelium with embryo-derived Mesp1der PSC instructs EHT and HSC emergence are very interesting. There are experimental details and inclusion of more data that can clarify/strengthen the findings:

7.1. There is inconsistency in the cell numbers presented to form the co-aggregates in the figure legends and the methods section.

7.2. The authors should include the flow plots showing how the sorting strategy for isolation of EC and DsRed+ PSC was performed (including the fluorophores used), as well as transplants with freshly isolated EC populations to discard any possibility of contamination with UBC-GFP+ hematopoietic cells in the input population that could have expanded in co-culture with Mesp1-PSC.

7.3. In addition to the confocal imaging showing CD45 expression within the co-aggregate, the authors should include flow cytometry data after culture to enable a more precise characterization of the cell populations and quantification.

7.4. The authors should include the efficiency of EC conversion to HSC. Are significant differences being observed between hemogenic EC and non-hemogenic EC?

7.5. Although EC derived from the IVC were able to originate HSC, the levels of chimerism after transplantation were low, suggesting some EC might not be able to become hemogenic. Is EHT similarly efficient between arterial and venous EC or in adult EC from a distinct organ, such the brain?

7.6. The authors should include the results from the transplants of co-aggregates formed with DsRedneg PSC in Figure4.

7.7. Richard, C et al., (doi: 10.1016/j.devcel.2013.02.011.) have shown that the subaortic mesenchyme controls endothelial Runx1 in the dorsal aorta, enabling EHT. The authors show an upregulation of Runx1 during culture suggesting that a similar mechanism might be occurring. The use of EC isolated from a Runx1 reporter mouse would unequivocally demonstrate that EC turn on endogenous Runx1 expression and undergo EHT and it would allow a quantification of how many non-

8

hemogenic EC undergo reprogramming to hemogenic fate.

7.8. The scRNA-Seq suggests that different hematopoietic lineages are derived during the reprogramming process. Do the authors think that HSC are derived within the 4 days of coculture or an HSC precursor that is able to complete the process of maturation in vivo, or both? Flow cytometric characterization of the co-aggregates and transplantation of the different cell subsets within the co-aggregate could bring more clarity of what cells are endowed with long-term reconstitution ability, or if transplantation of the entire co-aggregate is required to support engraftment after transplantation.

Methods should be written concisely, but should contain all elements necessary to allow interpretation and replication of the results. As a guideline, Methods sections typically do not exceed 3,000 words. The Methods should be divided into subsections listing reagents and techniques. When citing previous methods, accurate references should be provided and any alterations should be noted. Information must be provided about: antibody dilutions, company names, catalogue numbers and clone numbers for monoclonal antibodies; sequences of RNAi and cDNA probes/primers or company names and catalogue numbers if reagents are commercial; cell line names, sources and information on cell line identity and authentication. Animal studies and experiments involving human subjects must be reported in detail, identifying the committees approving the protocols. For studies involving human subjects/samples, a statement must be included confirming that informed consent was obtained. Statistical analyses and information on the reproducibility of experimental results should be provided in a section titled "Statistics and Reproducibility".

All Nature Cell Biology manuscripts submitted on or after March 21 2016 must include a Data availability statement as a separate section after Methods but before references, under the heading

10"Data Availability". For Springer Nature policies on data availability see <http://www.nature.com/authors/policies/availability.html>; for more information on this particular policy see <http://www.nature.com/authors/policies/data/data-availability-statements-data-citations.pdf>. The Data availability statement should include:

- Accession codes for primary datasets (generated during the study under consideration and designated as "primary accessions") and secondary datasets (published datasets reanalysed during the study under consideration, designated as "referenced accessions"). For primary accessions data should be made public to coincide with publication of the manuscript. A list of data types for which submission to community-endorsed public repositories is mandated (including sequence, structure, microarray, deep sequencing data) can be found here <http://www.nature.com/authors/policies/availability.html#data>.
- Unique identifiers (accession codes, DOIs or other unique persistent identifier) and hyperlinks for datasets deposited in an approved repository, but for which data deposition is not mandated (see here for details <http://www.nature.com/sdata/data-policies/repositories>).
- At a minimum, please include a statement confirming that all relevant data are available from the authors, and/or are included with the manuscript (e.g. as source data or supplementary information), listing which data are included (e.g. by figure panels and data types) and mentioning any restrictions on availability.
- If a dataset has a Digital Object Identifier (DOI) as its unique identifier, we strongly encourage including this in the Reference list and citing the dataset in the Methods.

We recommend that you upload the step-by-step protocols used in this manuscript to the Protocol Exchange. More details can be found at www.nature.com/protocolexchange/about.

All imaging data should be accompanied by scale bars, which should be defined in the legend. Cropped images of gels/blots are acceptable, but need to be accompanied by size markers, and to retain visible background signal within the linear range (i.e. should not be saturated). The boundaries of panels with low background have to be demarked with black lines. Splicing of panels should only be considered if unavoidable, and must be clearly marked on the figure, and noted in the legend with a statement on whether the samples were obtained and processed simultaneously. Quantitative comparisons between samples on different gels/blots are discouraged; if this is unavoidable, it should

11only be performed for samples derived from the same experiment with gels/blots were processed in parallel, which needs to be stated in the legend.

The total number of Supplementary Figures (not including the “unprocessed scans” Supplementary Figure) should not exceed the number of main display items (figures and/or tables (see our Guide to Authors and March 2012 editorial <http://www.nature.com/ncb/authors/submit/index.html#suppinfo>; <http://www.nature.com/ncb/journal/v14/n3/index.html#ed>). No restrictions apply to Supplementary Tables or Videos, but we advise authors to be selective in including supplemental data.

GUIDELINES FOR EXPERIMENTAL AND STATISTICAL REPORTING

13REPORTING REQUIREMENTS – We are trying to improve the quality of methods and statistics reporting in our papers. To that end, we are now asking authors to complete a reporting summary that collects information on experimental design and reagents. The Reporting Summary can be found here <https://www.nature.com/documents/nr-reporting-summary.pdf> If you would like to reference the guidance text as you complete the template, please access these flattened versions at <http://www.nature.com/authors/policies/availability.html>.

Author Rebuttal to Initial comments

Editorial comments:

A) Acknowledge and discussion related literature:

Reviewer 2

"Previous studies by Ding <https://doi.org/10.1002/dvdy.23923> with knockout animals revealed that loss of PDGFRA had no effect on fetal liver HSCs. However, present study has found a profound effect of

14Pdgfra knockout or inhibition with APA5 antibody on HSC and blood production. How this could be explained?"

"Another important difference between findings presented in this manuscript and the Ding paper is the developmental window at which contribution of PDGFRa+ cells to blood cells was observed. Ding found no contribution of PDGFRa+ cells to HSCs when tamoxifen was injected at E9.5. In contrast, this study revealed a substantial labeling blood cells following tamoxifen treatment at E9.5. This should be discussed."

Reviewer 3

"The authors show E9.5 labeled Pdgfra+ cells contribute to endothelial cells and HSC in the dorsal aorta. Ding, G. et al., (doi: 10.1002/dvdy.23923) showed by lineage tracing that Pdgfra+ mesoderm labeled at E7.5-E8 contributes to endothelial cells in the dorsal aorta and to adult hematopoietic cells, including KLS cells. The authors should acknowledge these findings in this section and add their comments on how their results add to the previous report."

"The previous findings of Saga, Y. et al, (doi: 10.1016/s1050-1738(01)00069-x.) showing contribution of Mesp1 mesoderm to the dorsal aorta should be acknowledged."

Response: The above have all been acknowledged and discussed in relation to our findings in the revised manuscript. Please see the point-by-point rebuttal to reviewer comments.

B) Improve quantification and statistical analysis:

Reviewer 1

"The authors should quantitate the total number of CFU-F present in the AGM region over the stages E9.5-13.5. It is important to know the total number present in a tissue (at each embryonic stage) and then compare this number to the total number recovered in the various sorted cell populations. Only then can the authors claim (page 6-line 20/21) "Taken together, these data show that PDGRFA marks all CFU-Fs in the E11.5 AGM...."

"The authors show numerous growth curves (example Fig. 1E) but average results of 3 studies. All of the individual growth curves for n=3 should be shown to assess the variance among the individual groups."

"There is no indication how many times the clonal analyses (Fig. 1F) were performed."

Reviewer 3

"...Several results show no statistical analysis, or they lack the reference that no statistical significance was observed..."

Response: We have addressed these requests in the revised manuscript. Importantly, all our data have now been reviewed and analysed by a statistician- Dr Jake Olivier, Professor of Mathematics and Statistics at UNSW Sydney, who is now an author on this manuscript. The raw data corresponding to each figure panel and statistical method and codes applied to generate significance have now been included as supplemental worksheets. The figure legends and methods sections have been updated. Please see the point-by-point rebuttal to reviewer comments.

C) Provide sufficient experimental details as requested by Reviewer 3:

"The authors should include information of how many embryo equivalents per recipient are used for direct transplantation of AGM cells (Figure 2)."

"The authors' findings that co-aggregation of DA endothelium and, particularly adult endothelium with embryo-derived Mesp1der PSC instructs EHT and HSC emergence are very interesting. There are experimental details and inclusion of more data that can clarify/strengthen the findings..."

Response: We have addressed all these points by including new experimental data. Please see the point-by-point rebuttal to reviewer comments.

D) All other referee concerns pertaining to strengthening existing data, providing controls, methodological details, clarifications, and textual changes as applicable should also be addressed.

Response: We have now addressed all such points. Please see the point-by-point rebuttal to reviewer comments.

E) Finally, please pay close attention to our guidelines on statistical and methodological reporting (listed below) as failure to do so may delay the reconsideration of the revised manuscript. In particular please provide:

- a Supplementary Figure including unprocessed images of all gels/blots in the form of a multi-page pdf file. Please ensure that blots/gels are labeled, and the sections presented in the figures are clearly indicated.

Response: We do not have unprocessed gels/blots in this manuscript but have included an Excel file with all numerical source data as a supplementary table as requested.

Reviewer comments (point-by-point response)

Reviewer #1:

Remarks to the Author:

The authors investigated whether cells with mesenchymal stem cell-like activity, which provide an essential niche for long-term HSCs (LT-HSCs) in the bone marrow, reside in the aorta6 gonad-mesonephros (AGM) and contribute to the structural development of the dorsal aorta and endothelial to hematopoietic transition (EHT). Using transgenic mice, they identify a lineage hierarchy

16for AGM stromal cells and traced E10.5/E11.5 aortic endothelium and HSCs to mesoderm derived (Mesp1) PDGFRA+ stromal cells (Mesp1der PSCs). They determined that these cells dominate the sub-endothelial and ventral stroma in the E10.5–E11.5 AGM but by E13.5 were replaced by neural crest (Wnt1) derived PDGFRA+ stromal cells (Wnt1der PSCs). Co-aggregating non-homogenic embryonic and adult endothelial cells with Mesp1der PSCs but not with Wnt1der 13 PSCs resulted in activation of a hematopoietic transcriptional program in embryonic and adult endothelial cells accompanied by EHT and generation of LT-HSCs. Dose-dependent inhibition of PDGFRA signaling or BMP, WNT, NOTCH signaling interrupted this reprogramming event. The authors postulate that such endothelial and PSC interactions may prove beneficial in translating generation of HSC in vitro.

This is novel insights into an area that has been of interest to developmental and stem cell scientists for many years. They directly contributed to new knowledge about the origins of stromal cells that regulate the endothelial cell behaviour in becoming hemogenic endothelium with HSC emergence. The methods used are state of the art and appropriate controls are used throughout. The flow of studies is easy to follow, and the salient points summarized at key steps. The conclusions are supported by the data presented.

A key deficiency in the presentation of the results are the sporadic use of statistical tests. To bolster confidence, the authors should statistically analyze all quantitative data for level of significance.

Response: We apologise for any omissions in this regard. To properly address this concern, we have collaborated with Dr Jake Olivier, Professor of Mathematics and Statistics at UNSW Sydney to perform a root and branch analysis of all the quantitative data.

The raw data corresponding to each relevant figure panel is now included as a separate sheet in the data file (Supplementary Table with numerical source data) along with the relevant statistical tests and codes that was used to assess the level of significance. The figure legends and methods section have also been revised accordingly. Professor Olivier is now a co-author of this manuscript.

There are some questions and points to consider:

1) The authors should quantitate the total number of CFU-F present in the AGM region over the stages E9.5-13.5. It is important to know the total number present in a tissue (at each embryonic stage) and then compare this number to the total number recovered in the various sorted cell populations. Only then can the authors claim (page 6-line 20/21) "Taken together, these data show that PDGRFA marks all CFU-Fs in the E11.5 AGM....".

Response: Thank you for this important suggestion. We have now assessed bulk CFU-Fs at these embryonic time points (Extended data figure 1B(ii); page 6; lines 1-3). As shown in Figure 1D, bearing in

mind that sorted cells are subjected to additional stress, in contrast to PDGFRA⁺ cells, PDGFRA⁻ cells in the AGM produced no large or small colonies (Figure 1D).

We have rephrased the sentence to “Taken together, these data show that CFU-F potential in the E11.5 AGM resides largely within PDGFRA⁺ cells and that *Nestin* expression marks a sub-population of PDGFRA⁺ cells with more restricted CFU-F and differentiation potential” (page 6; lines 20-23).

2) The authors show numerous growth curves (example Fig. 1E) but average results of 3 studies. All of the individual growth curves for n=3 should be shown to assess the variance among the individual groups.

Response: Thank you for this suggestion. We have revised all the relevant figures (Figure 1E, 4C (iii), Extended data figures 1H and 2C(iii)) accordingly.

3) There is no indication how many times the clonal analyses (Fig. 1F) were performed.

Response: We apologise for this omission. The primary colonies were from triplicate experiments and 144- 385 single cells were plated from 54 micro or 30 small or 27 large colonies, to assess secondary to quaternary CFU-Fs.

This information is now summarised in the figure legend and detailed in the methods and supplemental numerical source data table.

4) Fig. 1J and Movie 2 do not convince this reviewer that the cells formed lumenized vessel structures in vivo. One would need to see intraluminal red blood cells or evidence of systemically infused fluorescent dyes to be sure these are vessels.

Response: We performed this experiment by mixing 250,000 PDGFRA⁺ *Nestin*-GFP⁻CD31⁻PDGFRB⁻ FACsorted cells from E11.5 *Nestin*-GFP⁺ AGMs with matrigel (100uL), followed by sub-cutaneous implantation in the nuchal and inguinal regions of mice. Vascularised plugs were removed at 4-weeks post-implantation, fixed with 4% PFA, washed and embedded in OCT prior to cryosection, immunofluorescence staining and confocal microscopy. Unfortunately, most intraluminal red blood cells would have been lost during processing and perfusion of fluorescent dyes in conjunction with live imaging had not been planned. However, we have now included longitudinal and cross sections of lumenized vessel-like structures (Extended data Figure 1I) to support the data in Figure 1J. We have also amended the text from vessels to vessel-like structures (page 7; line 15).

5) Fig. 2C(ii) does not indicate which are primary or secondary recipient mice.

Response: We apologise for this omission. This Figure panel (now Fig. 2D (ii)) has been appropriately labelled.

6) The doses used in combination in Ext. data Fig. 6C (i-ii) were near dose limiting when used as single

agents. One would question whether the inhibition of CFU-C and LT-HSC activity is because all of the cells died. Some evidence of cell viability at those combined doses (or lowered combined doses that assure cell viability) are required.

Response: Thank you for this observation and comment. Cell viability was assessed by flow cytometry- there was no loss of viability with inhibitors even at the highest concentration (n=3).

This finding is presented in Extended data Figure 6C (i) of the revised manuscript (page 19, lines 8-10).

Reviewer #2:

Remarks to the Author:

In this manuscript, Chandrakanthan et al. investigated the origin and properties of stromal cells in AGM region. They show that periaortic stromal cells in this location are derived from Mesp1+ progenitors (Mesp1der) but later replaced by stromal cells derived from neural crest (Wnt1der). Coaggregation of Mesp1der or Wnt1der stromal cells with endothelial cells revealed that aggregates of Mesp1der stromal cells with AGM endothelium including nonhemogenic E13.5 endothelium and endothelium from hearts and aorta of 8-12 weeks old animals produced HSCs with robust engraftment potential. This effect was observed only with E10.5 or E11.5 Mesp1der stromal cells but not with E13.5 Wnt1der stromal cells. In addition, authors demonstrated that treatment of endothelial/stromal aggregates or ex vivo cultured E10.5 embryos with PDGFRa inhibitor reduced blood and HSC formation. The differences in Mesp1der stromal cells from ventral and dorsal fraction of were also revealed. Overall, these studies are well executed and make a significant novel contribution to our understanding of HSC development.

1. Previous studies by Ding <https://doi.org/10.1002/dvdy.23923> with knockout animals revealed that loss of PDGFRa had no effect on fetal liver HSCs. However, present study has found a profound effect of Pdgfra knockout or inhibition with APA5 antibody on HSC and blood production. How this could be explained?

Response: Thank you for this comment. Using Pdgfra-nGFP KI embryos, we showed that E10.5 and E11.5 AGM lacked CFU-Cs and transplantable HSCs in homozygotes (Extended data figure 2L (i)-(iii)). Ding et al (ref #36 in the submission) did not investigate AGM hematopoiesis or HSCs per se. Ding et al showed that FL LSK numbers in heterozygous and homozygous E12.5 Pdgfra-mer-Cre-mer KI were comparable by flow cytometry. It is important to note that the cellular composition of the FL LSK population is functionally heterogeneous (containing a mix of progenitor cells and HSCs), only a small fraction of these cells are HSCs (50-60; ref 18#). Because the functional status of Pdgfra-null HSCs was not investigated by Ding et al, it is not possible to conclude if HSCs were formed (or functionally healthy). For example, yolk sac-derived progenitors are thought to contribute to the FL LSK compartment, it is conceivable that in the absence HSCs the E12.5 FL LSKs could appear immunophenotypically normal.

We have expanded the discussion to elaborate these differences in the revised manuscript (page 23/24, lines 22-24/1-5).

2. Another important difference between findings presented in this manuscript and the Ding paper is the developmental window at which contribution of PDGFRa+ cells to blood cells was observed. Ding found no contribution of PDGFRa+ cells to HSCs when tamoxifen was injected at E9.5. In contrast, this study revealed a substantial labelling blood cells following tamoxifen treatment at E9.5. This should be discussed.

Response: Thank you for this comment. Ding et al showed that PDGFRA mesodermal contributions to FL and adult hematopoiesis was limited to cre-activation at E7.5-E8.5, whereas our data clearly showed PDGFRA contributions to AGM endothelium, and transplantable HSCs, when *Pdgfra* Cre-recombinase was activated at E9.5. These variations could be due to differences in cre-recombinase efficiency or subtle variations in fitness of reporter PDGFRA +/+ mesodermal cells (our transgenic model) and reporter PDGFRA KI/+ cells (Ding et al; ref #36).

We have included a consideration of this point in the revised manuscript (page 23, lines 11-16).

3. PDGFRa-positive cells in the AGM region of E9.5 embryos could still contain multipotential precursors, including Flk1+ mesodermal precursors which are known to contribute to HSC in the embryo proper. What is the proportion of FLK1+ cells within PDGFRa+ cells at E9.5 in the mouse model used in this study?

Response: The mesoderm of gastrulating mouse embryos has populations of cells that express FLK1+/PDGFRA+ (ref#71; Kataoka et al Development, Growth & Differentiation 1997) and their frequency decreases as the embryo develops (E8.5; 1.4% and E9.5; 0.2% by surface immunophenotyping; ref#36; Ding et al Developmental Dynamics 2013). In our model, FLK1+/Pdgfra-nGFP (GFP half-life; ~ 24-36hrs) and FLK1/Pdgfra expression at E9.5 was comparable (Rebuttal Figure A; 1.5% and 0.4% respectively). These data are in line with *Pdgfra* and *Flk1* expression in published single cell RNA-seq datasets of sorted endothelial (CD45⁻CD31⁺CD144⁺) and control (CD45⁻CD31⁻CD144⁻) cells, which show that *Pdgfra* HIGH control cells express low *Flk1* at E9.5 and that *Flk1* HIGH cells show endothelial commitment (Rebuttal Figure B; PMID:[32203131](https://pubmed.ncbi.nlm.nih.gov/32203131/); GEO- [GSM4140380](https://ncbi.nlm.nih.gov/geo/query/acc.cgi?acc=GSM4140380)).

4. Tracing experiments demonstrated that blood and hemogenic endothelium originate from PDGFR⁺ cells found in periaortic mesenchyme. Any evidence that these cells possess hematopoietic potential in vitro, for example in coculture with OP9?

Response: Thank you for this suggestion. We have now performed OP9 co-culture assays with E9.5 whole AGM and FACS-purified PSCs for orthogonal validation of our in vivo lineage tracing and transplant experiments and show robust CD31⁺/45⁺ generation. These data are presented in Figure 2B (i)-(iii) of the revised manuscript.

5. How embryos were exposed to APA5 and APB5 antibodies. Were they injected with antibodies or antibodies simply added to ex vivo cultures? In the later scenario, any evidence that antibodies can reach AGM by diffusion?

Response: E10 *ex vivo* embryo cultures were performed only with APA5 (Figure 6C (i)-(iii)). The antibody was added to cultures and AGM staining was confirmed in E11.5 cryosections. This is in keeping with a previous study (Takakura et al, *PDGFR alpha expression during mouse embryogenesis: immunolocalization analyzed by whole-mount immunohistostaining using the monoclonal anti-mouse PDGFR alpha antibody APA5*) which demonstrated that APA5 could deeply penetrate E6.5-16.5 embryos incubated with the antibody.

We have included this reference (Supplementary Information ref #6) in the revised manuscript.

6. Page 11 lines 20-23. Pdgfra-nGFP⁺/Mesp1-DsRed mice are introduced here for the first time. However, it is unclear whether Mesp1-DsRed is knock-in reporter, or it traces Mesp1- Cre recombination. If Mesp1-DsRed is knock-in reporter, why DsRed signal is seen in blood and endothelial cells. Expression of Mesp1 is downregulated in endothelial and blood cells.

Response: Thank you for pointing this out and we apologise for the confusion. These images were from an E11.5 AGM dissected from a compound transgenic mouse- Mesp1-cre x ZRed; Pdgfra-nGFP and show cells that were derived from *Mesp1* expressing cells (rather than those currently expressing *Mesp1*) in

conjunction with those currently/recently expressing *Pdgfra* (*Pdgfra*-nGFP (KI); GFP half-life ~ 24-36 hours).

In the revised manuscript, we have included a schematic showing the mouse cross along with the image (Extended data figure 3B (i)) for clarity.

7. Transcriptional analysis of adult heart and aortic endothelium before and after coaggregation would be very interesting.

Response: Thank you. The single cell transcriptomic analyses were performed using adult heart endothelium before and after (24hrs, 48hrs, 72hrs, 96hrs) co-aggregation with *Mesp1*^{der} PSCs (Figure 5D-G).

8. Please spell out PSC abbreviation.

Response: Thank you. We have ensured that PSC is defined at first use both in the abstract and main text (page 4; lines 6/7).

Reviewer #3:

Remarks to the Author:

Chandrakanthan, V. et al. investigated the origin and contribution of *Pdgfra*⁺ cells to hematopoietic stem cell (HSC) development in the dorsal aorta. The study shows *Pdgfra*⁺ cells from mesodermal origin contribute to endothelial cells and HSC in the dorsal aorta, and to sub-endothelial mesenchyme at the time HSC emerge. This mesoderm-derived mesenchyme supports endothelial-to-hematopoietic transition (EHT) from hemogenic endothelium and is later replaced by neural crest-derived *Pdgfra*⁺ cells that lack this ability, coinciding with the cessation of hematopoiesis in the DA. Mesoderm-derived *Pdgfra*⁺ cells not only enable EHT from hemogenic endothelium but also from non-hemogenic embryonic and adult endothelial cells, in a process dependent on *Pdgfra* signaling and BMP, WNT and Notch pathways.

The manuscript includes interesting, novel findings and builds upon previous reports, such as the contribution of *Mesp1*-mesoderm and *Pdgfra*⁺ cells to the dorsal aorta and to adult hematopoiesis, with transplantation assays that bring more robustness to the previous reports (references below) that were based on lineage tracing. The findings that *Mesp1*-derived PSC can instruct adult EC to undergo EHT and generate LT-HSC in vitro are interesting and significant, though this reviewer suggests the inclusion of more experimental details and the performance of experiments that might confer more robustness to the findings. Several results show no statistical analysis, or they lack the reference that no statistical significance was observed. Overall, the manuscript is well written and presents high-quality and significant data, but the clarification of some points and the performance of additional experiments,

22suggested below, may confer more robustness and validity to the findings and conclusions of the manuscript.

1. The authors show E9.5 labeled *Pdgfra*⁺ cells contribute to endothelial cells and HSC in the dorsal aorta. Ding, G. et al., (doi: 10.1002/dvdy.23923) showed by lineage tracing that *Pdgfra*⁺ mesoderm labeled at E7.5-E8 contributes to endothelial cells in the dorsal aorta and to adult hematopoietic cells, including KLS cells. The authors should acknowledge these findings in this section and add their comments on how their results add to the previous report.

Response: As the reviewer correctly points out, Ding et al (ref #36 in the original submission) used PDGFRa-MCM-LacZ embryos labeled at E7.5-8 and showed LacZ positivity in the E10.5 AGM including blood cells budding from the endothelial cell layer. They also labelled E7.5 PDGFRa-MCM-YFP embryos and showed YFP staining in B-, T- and KLS cells in the bone marrow of adult mice. We show that PDGFRA⁺ cells labelled at E9.5 also contribute to YFP⁺ AGM endothelial and blood clusters at E11.5 and that these AGM cells include transplantable HSCs (primary/secondary transplants).

We have now discussed our findings in relation to those by Ding et al (page 8, lines 4-6 and page 23, lines 11-16).

2. Rybtsov, S. et al. (doi.org/10.1084/jem.20102419) showed that at E11.5 cells with long-term repopulating ability reside within the VE-Cadherin⁺CD45⁺ population. Are all CD31⁺CD45⁺ labelled with *Pdgfra*-eYFP at E11.5 (Figure2)?

Response: When *Pdgfra*⁺ cells were labelled at E9.5, approximately 35% of E11.5 AGM CD31⁺/CD45⁺ cells were eYFP⁺ (Extended data figure 2F (i)-(ii)). E7.5-E8.5 labelled *Pdgfra*⁺ cells also contribute to AGM endothelial and hematopoietic progenitors at E11.5 (Ding et al; ref #36) but these would not have been captured by the E9.5 lineage trace. We have added these details to the revised text (page 9, lines 6-7/9-10). Thank you.

3. The authors should include information of how many embryo equivalents per recipient are used for direct transplantation of AGM cells (Figure 2).

Response: Thank you. We used 1 embryo equivalent/recipient throughout and have now included this information in the figure/legend and methods.

4. The previous findings of Saga, Y. et al, (doi: 10.1016/s1050-1738(01)00069-x.) showing contribution of *Mesp1* mesoderm to the dorsal aorta should be acknowledged.

Response: This has been addressed in the revised manuscript (page 11, lines 15-16).

5. Based on the confocal image results in Figure3 it appears *Mesp-1* derived cells, though more abundant ventrally, are also present dorsally. The transplants performed assessing the role of *Mesp1*-

DsRed+Pdgfra+ isolated from dorsal and ventral aorta suggest PSC cells are located both dorsally and ventrally. Do the authors see differences in numbers and a preferential localization in the ventral aorta?

Response: The reviewer is correct. PDGFRA⁺; *Mesp1*-DsRed cells were seven-fold more abundant in the ventral half of the dorsal aorta (AoV) compared with the dorsal half (AoD). We have now included confocal and flow cytometry data to clarify and enumerate this difference (Extended data figure 5E (i)-(ii); page 16, lines 18-20).

6. Most Pdgfra+ cells present in the aorta at E11.5 are not derived from *Mesp1*-mesoderm or *Wnt1* neural crest. Are *Mesp1*-derived PSC more limited in differentiation potential compared to *Mesp1*neg PSC? Assessment of Nestin expression in *Mesp1*der PSCs should be shown.

Response: The reviewer is correct. Although the origins of these non-*Mesp1* and non-*Wnt1* PSCs are not clear, *Mesp1* and *Wnt1* derived PSCs collectively accounted for most CFU-Fs in the AGM (Figure 3A (iv), 3B (iv) and Extended data figure 3F(ii), page 12, lines 21-24). The *in vitro* differentiation potential of *Mesp1*^{der}PSCs was broader than that of *Wnt1*^{der}PSCs (Extended data figure 3G). Furthermore, *Mesp1*^{der}PSCs and *Wnt1*^{der}PSCs each mirrored the differentiation potential of non-*Wnt1*^{der}PSCs and non-*Mesp1*^{der}PSCs respectively. These data are now included in the revised manuscript (Extended data figure 3G).

We have also now included E11.5 AGM confocal microscopy and flow cytometry data from *Mesp1*-DsRed; *Pdgfra*-nGFP and from *Mesp1*-DsRed; *Nestin*-GFP compound transgenic mice side by side for the readers to appreciate the distribution of *Pdgfra*-nGFP and *Nestin*-GFP in *Mesp1*-DsRed embryos (Extended data figure 3B). Flow cytometry panels from *Nestin*-GFP; *Mesp1*-DsRed compound transgenic embryos are included to show *Nestin*-GFP expression in *Mesp1*^{der}PSCs (Extended data figure 3B, page 12, lines 1-5).

7. The authors' findings that co-aggregation of DA endothelium and, particularly adult endothelium with embryo derived *Mesp1*der PSC instructs EHT and HSC emergence are very interesting. There are experimental details and inclusion of more data that can clarify/strengthen the findings:

7.1. There is inconsistency in the cell numbers presented to form the co-aggregates in the figure legends and the methods section.

Response: We apologise - the numbers in the figure legend were correct and those in the Methods section have been corrected accordingly (Supplementary Information page 18, line 12-14).

7.2. The authors should include the flow plots showing how the sorting strategy for isolation of EC and DsRed+ PSC was performed (including the fluorophores used), as well as transplants with freshly isolated EC populations to discard any possibility of contamination with UBC-GFP+ hematopoietic cells in the input population that could have expanded in co-culture with *Mesp1*-PSC.

Response: Thank you for this suggestion. The sorting strategy (Extended data figure 4C(i)-(ii)) is now included.

We agree with the reviewer that transplants with freshly isolated EC populations is an important control. We had included this control with our co-aggregate transplants but had inadvertently left this out of the original figure. This has been corrected in the revised manuscript (Figure 4A (iv)).

7.3. In addition to the confocal imaging showing CD45 expression within the co-aggregate, the authors should include flow cytometry data after culture to enable a more precise characterization of the cell populations and quantification.

Response: Thank you for this suggestion. These data are now included (Extended data figure 4D).

7.4. The authors should include the efficiency of EC conversion to HSC. Are significant differences being observed between hemogenic EC and non-hemogenic EC?

Response: The efficiencies of various input EC to HSC conversion (calculated from transplantation assays using serial dilutions of co-aggregates and re-aggregates) are now shown along with the statistical significance of any observed difference between input EC type (Extended data figure 5B(i)-(ii)).

7.5. Although EC derived from the IVC were able to originate HSC, the levels of chimerism after transplantation were low, suggesting some EC might not be able to become hemogenic. Is EHT similarly efficient between arterial and venous EC or in adult EC from a distinct organ, such the brain?

Response: The number of starting endothelial cells in each co-aggregate was 25,000 and recipients in Figure 4A (iv) were all injected with one co-aggregate/recipient. E10.5 and E11.5 whole AGM re-aggregates showed the highest donor chimerism (51-69% and 54-71%) respectively. From purified EC/PSC co-aggregates, donor chimerism varied from 42-58% (E10.5 AGM ECs/10.5 PSCs) to 2-8% (adult IVC EC/ 10.5 or 11.5 PSCs) with those from the E13.5 AGM, adult Heart, adult Aorta ECs ranging between these extremes. We have now performed co-aggregate cultures with ECs purified from adult lung (as a distinct organ) with E10.5 and E11.5 PSC followed by CFU-C and transplantation assays as suggested. Donor chimerism was comparable with that seen with E11.5AGM ECs and PSCs (Figure 4A (iv)).

Taken together, although it is possible that some endothelial cells may not be amenable to EHT by co-aggregation with PSCs, those tested by us were all amenable to transformation, albeit at varying efficiency. These data were suggestive of inherent differences in endothelial cell types that made them more or less receptive to *Mesp1*^{der}PSC induced transformation and worthy of further investigation although beyond the scope of this manuscript. We have included this interpretation in the revised manuscript (page 16, lines 1-3).

7.6. The authors should include the results from the transplants of co-aggregates formed with DsRedneg PSC in Figure 4.

Response: These data were inadvertently left out and have been included in the revised manuscript (Figure 4A (iv)).

7.7. Richard, C et al., (doi: 10.1016/j.devcel.2013.02.011.) have shown that the subaortic mesenchyme controls endothelial Runx1 in the dorsal aorta, enabling EHT. The authors show an upregulation of Runx1 during culture suggesting that a similar mechanism might be occurring. The use of EC isolated from a Runx1 reporter mouse would unequivocally demonstrate that EC turn on endogenous Runx1 expression and undergo EHT and it would allow a quantification of how many non-hemogenic EC undergo reprogramming to hemogenic fate.

Response: Although we agree that such a reporter mouse could serve as an excellent tool to demonstrate *Runx1* expression in EC following co-aggregation with PSCs, we do not have this at our disposal. However, our single cell gene expression data already serves this purpose. We tracked endogenous *Runx1* transcripts in FACS purified CD31+/VE-CAD+ ECs harvested from PSC co-aggregates over 96 hours (Figure 5D-H). As *Runx1* expression is switched on in ECs, there is a concomitant decrease in *Pecam 1* (CD31)/*Cdh5* (VE-CAD) gene expression (although the corresponding proteins linger on the cell surface allowing FACS based isolation) and increase in *Itga2b* (CD41), and *Ptpcr* (CD45) expression in *Runx1* expressing cells.

We have now included a new heat-map showing these changes in single cells (Figure 5H) and used these data to quantify the number of non-hemogenic cardiac endothelial cells that underwent reprogramming to a hemogenic fate when co-aggregated with E11.5 *Mesp1*^{der}PSCs. 12% and 21% of ECs expressed *Runx1* at >1 count per ten thousand (cptt) or >0 cptt respectively (page 20, lines 16-18).

7.8. The scRNA-Seq suggests that different hematopoietic lineages are derived during the reprogramming process. Do the authors think that HSC are derived within the 4 days of coculture or an HSC precursor that is able to complete the process of maturation *in vivo*, or both? Flow cytometric characterization of the co-aggregates and transplantation of the different cell subsets within the co-aggregate could bring more clarity of what cells are endowed with long-term reconstitution ability, or if transplantation of the entire co-aggregate is required to support engraftment after transplantation.

Response: Thank you for this excellent suggestion. We have included flow cytometry data of the co-aggregates (Extended data figure 4D) and performed transplantation assays to investigate the contributions of EC (UBC-GFP) derived CD45⁺ and CD45⁻ cells with/without *Mesp1*-DsRed PSCs to donor chimerism (Figure 4B (i)-(ii)). We were able to conclude the following (i) endothelial cell derived (GFP+) CD45⁺ cells in the co-aggregates could engraft and contribute to long-term reconstitution without requiring on-going support from *Mesp1*-DsRed PSCs. (ii) co-transplantation of *Mesp1*-DsRed PSCs could not facilitate the engraftment/*in vivo* maturation of CD45⁻ cells. (iii) There was a modest increase in donor chimerism when CD45⁺ cells, were co-transplanted with *Mesp1*-DsRed PSCs (page 16, lines 4-16).

Decision Letter, first revision:

1st April 2022

Dear Dr. Pimanda,

Thank you for submitting your revised manuscript "Mesoderm-Derived PDGFRA+ Cells Regulate the Emergence of Hematopoietic Stem Cells in the Dorsal Aorta." (NCB-P46101A). It has now been seen by the original referees and their comments are below. The reviewers find that the paper has improved in revision, and therefore we'll be happy in principle to publish it in Nature Cell Biology, pending minor revisions to satisfy the referees' final requests and to comply with our editorial and formatting guidelines.

Thank you again for your interest in Nature Cell Biology Please do not hesitate to contact me if you have any questions.

Best wishes,

Stelios

Stylianos Lefkopoulos, PhD
He/him/his
Associate Editor
Nature Cell Biology
Springer Nature
Heidelberger Platz 3, 14197 Berlin, Germany

E-mail: stylianos.lefkopoulos@springernature.com

Twitter: @s_lefkopoulos

Reviewer #1 (Remarks to the Author):

The authors have revised the manuscript. My questions have been adequately addressed by their responses. The added statistical rigor provides confidence in the results.

27Reviewer #2 (Remarks to the Author):

All my comments are adequately addressed. One minor notice. Fig.4Aiii should mention if all CFU-C were GFP+.

Reviewer #3 (Remarks to the Author):

The authors have replied to the main concerns by improving the statistical analysis, providing more experimental details and quantitative results, and by performing additional experiments. No further experiments are required, however the following modifications in the text will bring more clarification:

1. This reviewer was referring to the acknowledgement of the findings from Saga, Y. et al, 2000 (Mesp1 Expression Is the Earliest Sign of Cardiovascular Development (doi: 10.1016/s1050-1738(01)00069-x.)), where lineage tracing studies show contribution of Mesp1+ cells to the dorsal aorta.
2. The description of the results in Page 12 lines 12-13 and Page 12 line 24-Page 13 lines 1-3 should be more accurate. The authors say "Mesp1-eYFP+ and Mesp1-eYFP- PDGFRA+ cells generated comparable numbers of CFU-Fs (Fig. 3A (iv); Extended Data Fig. 3C)". That is shown in Fig. 3A but not in Extended Fig. 3C, where statistically significant differences are observed between E10.5 AGM CFU-F derived from Mesp1-eYFP+ or Mesp1-eYFP- PDGFRA+ cells. The same comment applies for page 12 line 24, where authors say "However, 1:3 Wnt1-eYFP+ cells were PDGFRA+ (Fig. 3B (iii)) and these cells formed CFU-Fs with comparable efficiency to Mesp1-eYFP+/PDGFRA+ cells (compare Fig. 3B (iv) with Fig. 3A (iv) and Extended Data Fig. 3E with Extended Data Fig. 3C) in both E11.5 and E10.5 embryos.". The number of CFU-F appears to be comparable for colonies from E11.5 AGM (Fig. 3) but not from E10.5 AGM (Extended Fig. 3).
3. In Fig. 4A(ii) GFP is labelled as CD31-GFP, though GFP should be indicative of UBC-GFP+ endothelial cells.
4. Please explain what Mix CFU-C colonies in Fig. 6C(iv) and Extended Fig.2 represent.
5. The authors should include in the methods and figure legends the statistical test used for analysis of the Transplants results.

12th April 2022

Dear Dr. Pimanda,

Thank you for your patience as we've prepared the guidelines for final submission of your Nature Cell Biology manuscript, "Mesoderm-Derived PDGFRA+ Cells Regulate the Emergence of Hematopoietic Stem Cells in the Dorsal Aorta." (NCB-P46101A). Please carefully follow the step-by-step instructions

28provided in the attached file, and add a response in each row of the table to indicate the changes that you have made. Please also check and comment on any additional marked-up edits we have proposed within the text. Ensuring that each point is addressed will help to ensure that your revised manuscript can be swiftly handed over to our production team.

We would like to start working on your revised paper, with all of the requested files and forms, as soon as possible (preferably within one week). Please get in contact with us if you anticipate delays.

In recognition of the time and expertise our reviewers provide to Nature Cell Biology's editorial process, we would like to formally acknowledge their contribution to the external peer review of your manuscript entitled "Mesoderm-Derived PDGFRA+ Cells Regulate the Emergence of Hematopoietic Stem Cells in the Dorsal Aorta.". For those reviewers who give their assent, we will be publishing their names alongside the published article.

Nature Cell Biology offers a Transparent Peer Review option for new original research manuscripts submitted after December 1st, 2019. As part of this initiative, we encourage our authors to support increased transparency into the peer review process by agreeing to have the reviewer comments, author rebuttal letters, and editorial decision letters published as a Supplementary item. When you submit your final files please clearly state in your cover letter whether or not you would like to participate in this initiative. Please note that failure to state your preference will result in delays in accepting your manuscript for publication.

Cover suggestions

As you prepare your final files we encourage you to consider whether you have any images or illustrations that may be appropriate for use on the cover of Nature Cell Biology.

Nature Cell Biology has now transitioned to a unified Rights Collection system which will allow our Author Services team to quickly and easily collect the rights and permissions required to publish your work. Approximately 10 days after your paper is formally accepted, you will receive an email in providing you with a link to complete the grant of rights. If your paper is eligible for Open Access, our Author Services team will also be in touch regarding any additional information that may be required to arrange payment for your article.

Please note that *Nature Cell Biology* is a Transformative Journal (TJ). Authors may publish their research with us through the traditional subscription access route or make their paper immediately open access through payment of an article-processing charge (APC). Authors will not be required to make a final decision about access to their article until it has been accepted. Find out more about Transformative Journals

Please use the following link for uploading these materials:
[REDACTED]

Best regards,

Adam Lipkin
Staff
Nature Cell Biology

30On behalf of

Stylianos Lefkopoulos, PhD
He/him/his
Associate Editor
Nature Cell Biology
Springer Nature
Heidelberger Platz 3, 14197 Berlin, Germany

E-mail: stylianos.lefkopoulos@springernature.com
Twitter: @s_lefkopoulos

Reviewer #1:

Remarks to the Author:

The authors have revised the manuscript. My questions have been adequately addressed by their responses. The added statistical rigor provides confidence in the results.

Reviewer #2:

Remarks to the Author:

All my comments are adequately addressed. One minor notice. Fig.4Aiii should mention if all CFU-C were GFP+.

Reviewer #3:

Remarks to the Author:

The authors have replied to the main concerns by improving the statistical analysis, providing more experimental details and quantitative results, and by performing additional experiments. No further experiments are required, however the following modifications in the text will bring more clarification:

1. This reviewer was referring to the acknowledgement of the findings from Saga, Y. et al, 2000 (Mesp1 Expression Is the Earliest Sign of Cardiovascular Development (doi: 10.1016/s1050-1738(01)00069-x.)), where lineage tracing studies show contribution of Mesp1+ cells to the dorsal aorta.

2. The description of the results in Page 12 lines 12-13 and Page 12 line 24-Page 13 lines 1-3 should be more accurate. The authors say "Mesp1-eYFP+ and Mesp1-eYFP- PDGFRA+ cells generated comparable numbers of CFU-Fs (Fig. 3A (iv); Extended Data Fig. 3C)". That is shown in Fig. 3A but not in Extended Fig. 3C, where statistically significant differences are observed between E10.5 AGM CFU-F derived from Mesp1-eYFP+ or Mesp1-eYFP- PDGFRA+ cells. The same comment applies for page 12 line 24, where authors say "However, 1:3 Wnt1-eYFP+ cells were PDGFRA+ (Fig. 3B (iii)) and

31these cells formed CFU-Fs with comparable efficiency to *Mesp1*-eYFP+/PDGFRA+ cells (compare Fig. 3B (iv) with Fig. 3A (iv) and Extended Data Fig. 3E with Extended Data Fig. 3C) in both E11.5 and E10.5 embryos.". The number of CFU-F appears to be comparable for colonies from E11.5 AGM (Fig. 3) but not from E10.5 AGM (Extended Fig. 3).

3. In Fig. 4A(ii) GFP is labelled as CD31-GFP, though GFP should be indicative of UBC-GFP+ endothelial cells.
4. Please explain what Mix CFU-C colonies in Fig. 6C(iv) and Extended Fig.2 represent.
5. The authors should include in the methods and figure legends the statistical test used for analysis of the Transplants results.

Author Rebuttal, first revision:

Reviewer #1 (Remarks to the Author):

The authors have revised the manuscript. My questions have been adequately addressed by their responses. The added statistical rigor provides confidence in the results.

Thank you

Reviewer #2 (Remarks to the Author):

All my comments are adequately addressed. One minor notice. Fig.4Aiii should mention if all CFU-C were GFP+.

This is a good point. Yes, they were, and this is now mentioned in the revised text (page 14, line15).

Reviewer #3 (Remarks to the Author):

The authors have replied to the main concerns by improving the statistical analysis, providing more experimental details and quantitative results, and by performing additional experiments. No further experiments are required, however the following modifications in the text will bring more clarification: 1. This reviewer was referring to the acknowledgement of the findings from Saga, Y. et al, 2000 (*Mesp1* Expression Is the Earliest Sign of Cardiovascular Development (doi: 10.1016/s1050-1738(01)00069-x.)), where lineage tracing studies show contribution of *Mesp1*+ cells to the dorsal aorta.

Our apologies. We have now included the reference (#49) and revised the text accordingly (page 10, lines 20-22). Thank you.

Original: *Mesp1* is expressed in heart precursor cells and is required for the formation of a single heart tube [Saga Y et al 1999].

New: Lineage tracing studies using *Mesp1*-cre mice have previously shown *Mesp1* derived cell contributions to endothelial cells of the dorsal aorta [Saga Y et al 2000].

2. The description of the results in Page 12 lines 12-13 and Page 12 line 24-Page 13 lines 1-3 should be more accurate. The authors say “*Mesp1*-eYFP+ and *Mesp1*-eYFP- PDGFRA+ cells generated comparable numbers of CFU-Fs (Fig. 3A (iv); Extended Data Fig. 3C)”. That is shown in Fig. 3A but not in Extended Fig. 3C, where statistically significant differences are observed between E10.5 AGM CFU-F derived from *Mesp1*-eYFP+ or *Mesp1*-eYFP- PDGFRA+ cells. The same comment applies for page 12 line 24, where authors say “However, 1:3 *Wnt1*-eYFP+ cells were PDGFRA+ (Fig. 3B (iii)) and these cells formed CFU-Fs with comparable efficiency to *Mesp1*-eYFP+/PDGFRA+ cells (compare Fig. 3B (iv) with Fig. 3A (iv) and Extended Data Fig. 3E with Extended Data Fig. 3C) in both E11.5 and E10.5 embryos.”. The number of CFU-F appears to be comparable for colonies from E11.5 AGM (Fig. 3) but not from E10.5 AGM (Extended Fig. 3).

The reviewer is correct- thank you. We have revised the text accordingly.

Original (page 12 lines 12-13): “However, in both E11.5 and E10.5 embryos, *Mesp1*-eYFP+ and *Mesp1*-eYFP- PDGFRA+ cells generated comparable numbers of CFU-Fs (Fig. 3A (iv); Extended Data Fig. 3C)”.

New (page 11, lines 9-11): “Whereas *Mesp1*-eYFP- PDGFRA+ cells from E10.5 AGMs generated significantly lower numbers of CFU-Fs than *Mesp1*-eYFP+ PDGFRA+ cells (Extended Data Fig.5C), this difference was not observed in E11.5 AGMs (Fig.3A(iv)).”

Original (page 12, line 24/ page 13, lines 1-3): “However, 1:3 *Wnt1*-eYFP+ cells were PDGFRA+ (Fig. 3B (iii)) and these cells formed CFU-Fs with comparable efficiency to *Mesp1*-eYFP+/PDGFRA+ cells (compare Fig. 3B (iv) with Fig. 3A (iv) and Extended Data Fig. 3E with Extended Data Fig. 3C) in both E11.5 and E10.5 embryos.”

New (page 11, lines 22-25): “However, 1:3 *Wnt1*-eYFP+ cells were PDGFRA+ (Fig.3B(iii)) and these cells formed significantly fewer CFU-Fs than *Mesp1*-eYFP+/PDGFRA+ cells at E10.5 (compare Extended Data Fig.5E with 5C) but their contributions were comparable at E11.5 (compare Fig.3B(iv) with Fig.3A(iv)).”

3. In Fig. 4A(ii) GFP is labelled as CD31-GFP, though GFP should be indicative of UBC-GFP+ endothelial cells.

Thank you. The reviewer is correct, and the label in Fig.4A(ii) has been changed accordingly

4. Please explain what Mix CFU-C colonies in Fig. 6C(iv) and Extended Fig.2 represent.

There were a few colonies where it was impossible to differentiate between co-localisation of separate CFU-Cs from an early split from a single CFU-C. Rather than discount these, albeit negligible numbers, they were included under this label for completeness.

We have now defined Mix CFU-C in the legends (Figure 6 (now Figure 7); page 32, lines 2-3), and Extended Data Figure 2 (now Extended Data Figure 4); page 4, lines 9-10). Thank you.

5. The authors should include in the methods and figure legends the statistical test used for analysis of the Transplants results.

These have now been included in the methods (page 51, lines 3-4) and in each of the relevant figure legends.

Final Decision Letter:

Dear John,

I am pleased to inform you that your manuscript, "Mesoderm-Derived PDGFRA+ Cells Regulate the Emergence of Hematopoietic Stem Cells in the Dorsal Aorta.", has now been accepted for publication in Nature Cell Biology. Congratulations to you and your team!

Once your paper has been scheduled for online publication, the Nature press office will be in touch to confirm the details. An online order form for reprints of your paper is available at <https://www.nature.com/reprints/author-reprints.html>. All co-authors, authors' institutions and

34authors' funding agencies can order reprints using the form appropriate to their geographical region.

Please note that *Nature Cell Biology* is a Transformative Journal (TJ). Authors may publish their research with us through the traditional subscription access route or make their paper immediately open access through payment of an article-processing charge (APC). Authors will not be required to make a final decision about access to their article until it has been accepted. Find out more about Transformative Journals

If you have not already done so, we strongly recommend that you upload the step-by-step protocols used in this manuscript to the Protocol Exchange (www.nature.com/protocolexchange), an open online resource established by Nature Protocols that allows researchers to share their detailed experimental know-how. All uploaded protocols are made freely available, assigned DOIs for ease of citation and are fully searchable through nature.com. Protocols and Nature Portfolio journal papers in which they are used can be linked to one another, and this link is clearly and prominently visible in the online versions of both papers. Authors who performed the specific experiments can act as primary authors for the Protocol as they will be best placed to share the methodology details, but the Corresponding Author of the present research paper should be included as one of the authors. By uploading your Protocols to Protocol Exchange, you are enabling researchers to more readily reproduce or adapt the methodology you use, as well as increasing the visibility of your protocols and papers. You can also establish a dedicated page to collect your lab Protocols. Further information can be found at www.nature.com/protocolexchange/about

You can use a single sign-on for all your accounts, view the status of all your manuscript submissions

35and reviews, access usage statistics for your published articles and download a record of your refereeing activity for the Nature Portfolio.

With kind regards,
Stelios

Stylianos Lefkopoulos, PhD
He/him/his
Associate Editor
Nature Cell Biology
Springer Nature
Heidelberger Platz 3, 14197 Berlin, Germany

E-mail: stylianos.lefkopoulos@springernature.com
Twitter: @s_lefkopoulos

** Visit the Springer Nature Editorial and Publishing website at www.springernature.com/editorial-and-publishing-jobs for more information about our career opportunities. If you have any questions please click here.**